# Carbapenem-resistant *Klebsiella pneumoniae* capsular types, antibiotic resistance and virulence factors in China: a longitudinal, multi-centre study

Fupin Hu [1,7], Yuqing Pan[2,7], Heng Li [3,7], Renru Han[1,7], Xiao Liu[3,7], Ruijing Ma[2], Yongqin Wu[2], Heyuan Lun[2], Xiaohua Qin[1], Jiayin Li[2], Aixi Wang[2], Min Zhou[4], Bing Liu[4], Zhemin Zhou [3,5] ✉ & Ping He [2,6] ✉

Epidemiological knowledge of circulating carbapenem-resistant *Klebsiella pneumoniae* (CRKP) is needed to develop effective strategies against this public health threat. Here we present a longitudinal analysis of 1,017 CRKP isolates recovered from patients from 40 hospitals across China between 2016 and 2020. Virulence gene and capsule typing revealed expansion of CRKP capsule type KL64 (59.5%) alongside decreases in KL47 prevalence. Hypervirulent CRKP increased in prevalence from 28.2% in 2016 to 45.7% in 2020. Phylogenetic and spatiotemporal analysis revealed Beijing and Shanghai as transmission hubs accounting for differential geographical prevalence of KL47 and KL64 strains across China. Moderate frequency capsule or O-antigen loss was also detected among isolates. Non-capsular CRKP were more susceptible to phagocytosis, attenuated during mouse infections, but showed increased serum resistance and biofilm formation. These findings give insight into CRKP serotype prevalence and dynamics, revealing the importance of monitoring serotype shifts for the future development of immunological strategies against CRKP infections.

Carbapenem-resistant *Klebsiella pneumoniae* (CRKP) has emerged as a major public health threat[1] due to its multidrug-resistant nature and ability to cause infections with high mortality rates[2]. According to the 2023 China Antimicrobial Surveillance Network (CHINET), *K. pneumoniae* has become the second most commonly isolated bacterium in clinics in China, and the resistance rate to meropenem has steadily increased from 2.9% in 2005 to 30.0% in 2023 (http://www.chinets.com/Data/GermYear). This increase in resistance prevalence has, in turn, limited therapeutic options and increased the demand for novel strategies to control CRKP infection.

Vaccines targeting surface polysaccharide antigens, such as the capsular polysaccharide (CPS, K antigen) or lipopolysaccharide (LPS,

[1]Institute of Antibiotics, Huashan Hospital, Fudan University, Shanghai, China. [2]Department of Immunology and Microbiology, Shanghai Jiao Tong University School of Medicine, Shanghai, China. [3]Key Laboratory of Alkene-carbon Fibers-based Technology & Application for Detection of Major Infectious Diseases, MOE Key Laboratory of Geriatric Diseases and Immunology, Pasteurien College, Suzhou Medical College, Soochow University, Suzhou, China. [4]Department of Pulmonary and Critical Care Medicine, Ruijin Hospital, Shanghai Jiao Tong University School of Medicine, Shanghai, China. [5]National Key Laboratory of Intelligent Tracking and Forecasting for Infectious Diseases, National Institute for Communicable Disease Control and Prevention, Chinese Center for Disease Control and Prevention, Beijing, China. [6]NHC Key Laboratory of Parasite and Vector Biology, National Institute of Parasitic Diseases, Chinese Center for Disease Control and Prevention, Shanghai, China. [7]These authors contributed equally: Fuping Hu, Yuqing Pan, Heng Li, Renru Han, Xiao Liu. ✉e-mail: zmzhou@suda.edu.cn; hpatsh@sjtu.edu.cn

**Fig. 1 | The capsular genotypes and antimicrobial profiling of CRKP isolates collected from 26 Chinese provinces or municipalities, 2016–2020.** **a**, Percentage of K types between different years. The five most common K types (KL64, KL47, KL19, KL25 and KL10) and other K types are shown. **b**, Map of CRKP clinical isolates collected from 26 provinces or municipalities in China. The included areas are overlaid with blue flags if there were <10 CRKP isolates, otherwise with pie charts of the K-type percentages. The size of these pie charts is proportional to the number of CRKP isolates. Histograms around the map show

the proportions of annual K types in certain areas, as indicated by the arrows. The grey bar indicates that the number of CRKP strains in that year was less than 3. **c**, Five common carbapenem resistance genes ($bla_{IMP}$, $bla_{NDM}$, $bla_{VIM}$, $bla_{KPC}$ and $bla_{OXA-48-like}$) were identified in 1,017 CRKP isolates. The number of CRKP isolates in each K type, $n$, is given above. The resistance rates to imipenem, tigecycline, polymyxin B, CZA and amikacin in different K-type strains are shown as a heat map below (see Supplementary Table 2 for details).

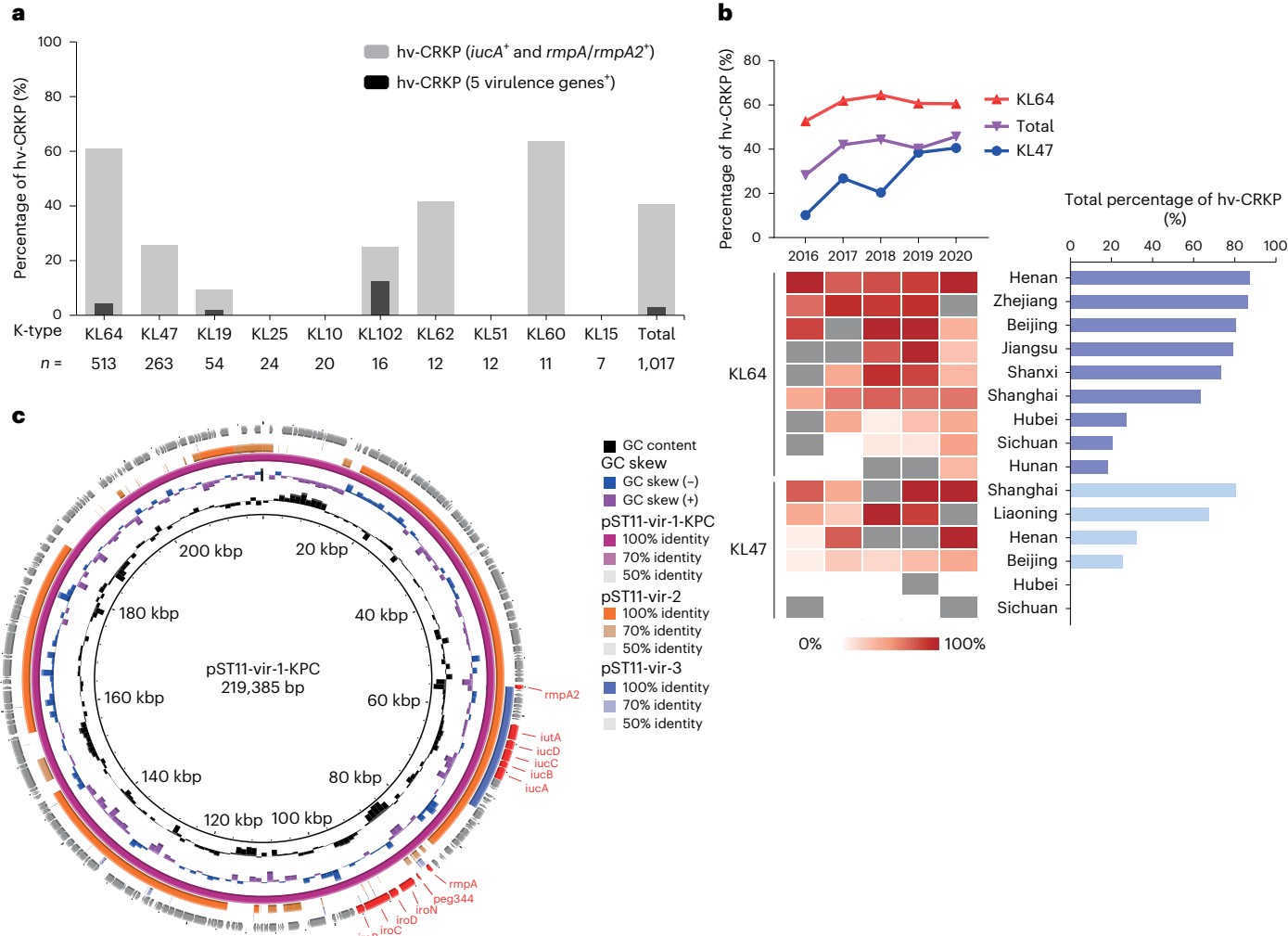

**Fig. 2 | Distribution of AMR and hypervirulence-associated genes among CRKP strains. a**, The proportion of hv-CRKP in different K-type CRKP isolates in China during 2016–2020. The grey bars indicate strains with *iucA* and *rmpA/rmpA2* genes, and the black bars indicate strains with all five virulence genes (*rmpA*, *rmpA2*, *iroB*, *iucA* and *peg344*). The number of CRKP strains of K type, *n*, is given below. **b**, Trends in the proportion of hv-CRKP in all CRKPs and those from KL64 and KL47. Annual and total hv-CRKP rates in KL64 and KL47 strains in different regions are shown below. Grey blocks indicate years with less than three KL64 or KL47 strains. **c**, Circular alignment of virulent plasmids pST11-Vir-2 (CP039526) and pST11-Vir-3 (CP048945) referring to pST11-Vir-1-KPC (AY378100) via BRIG. Representative genes are labelled in the outermost circle, with the hypervirulence-associated genes in red.

O antigen), have been proposed as a promising alternative to antimicrobials for combating multidrug-resistant pathogens[3,4]. Vaccines and monoclonal antibodies have shown protection against *K. pneumoniae* in experimental animal models[5–8] but have not yet progressed beyond clinical trials for human use[9]. Furthermore, *K. pneumoniae* contains at least 79 capsular types (K types) that differ both genetically and phenotypically[3,10], posing a major challenge to achieving broad vaccine coverage. Consequently, there is a need for up-to-date epidemiological insight into the most prevalent and clinically relevant serotypes (type of surface antigens of the organism, such as K or O antigens) to effectively inform and guide vaccine development strategies.

The prevalence of *K. pneumoniae* varies between countries and hosts. In recent years, several studies have reported on the prevalence of *K. pneumoniae* infection in different countries, including the distribution and prevalence of drug resistance, virulence and capsular genotypes[11–13]. However, little is known about its sero-epidemiology in China, mainly due to the limited availability of commercial antibodies for serological typing and the inability of genetic typing to detect phenotypic variation resulting from genetic mutations or disruptions in polysaccharide synthesis-related genes[14–17]. The widespread occurrence and expansion of CRKP-associated infection across China

necessitates a systematic analysis of the population structure, resistance mechanisms and virulence evolution of strains isolated from different areas.

In this Article, we conducted a nationwide, longitudinal, retrospective, epidemiological investigation of CRKP in China. We provided comprehensive data on the distribution and longitudinal evolution of K types, antimicrobial resistance (AMR) and virulence factors profiles among CRKP strains in the country. In particular, we identified a substantial proportion of capsular- or O-antigen-deficient strains in various infection samples and experimentally investigated the potential bases for the acquired survival advantage of these capsular-deficient strains. These findings on serotype prevalence provide initial insight in the development of effective strategies to control CRKP infection.

## Results

### CRKP isolate capsule and antimicrobial profiling

During 2016–2020, a total of 1,017 non-duplicate CRKP isolates were collected from *K. pneumoniae*-infected adult patients admitted to 40 hospitals located in 26 provinces or municipalities in China (Extended Data Fig. 1 and Supplementary Table 1). Antimicrobial susceptibility testing results are summarized in Supplementary Note and

Supplementary Table 2. Carbapenemase genes were detected by PCR in 98.2% (999/1,017) of the CRKP isolates, the vast majority of which were $bla_{KPC-2}$ (89.4%, 909/1,017), followed by $bla_{NDM}$ (4.8%, 49/1,017) and $bla_{OXA-48-like}$ (1.8%, 18/1,017) (Supplementary Table 3). We identified 18 (1.8%) non-carbapenemase-producing CRKP isolates, of which 15 (83.3%) had both the porin (Ompk35/Ompk36) mutation and extended-spectrum β-lactamase genes (Supplementary Table 3).

Capsular genotyping revealed 58 different K loci; the majority belonged to K types KL64 (50.4%) and KL47 (25.9%) (Supplementary Table 4). The frequencies of these two K types varied with time and geography. We observed a significant increase in KL64 from 35.0% in 2016 to 59.6% in 2020, accompanied by a decrease in KL47 from 42.3% to 17.8% (Fig. 1a; P value for trend = $5.2 \times 10^{-5}$, $2.1 \times 10^{-9}$, respectively). Furthermore, KL64 accounted for the majority of the isolates from eastern and central China, including Shanghai, Zhejiang and Sichuan, whereas isolates from northern and northeastern China, that is, Beijing, Liaoning and Tianjin, were predominantly KL47 (Fig. 1b). Other K types were less common (1–54 isolates; Supplementary Table 4) and may be endemic regionally. For example, KL19 accounted for 13.4% of isolates from Shanghai, Zhejiang and Jiangxi in eastern China, while KL10 accounted for 18.1% of isolates from Liaoning and Heilongjiang in northeastern China (Fig. 1b).

We also showed the differences in phenotype of antimicrobial susceptibility and distribution of carbapenemase genes by sequencing-based PCR between K types. The top seven K types were predominantly associated with $bla_{KPC-2}$, which was susceptible to ceftazidime–avibactam (CZA) and resistant to amikacin (except KL19) (Fig. 1c), while 56.5% of the isolates in KL51 and KL60 carried $bla_{OXA-48-like}$. Notably, more than a third of the KL51, KL15 and other rare K-type isolates carried $bla_{NDM}$ and showed resistance to CZA (Fig. 1c).

### Virulence-associated gene carriage in CRKP isolates

We screened for the presence of five hypervirulence-associated genes (*rmpA*, *rmpA2*, *iroB*, *iucA* and *peg344*) by PCR in all CRKP isolates[13,18,19] and classified 413 (40.6%) isolates carrying both *iucA* and *rmpA/rmpA2* genes as hypervirulent CRKPs (hv-CRKPs)[20], including 31 isolates with all five hypervirulence-associated genes. The majority of hv-CRKPs were from K types KL64 (312/413, 75.5%) and KL47 (67/413, 16.2%). Notably, among the top ten K-type strains, KL60 contained the highest proportion of hv-CRKP strains (7/11, 63.6%), followed by KL64 (312/513, 60.8%), KL62 (5/12, 41.7%) and KL47 (67/263, 25.5%) (Fig. 2a).

The proportion of hv-CRKP increased over the five years, from 28.2% in 2016 to 45.7% in 2020 (P value for trend = 0.0061), which could be attributed to both the population expansion of KL64 and a threefold increase in hv-CRKP levels in KL47, from 10.1% in 2016 to 40.5% in 2020 (P value for trend = $1.1 \times 10^{-4}$). By contrast, the frequency of hv-CRKP in KL64 remained stable over the period 2016–2020 (P value for trend = 0.61) (Fig. 2b). High levels of hv-CRKP in KL64 were found in Henan (88.2%), Zhejiang (87.8%), Beijing (81.8%) and Jiangsu (80.7%), while most KL47 strains in Shanghai (81.3%) and Liaoning (69.0%) were hv-CRKP. Conversely, low hv-CRKP frequencies were found in both KL64 (28.2%, 21.2%) and KL47 (0%, 0%) isolates from Hubei and Sichuan throughout the study period. Notably, the prevalence of KL47 hv-CRKPs increased significantly in Liaoning (2016–2019, P value for trend = 0.019) and Beijing (2016–2020, P value for trend = 0.037) (Fig. 2b).

The varying levels of hv-CRKP in KL64 and KL47 strains were partially explained by their associated virulence plasmids (Fig. 2c). Details of the levels of hv-CRKP in KL64 and KL47 strains with associated virulence plasmids are described in Supplementary note.

### Genomic epidemiology of KL47 and KL64 CRKP in China

To obtain a phylogenetic overview of the two most common K-type CRKPs, we performed whole-genome sequencing on 207 KL47 and 319 KL64 isolates and found that >99% (523/526) of them belonged to sequence type ST11 (Supplementary Tables 5 and 6). These 523 ST11 isolates were integrated with additional 3,047 public ST11 genomes to form a global genomic collection of 3,570 isolates. After filtering recombination regions, a maximum likelihood phylogeny of the ST11 lineage was estimated based on the remaining non-repetitive, non-recombinant single-nucleotide polymorphisms in the core genome. We manually separated the tree into four major lineages, of which L2 and L3 were broadly consistent with clonal groups identified based on the cgLIN codes[14] and L1 is a subclade of the L2 lineage that consisted of almost exclusively Chinese isolates (99.1%, 2,111/2,130). Based on kleborate predictions, we found a strong correlation between the 44 K types and 6 O types in ST11, with KL64:O2 and KL47:OL101 accounting for 61% (2,187/3,570) of the strains, especially those in L1. The most common carbapenemase in ST11 was $bla_{KPC-2}$, followed by $bla_{OXA-48}$ and $bla_{NDM-1}$ (Fig. 3a).

### Spatiotemporal transmission of L1 CRKP within China

Most of the strains in lineage L1 were from China (2,111 strains), followed by the USA (7), Oman (3), Singapore (3), and four other countries. Temporal signals were found in lineage L1 based on regression of root-to-tip genetic distance against sampling time ($R_2 = 0.18$, $P = 8.2 \times 10^{-12}$) and further demonstrated based on two date randomization tests (Extended Data Fig. 2a–c and Methods). Therefore, based on the phylogeny above, we estimated the spatiotemporal dynamics of lineage L1 using BactDating and inferred the fluctuation of its effective population sizes using skygrowth in R. The ancestral inter-regional transmissions and serovar changes were also inferred using both TreeTime and the Analyses of Phylogenetics and Evolution (APE) package in R. The most recent common ancestor of L1 was estimated to be present before 1983 (95% confidence interval (95%CI) 1979–1987) in serotype KL47:OL101, whereas the KL64:O2 serotype was estimated to emerge in 2003 (95%CI 2002–2004) as a subclade of L1 and experienced rapid national dissemination in the past decade. Notably, we also predicted 14 other K genotypes in L1, some of which formed local clusters, that is, KL25 in Sichuan and KL110 in Jiangsu (Fig. 3b and Supplementary Table 7).

**Fig. 3 | The population structure of ST11 strains and transmission patterns of the L1 lineage. a**, The population structure of ST11 strains. The metadata is shown on the right-hand side. The grey boxes show the four lineages identified along the tree. CARB, carbapenemase; CGs, clonal groups predicted by cgLIN codes. **b**, The maximum likelihood tree of lineage L1 with branches recalibrated by BactDating. Dates next to particular branches indicate their estimated date of origin with the 95%CI in brackets. Branches are colour-coded according to their estimated regional origin with root nodes of KL47 and KL64 clades shown nearby. Additional geographic estimates of the nodes have been calculated using three models implemented in the APE package in R and visualized in Extended Data Fig. 3. The major regional transmissions (≥8 isolates in the target region) in each of the KL47 (8 events; blue) and KL64 (12 events; orange) clades are shown as grey arcs with serial numbers on the outside. The outer rings show regions, O types and K types, carbapenemase-carrying plasmids, virulence plasmids and each of the five hypervirulence genes from inside to outside, each colour-coded according to the key. The star in the KL64 clade shows a cluster of hv-CRKP isolates, with their times of origin and 95%CI nearby. There were 19 clusters of 14 rare serotypes as predicted by kleborate in lineage L1, which were labelled accordingly. **c**, National distributions of the ST11 L1 strains. Each region is colour-coded according to the percentage of KL64, and the arrows indicate regions that received ≥5 inter-regional transmissions from Shanghai (red) or Beijing (blue), with the thickness proportional to the number of transmissions. Distinct patterns were found between Shanghai and Beijing transmissions. Shanghai strains were mainly transmitted to the eastern and central regions, whereas Beijing strains were transmitted to the northern and northeastern regions, in line with the patterns of inter-regional health seeking in China. **d**, The number of observed transmissions originated from each region extracted from the L1 tree in **b**. The two predominant sources (Shanghai and Beijing) are highlighted in red.

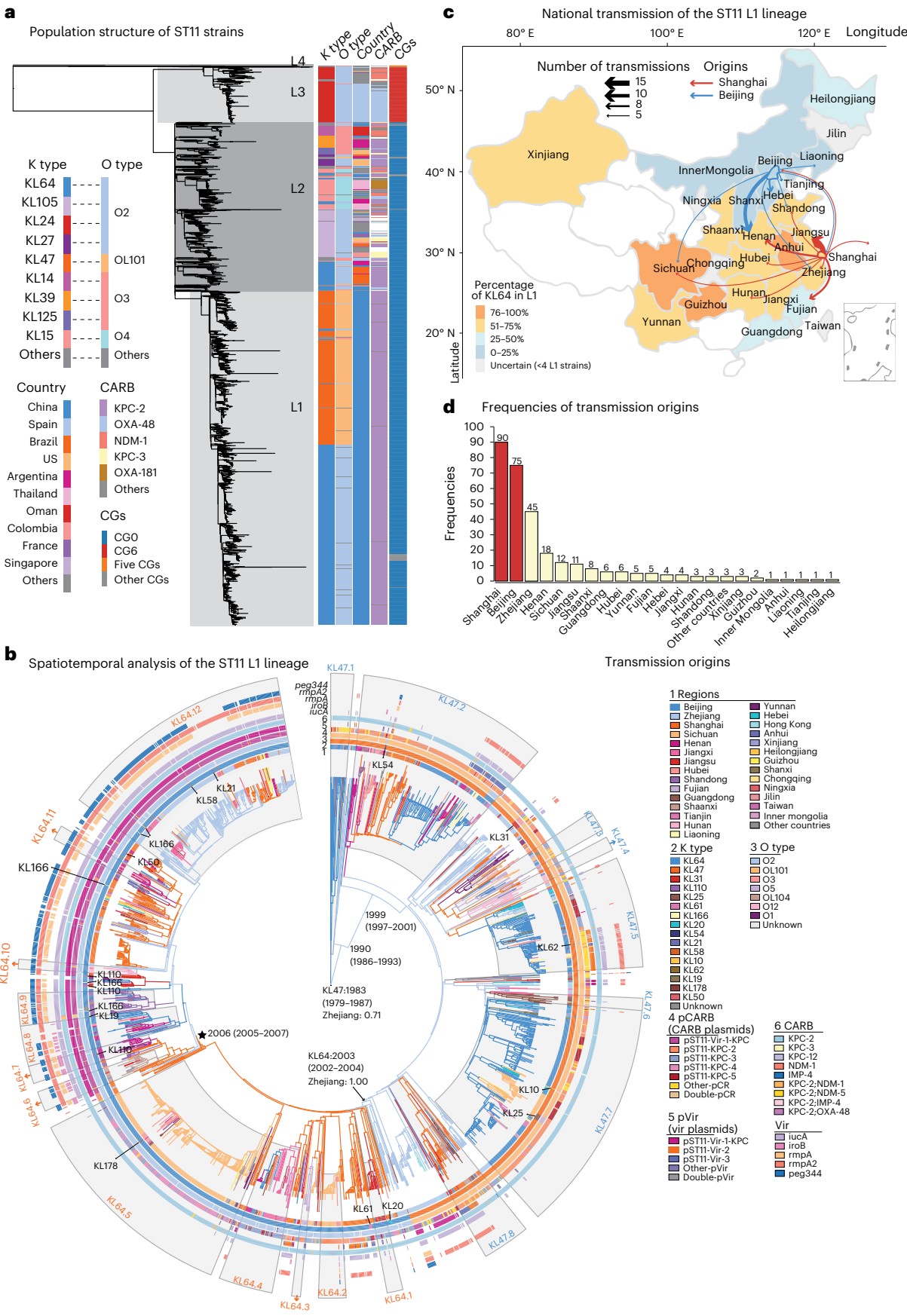

The KL47 strains were estimated to originate in either Zhejiang or Beijing by different models, although all agreed on its long-term persistence in Zhejiang since 1990 (95% CI 1986–1993) (Fig. 3b and Extended Data Fig. 3). By contrast, the KL64 clade was estimated to originate in Zhejiang by all models. We predicted a total of 308 cross-regional transmissions along the phylogeny, which were branches that connect nodes with different geographic inferences by TreeTime. We found that 53.6% (165) of these transmissions had parental nodes in either Beijing or Shanghai, the two major sources of transmissions (Fig. 3c). Among them, Beijing was estimated to contribute 75 transmissions, mostly to northern and northeastern regions, and Shanghai was estimated to contribute 90 transmissions, mainly to eastern and central regions (Fig. 3d).

Furthermore, the majority of the transmissions leave only one or a few isolates in the target regions. We, therefore, focused on 20 major events that were evidenced by ≥8 isolates in the target regions (Fig. 3b). Many of the major transmissions in the KL47 strains involve Beijing and its neighbouring, northern regions such as Henan (event KL47.2), Liaoning (event KL47.2) and Tianjin (events KL47.2 and KL47.7), contributing to the predominance of KL47 clade in northern China (blue route; Fig. 3c). Conversely, KL64 strains were estimated to experience nine major transmissions from Shanghai to central and eastern China including Jiangsu (event KL64.1), Hubei (event KL64.2), Sichuan (event KL64.5) and other regions (red route, Fig. 3c), resulting in their increasing KL64 frequencies.

### Non-capsulated CRKP exist among KL64 and KL47 isolates

All isolates identified as KL64 or KL47 by genotyping were subjected to serologic typing, and 7.4% (38/513) KL64 and 12.2% (32/263) KL47 isolates, respectively, did not react with the corresponding antisera in the Quellung reaction (Extended Data Fig. 4 and Supplementary Table 8). These isolates appeared translucent, showed a significant reduction in polysaccharide capsule uronic acid (Fig. 4a) and were confirmed as NEKp (non-capsulated *K. pneumoniae*) strains by electron microscopy (Fig. 4b). The frequency of NEKp did not change significantly over the study period (Supplementary Table 9; *P* value for trend = 0.97 and 0.99) nor between samples from different body sites (Fig. 4c; *P* = 0.47).

Various mutations, including insertion sequences (ISs), point mutations and deletions, were found in the *cps* region or the *rfaH* gene of NEKp strains, mainly affecting the initial glycosyltransferase genes (*wbaP* for KL47 and *wcaJ* for KL64) (Fig. 4d and Supplementary Table 8). Capsule-deficient phenotypes were replicated in the mutant strains 18–81-KL47Δ*wbaP* and 20–445-KL64Δ*wcaJ*, and their capsule production could be restored upon complementation (Fig. 4a and Extended Data Fig. 4). Complementation with *wcaJ*/*wbaP* also restored capsule production in clinical NEKp strains with corresponding gene disruptions, further supporting the link between *wcaJ*/*wbaP* disruptions and the capsule-deficient phenotype (Extended Data Fig. 4).

To investigate whether such capsule deficiency occurs during in vitro culture in the laboratory, we cultured ten random strains for each of KL47 and KL64 in blood agar plate for 48 h and found very low levels (<0.56%) of capsule-deficient mutants (Extended Data Fig. 5).

In addition, we metagenomically sequenced 49 unprocessed clinical samples with CRKP infections. Among the 11 samples with >5-fold coverage, three had disrupted *cps* genes, including nonsense mutations or ISKpn26 insertions in *wcaJ*, and other disruptions, demonstrating the presence of capsule-deficient mutations in vivo (Fig. 4e and Supplementary Table 10).

### Characterization of NEKp strains under infection pressures

We compared NEKp/capsule-deficient (isogenic Δ*cps*) strains with EKp (encapsulated *K. pneumoniae*) strains in vitro and in vivo to assess the impact of capsule synthesis on fitness. Consistent with previous studies[21], NEKp/Δ*cps* strains showed increased biofilm production and higher susceptibility to macrophage phagocytosis (Fig. 5a–c). Mouse intraperitoneal infection experiments showed that the isogenic Δ*cps* mutants showed reduced lethality and competitive fitness for dissemination to the lung, liver and spleen compared to the parental EKp strains (Fig. 5d–h). Surprisingly, the serum-killing assay revealed a higher proportion of serum-resistant strains in NEKp strains than in EKp strains (Fig. 6a; KL47 *P* = 0.018, KL64 *P* = 2.0 × 10^{−4}). Notably, the isogenic Δ*cps* mutants did not show significant growth advantages over their parents in the Luria–Bertani (LB) media (Extended Data Fig. 6), suggesting that reduced metabolic load due to abolished capsule synthesis is unlikely to be a causative factor. By comparing the antiserum capacity of parental EKp strains, capsule polysaccharide depolymerase-treated (capsule-stripped) EKp strains, isogenic Δ*cps* and Δ*cps*Δ*ops* mutants, we demonstrated that both CPS and O-antigen polysaccharide (OPS) play a critical role in serum resistance (Fig. 6b and Extended Data Fig. 7). In addition, we found that isogenic Δ*cps* mutants (except O⁻ strains) showed increased serum resistance along with enhanced O antigen expression compared to the capsule-stripped EKp strains (Fig. 6b and Extended Data Fig. 7). Transcriptomic and real-time PCR (RT-PCR) analyses also revealed the overexpression of LPS and peptidoglycan synthesis genes in Δ*cps* strains compared to parental strains (Extended Data Fig. 8a). These results suggest that disruption of capsule synthesis enhances the synthesis of O-antigen and ultimately increases antiserum capacity. We speculated that the cessation of CPS synthesis might lead to reduced consumption of undecaprenyl phosphate (Und-P) pool, a limiting factor in the biosynthesis of glycans[22]. This could potentially redirect Und-P for excess O-antigen synthesis. Our observations suggest a potential link between altered Und-P dynamics and changes in serum resistance, as indicated in Extended Data Fig. 8b. Nonetheless, the exact mechanisms and precise Und-P flux alterations following CPS synthesis disruption need to be further investigated.

Furthermore, compared to the parental KL47/KL64 EKp strains, the isogenic Δ*cps* mutants showed higher resistance to heat-inactivated sera collected from patients infected with *K. pneumoniae* KL47/KL64 in the serum bactericidal activity (SBA) assay, suggesting capsule depletion as a potential mechanism for evading humoral immune responses (Fig. 6c).

Collectively, NEKp strains showed a fitness advantage by increasing biofilm formation, serum resistance and anti-SBA but deficiencies in dissemination and anti-phagocytosis (Fig. 6d).

**Fig. 4 | Capsule deficiency in clinical isolates. a**, CPS of NEKp and selected paired EKp strains measured by uronic acid assay. Paired EKp strains were randomly selected from regions and samples similar to those of NEKp. Parental EKp strains and their isogenic Δ*cps* mutants are indicated by arrows. The 20–445Δ*cps* and 18–81Δ*cps* mutants had a significantly lower amount of polysaccharide capsule uronic acid compared to WT strains. The number of EKp and NEKp strains of K type KL64 or KL47, *n*, is given below. Data are shown as the mean ± s.e.m. Each point represents the mean value of three biological replicates of one strain. For clarity, individual error bars for each strain are not shown. Unpaired two-sided Welch's *t*-test was performed for statistical analysis. **b**, Transmission electron microscopy images of representative EKp (KL64 strain 19–241 and KL47 strain 18–729) and NEKp (KL64 strain 19–242 and KL47 strain

18–737) strains. Each electron micrograph shows the representative result of three repetitions. **c**, Frequency of NEKp strains isolated from different sites of infection. The number of CRKP strains in different infection sites is shown below. Pearson's chi-squared test was used to analyse the proportion difference of NEKp strains in different infection samples. **d**, *cps*-locus and *rfaH* gene variants in NEKp isolates. Mutations found in the capsule biosynthesis genes in KL64 and KL47 NEKp isolates. The black flag indicates the location of the mutation. Red, black and blue lines indicate ISs, single-nucleotide polymorphisms and deletions, respectively. *n*, the total number of mutations observed in this gene. **e**, Metagenomic analysis of *cps*-locus variants in unprocessed clinical samples (for details, see Supplementary Table 10). NS, not significant.

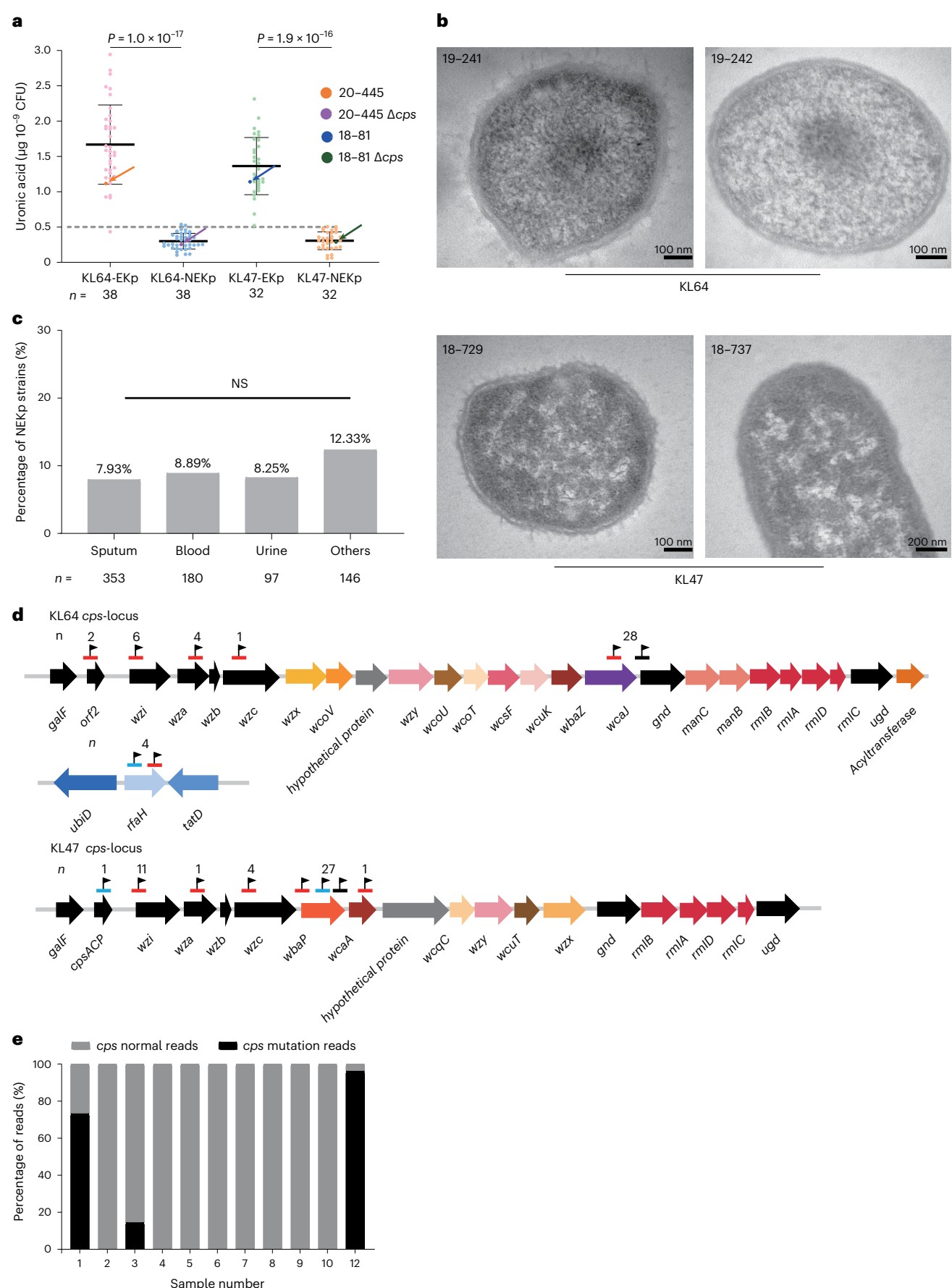

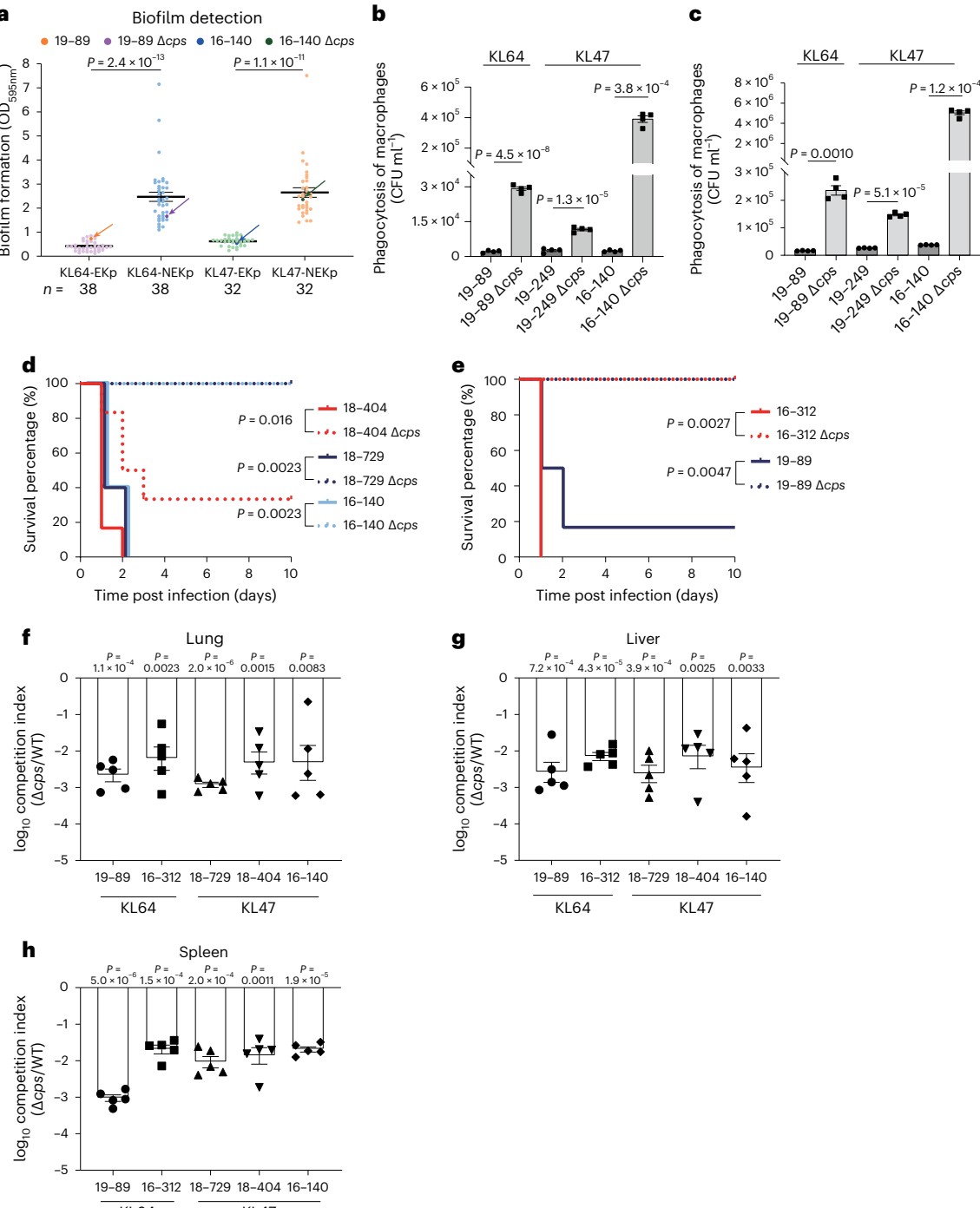

**Fig. 5 | Evaluation of the effect of capsule synthesis on fitness in vitro and in vivo. a**, Biofilm formation of NEKp strains and the selected paired EKp strains. Parental EKp strains and their isogenic *Δcps* mutants are indicated by arrows. Data are shown as the mean ± s.e.m. Each point represents the mean value of three biological replicates of one strain. For clarity, individual error bars for each strain are not shown. Unpaired two-sided Welch's *t*-test was performed for statistical analysis. **b**, Phagocytosis of EKp and NEKp strains by RAW 264.7 macrophages. **c**, Phagocytosis of EKp and NEKp strains by peritoneal macrophages. Peritoneal macrophages were collected from 7-week-old female C57BL/6J mice. In **b** and **c**, WT parental EKp strains were more resistant to phagocytosis than isogenic *Δcps* strains. Data represent the mean ± s.e.m. of four biological replicates of one strain. Unpaired two-sided Student's *t*-test with Welch's correction was performed for statistical analysis in **b** and **c**. **d**–**h**, The capsule of KL47 (**c**,**f**–**h**) and KL64 (**e**,**f**–**h**) *K. pneumoniae* impose competitive

fitness in the mouse intraperitoneal infection model. The KL47 (**d**) and KL64 (**e**) survival rate was evaluated after intraperitoneal infection of C57BL/6J mice with EKp strains or their isogenic *Δcps* mutants at the assigned dose (19–89 or 19–89*Δcps*: $3 \times 10^7$ CFU; 16–312, 18–404, 18–729 and 16–140 or their *Δcps* mutants: $6 \times 10^7$ CFU). Kaplan–Meier analysis and log-rank test was performed for statistical analysis. *n* = 6 (18–404 and 19–89 or their *Δcps* mutants), *n* = 5 (18–729, 16–140 and 16–312 or their *Δcps* mutants). Bacterial load was assessed after intraperitoneal infection of C57BL/6J mice with a 1:1 ratio of WT to isogenic *Δcps* mutant. WT and *Δcps* mutants in the lung (**f**), liver (**g**) and spleen (**h**) were quantified at 24 hpi to calculate the competitive index value. Two-sided one-sample *t*-test was performed to compare the mean of $\log_{10}$-transformed competitive index to a hypothetical value of 0 for each group. Each dot represents an individual mouse. *n* = 5.

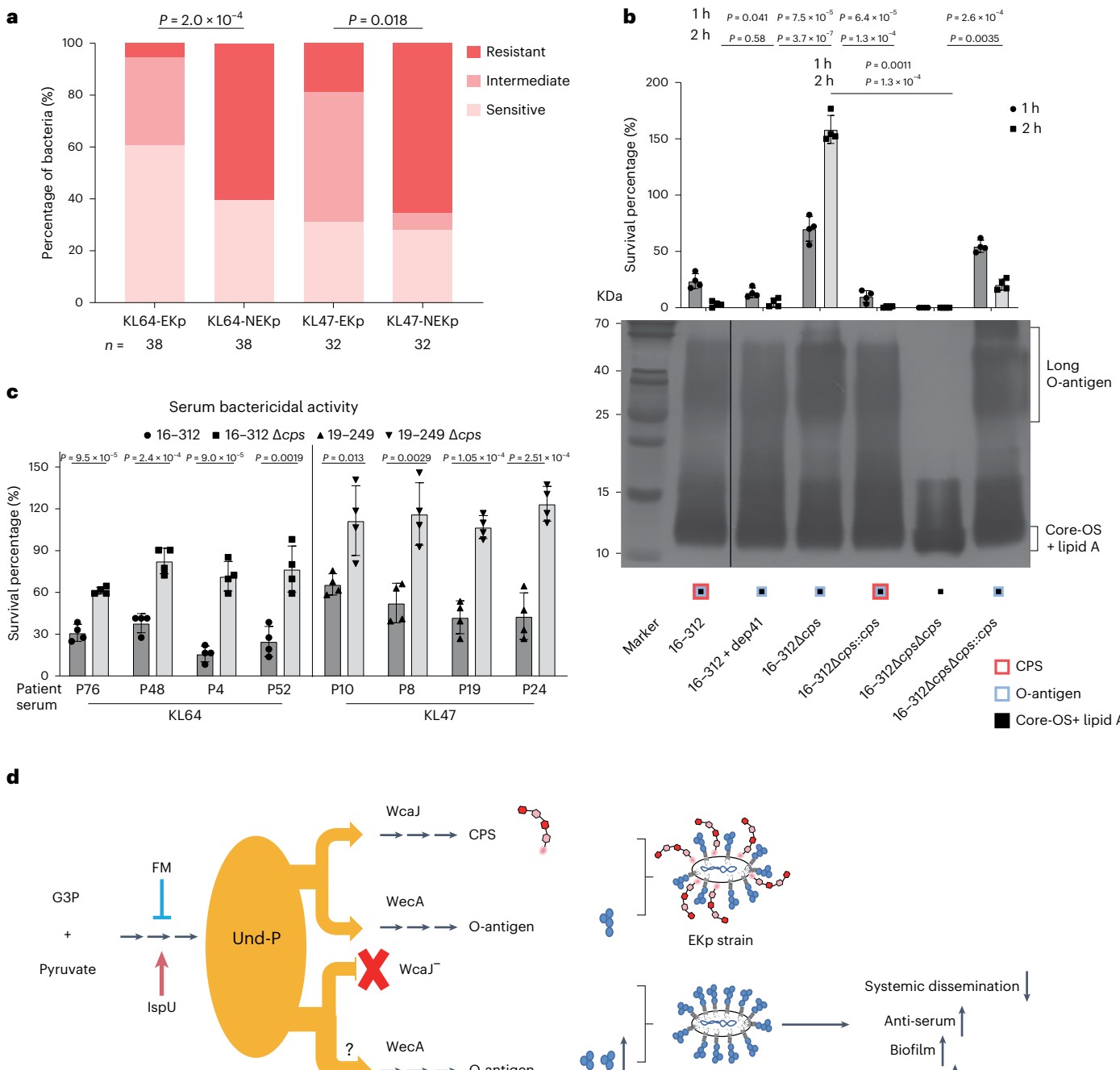

**Fig. 6 | NEKp strains have survival advantages under infection pressure.**
**a**, Distribution of serum resistance levels of NEKp strains and selected paired EKp strains. Bacteria were incubated in 75% baby rabbit serum, and the antiserum levels were determined. Colours represent different levels of serum resistance. The mean rank difference of two groups was analysed by Cochran–Mantel–Haenszel test. *n* indicates the number of strains. **b**, Effect of capsule deficiency on O-antigen expression and its association with serum resistance. Serum-killing assays were performed by incubating the EKp strain 16–312, KL64 CPS depolymerase (Dep41[15])-pretreated strain 16–312 (capsule-stripped), 16–312Δ*cps*, 16–312Δ*cps*Δ*ops* and strains complemented with *cps* or *ops* with 75% baby rabbit serum for 1 or 2 h at 37 °C. The percentage of survival was determined as the number of surviving bacteria relative to the initial bacterial addition. Unpaired two-sided Student's *t*-test with Welch's correction was used for statistical analysis. Data are shown as the mean ± s.e.m. of three biological replicates and represent one experiment of three independent experiments. LPS was prepared by hot phenol–water method and analysed by silver staining. Core-OS, core-

oligosaccharide. **c**, SBA against parental EKp and isogenic Δ*cps* strains by serum from the patients infected with KL47 or KL64 *K. pneumoniae*. *K. pneumoniae* strains were incubated with the heat-inactivated serum from patients for 1 h at 37 °C in the presence of baby rabbit complement. Survival ratios were normalized to the survival strains in the presence of baby rabbit complement alone. Data are the mean ± s.e.m. of four biological replicates. Unpaired two-sided Student's *t*-test was used for statistical analysis. **d**, Model for the surface polysaccharide switch affects *K. pneumoniae* fitness during infection. The de novo biosynthesis of Und-P can be inhibited by fosmidomycin and increased by overexpression of ispU. CPS synthesis gene mutation results in the inability to utilize Und-P, thereby probably increasing the flow of Und-P to O antigen synthesis, with the consequence of increased biofilm formation, maintained or even increased antiserum capacity and facilitated evasion of antibody-mediated killing, although deficient in dissemination. Gene ispU encodes the undecaprenyl-pyrophosphate synthase; G3P, glyceraldehyde-3-phosphate.

## O-antigen-deficient strains in KL64 or KL47 CRKP isolates

Based on the O-antigen-specific phage and silver staining assay, we observed a prevalence of O⁻ strains in both KL47 (13/64, 20.3%) and KL64 (22/76, 29.0%) CRKPs (Extended Data Figs. 9a–e and 10a and Supplementary Table 8). Sequence analysis revealed IS insertions in the *rfb* region of the O⁻ strains, particularly in the genes *wbbL* and *wbbB* in KL47 strains, and *wbbO*, *wbbN* and *wbbM* in KL64 strains (Extended Data Fig. 10b and Supplementary Table 8). Evaluation of the genomic data also revealed a similar prevalence of O⁻ strains in 23.2% (48/207) and 22.3% (71/319) of the KL47 and KL64 isolates, respectively (Extended Data Fig. 10c). The frequency of O⁻ strains showed no difference between body samples (Extended Data Fig. 10d; *P* = 0.13). In addition, examination of the metagenomic reads revealed that disrupted O antigen clusters were found in 3 out of the 11 samples (>5× depth), demonstrating the presence of O-antigen disrupting mutations in vivo (Extended Data Fig. 10e).

## Discussion

This study describes a nationwide, longitudinal retrospective study on CRKP epidemiology in China. We demonstrated the dominance of two K types of KL47 and KL64 in China during 2016–2020, which together accounted for 76.3% of CRKPs. This indicates that KL47 and KL64 capsule types could be considered for future vaccine development. Furthermore, we highlighted the regionally biased distribution of K types, with KL64 predominance in eastern and central regions and KL47 in the north and northeast, as well as region-specific endemic K types including KL19 in eastern China and KL10 in northeastern China. Our findings on K-type prevalence provide a critical foundation for the development of future immunoprophylactic approaches against CRKP infections.

Based on the largest phylogeny of ST11 to our knowledge, we catalogued its global genetic diversity into 12 lineages and attributed the prevalence of KL64 and KL47 in China to the clonal expansion of lineage L1, which has been epidemic in China for decades[23]. Notably, we also found at least 14 other K types in lineage L1, which might potentially emerge in the future.

We inferred the national transmissions of the epidemic L1 lineage based on its phylogeny. The top two inter-regional healthcare providers, Shanghai and Beijing, were estimated to account for 53% of the inter-regional transmission of L1 strains. Besides, Zhejiang and Henan, part of the largest sources of outbound patients, were also the third and fourth sources of transmission. Moreover, we predicted high frequencies of transmissions from Beijing to other northern regions, and from Shanghai to central and eastern regions (Fig. 3c), which are in line with the different inter-regional healthcare-seeking patterns among patients in eastern and northern China[24]. The visiting patients may have become infected with CRKP during their stay in cities and brought infection back to the hospitals in their hometown, or vice versa. Thus, the different prevalence of K types between Shanghai and Beijing directly influenced the seroepidemiology of their neighbouring regions, resulting in the current regional distribution of K types (as in Fig. 1b). Our in silico predictions of the national transmissions of lineage L1 highlighted the needs of systematic investigations on the association between CRKPs and the inter-regional healthcare-seeking behaviours.

We attributed 98.2% of the carbapenem resistance to plasmid-mediated carbapenemase acquisition. Similar to previous reports[25,26], 89.4% of the CRKPs throughout China carried *bla*KPC-2 and remained susceptible to CZA. Meanwhile, it is noteworthy that many strains from KL15, KL51 and other rare K types carried *bla*NDM/*bla*IMP, which showed resistance to CZA. The emergence of hv-CRKP strains harbouring both carbapenemase and hypervirulence genes has been associated with increased clinical mortality and has become a global concern, particularly in China[19]. Our study revealed a high yet steadily increasing prevalence of hypervirulent strains (40.6%) among CRKPs in China. Interestingly, most of the hv-CRKPs emerged through the acquisition of virulence plasmids into existing CRKP lineages, such as KL47 and KL64,

rather than vice versa, suggesting a greater fitness cost of acquiring carbapenemases than virulence genes.

Interestingly, 12.2% and 7.4% of genotyped KL47 and KL64 CRKP isolates, respectively, showed NEKp phenotypes. These are unlikely to be in vitro artefacts during laboratory subculture, as we also identified disrupted capsular genes in the unprocessed clinical samples by metagenomic analysis. Most of the conversion involved the insertion of an IS element into the initial glycosyltransferase genes in the *cps* regions, which was unexpected as the IS elements are normally considered to be selfish DNA and not involved in the metabolic process of the host[27].

The in vivo presence of the capsule-deficient bacteria and their role during the infection has long been a mystery. In particular, NEKp showed significantly reduced transmission capacity and heterogeneity in serum resistance ability[28–30], raising the question of whether their capsule deficiency was beneficial or simply a stochastic evolutionary dead-end. Contrary to previous reports mainly in urinary tract infections[21], we found similar levels of NEKp from different sites of infection. Our results suggested that the fitness advantage of NEKp is not due to low metabolic burden. Instead, we speculated that in vivo maintenance of NEKp may involve metabolic redistribution of a key resource, Und-P, which is required for the synthesis of extracytoplasmic glycan polymers including peptidoglycan, CPS and O-antigen[31]. However, in-depth in vivo studies are needed to understand the role of the surface polysaccharide switch along the infection process.

Yet, such in vivo growth advantages have been under strong purifying selection in long-term evolution. Consistent with previous reports[32], the NEKp strains were scattered throughout the phylogenetic tree and did not form clusters. Given that the frequency of NEKp strains was stable over the study period and the importance of the capsule for *K. pneumoniae* survival in vitro[29] and dissemination upon infection[28], it is tempting to postulate that these NEKp strains have reduced transmissibility and exist as evolutionary terminal forms without the risk of massive clonal transmission.

We also demonstrated the widespread prevalence of O⁻ strains among clinical isolates and proved their existence in vivo using metagenomic analysis. Due to the lack of commercial antiserum specific for O2a or OL101, we invented a phage-based method that tests the presence of the O-antigen by using its specific targeting phages after depolymerizing the capsules using depolymerase. We found high levels (approximately 25%) of O⁻ strains in the two predominant K-type CRKPs. Rough strains induced by the insertion of IS in the *rfb* regions have only been rarely reported[33], and our results suggest that vaccines and monoclonal antibodies targeting O-antigens may have lower in vivo efficiencies than what is expected.

A limitation of this work is that the strains were predominantly collected from patients admitted to large hospitals and therefore may not be representative of the epidemiological trends among patients in smaller, community-based hospitals. In addition, the lack of clinical data from patients hampered our ability to analyse the association between NEKp strains and antimicrobial use and patient immune status.

In conclusion, we describe a 5-year multi-centre surveillance of CRKP-associated infections in China and provide a comprehensive overview of the distribution and evolution of the capsule, AMR and key virulence characteristics of the CRKP strains. We also describe a balanced evolution of CRKPs that has resulted in the maintenance of moderate but stable levels of capsular- and O-antigen-deficient strains in clinical samples.

## Methods

### Ethics statement

All of our research complies with the relevant ethical regulations. The study protocol was approved by the Institutional Review Board of Huashan Hospital, Fudan University (number 2018-408 and number 2019-460). Patients' serum, sputum, urine, bile and pus were collected

at Ruijin Hospital affiliated with Shanghai Jiaotong University. Approval was obtained from the Ethics Committee of the Ruijin Hospital, Shanghai Jiaotong University (number 2017-205). Verbal consent was obtained from the patients before their participation. All patients and data included in the study were anonymized.

All animal procedures were performed in accordance with the Regulations for the Administration of Affairs Concerning Experimental Animals and approved by the Animal Ethics Review Committee of the School of Pharmacy, Fudan University (project number 2023-03-HSYY-QXH-31) and the Animal Ethics Review Committee of Shanghai Jiaotong University (project number A-2023-009). All experimental C57BL/6J mice (specific pathogen-free grade) were female, 7 weeks old and purchased from Shanghai Jiao Tong University School of Medicine. Mice were housed in a standard laboratory environment with a 12 h light/12 h dark cycle, 21–24 °C ambient temperature and 40–60% relative humidity. Mice were given free access to the standard chow diet and drinking water. Mice were randomly allocated into different groups. The number of mice used in each experiment is indicated in the figure legends. No mice or data points were excluded from the analyses.

### Strain collection and antimicrobial susceptibility testing

A total of 1,017 CRKP strains were non-duplicate sequentially collected from *K. pneumoniae* infected patients admitted to 40 hospitals in 26 Chinese cities every October 1–31 between 2016 and 2020 by the China Antimicrobial Surveillance Network (www.chinets.com). All isolates were identified by matrix-assisted laser desorption ionization–time of flight mass spectrometry (bioMérieux, France). Minimal inhibitory concentrations (MICs) were determined using the reference broth microdilution method recommended by the Clinical and Laboratory Standards Institute (CLSI), which was used to categorize *K. pneumoniae* as susceptible (S), resistant (R) or intermediate (I). The breakpoints of *K. pneumoniae* according to CLSI M100-ED31 were interpreted for imipenem ($S \leq 1$ mg l$^{-1}$, $R \geq 4$ mg l$^{-1}$), meropenem ($S \leq 1$ mg l$^{-1}$, $R \geq 4$ mg l$^{-1}$), CZA ($S \leq 8/4$ mg l$^{-1}$, $R \geq 16/4$ mg l$^{-1}$), cefepime ($S \leq 2$ mg l$^{-1}$, $R \geq 16$ mg l$^{-1}$), ceftazidime ($S \leq 4$ mg l$^{-1}$, $R \geq 16$ mg l$^{-1}$), ceftriaxone ($S \leq 1$ mg l$^{-1}$, $R \geq 4$ mg l$^{-1}$), cefoperazone–sulbactam (according to cefoperazone $S \leq 16$ mg l$^{-1}$, $R \geq 64$ mg l$^{-1}$), piperacillin–tazobactam ($S \leq 16$ mg l$^{-1}$, $R \geq 128$ mg l$^{-1}$), cefuroxime ($S \leq 8$ mg l$^{-1}$, $R \geq 32$ mg l$^{-1}$), cefazolin ($S \leq 2$ mg l$^{-1}$, $R \geq 8$ mg l$^{-1}$), amikacin ($S \leq 16$ mg l$^{-1}$, $R \geq 64$ mg l$^{-1}$), trimethoprim–sulfamethoxazole ($S \leq 2/38$ mg l$^{-1}$, $R \geq 4/76$ mg l$^{-1}$), aztreonam ($S \leq 4$ mg l$^{-1}$, $R \geq 16$ mg l$^{-1}$), ciprofloxacin ($S \leq 0.25$ mg l$^{-1}$, $R \geq 1$ mg l$^{-1}$) and levofloxacin ($S \leq 0.5$ mg l$^{-1}$, $R \geq 2$ mg l$^{-1}$)[34]. Tigecycline MICs were interpreted using the US Food and Drug Administration MIC breakpoints for Enterobacterales ($S \leq 2$ mg l$^{-1}$, $R \geq 8$ mg l$^{-1}$) (https://www.fda.gov/drugs/development-resources/antibacterial-susceptibility-test-interpretive-criteria). European Committee on Antimicrobial Susceptibility Testing (41) (EUCAST) MIC interpretive breakpoints were used for polymyxin B ($S \leq 2$ mg l$^{-1}$, $R \geq 4$ mg l$^{-1}$). *Escherichia coli* ATCC 25922 and *E. coli* ATCC 35218 were tested as the quality control strains for antimicrobial susceptibility testing. The *K. pneumoniae* strains resistant to at least one of the carbapenem antibiotics (meropenem or imipenem) were defined as CRKP. Abbreviations and their expansions in this study are listed in Supplementary Table 11.

More than half of the patients were male (667; 65.6%). The mean age of the patients was 56.17 (±18.70) years. The most common specimen sources were sputum (458/1,017, 45.03%), blood culture (242/1,017, 23.80%) and urine (120/1,017, 11.80%). See details in Supplementary Table 1. Most isolates (41.00%) were collected from patients in medical intensive care units.

### K-typing by *wzi* genotyping

To identify the K types of *K. pneumoniae* strain, the *wzi* allele was amplified by PCR and aligned with the *wzi* sequences deposited in the database of Institut Pasteur (http://bigsdb.web.pasteur.fr)[31]. KL14, KL64,

KL15, KL51 and KL102 were further identified by PCR amplification of the variable sequence of the *wzy* gene[35–37]. Besides, the Kaptive software was also used to identify the capsule synthesis loci (K-loci or KL)[38]. The primers used for genotyping are listed in Supplementary Table 12.

### PCR screening for carbapenem resistance genes and virulence-associated genes

The presence of common carbapenem resistance genes (*bla*$_{IMP}$, *bla*$_{NDM}$, *bla*$_{VIM}$, *bla*$_{KPC}$ and *bla*$_{OXA-48-like}$) and virulence-associated genes (*rmpA, rmpA2, iroB, iucA* and *peg344*) was confirmed by PCR using the primers described in Supplementary Table 12. The positive PCR amplicons of carbapenem resistance genes were sequenced and aligned with the sequences released from GenBank (www.ncbi.nlm.nih.gov/blast/).

### Whole-genome sequencing and bioinformatics analysis

We performed whole-genome sequencing on 207 KL47 and 319 KL64 isolates to obtain a phylogenetic overview of the two most common K-type CRKPs (strains were randomly selected for sequencing, and the selection criteria are shown in Extended Data Fig. 1) and on non-carbapenemase-producing CRKP strains to analyse other mechanisms of carbapenem resistance. Genomic libraries were prepared using the Rapid Plus DNA Lib Prep Kit for Illumina (catalogue number RK20208) and sequenced using Illumina NovaSeq 6000. The obtained reads were trimmed by Trimmomatic v0.39, and strain assemblies were generated using EToKi assemble[39]. The assembled genomes were annotated using PROKKA[40]. ISs were predicted using ISfinder (http://www-is.biotoul.fr). AMR genes and virulence genes were predicted using Kleborate[41].

### Phylogenetic analysis

Three of the 526 genomic sequenced strains were not in ST11 and therefore not included in the phylogenetic analysis. The remaining 523 genomes, plus all public ST11 genomes, were aligned to a reference genome (GCF_011066505.1; sample from blood, Hong Kong, China, 2016) using the EToKi align module to obtain a multiple sequence alignment of the nonrepetitive core genomic regions shared by ≥95% of the genomes. The resulting alignments were subjected to a maximum-likelihood phylogeny using the EToKi phylo module; the recombinant regions were removed using RecHMM[42] and were visualized using Phandango[43]. Furthermore, we used EToKi MLSType to genotype all ST11 genomes based on the scgMLST_629S scheme hosted in Institut Pasteur (https://bigsdb.pasteur.fr/klebsiella/). All the obtained core genome multilocus sequence typing (cgMLST) profiles were separated into clonal groups using the LINcoding script (https://gitlab.pasteur.fr/BEBP/LINcoding) with a cutoff of 69.7933% allele similarity as described in https://bigsdb.pasteur.fr/klebsiella/cgmlst-lincodes/.

### Inferences of population dynamics for lineage L1

The origin time of the ST11 population was estimated using BactDating[44]. Parallel chains of $5 \times 10^6$ samples each were run for each of the substitution models of 'strictgamma', 'mixedgamma', and 'carc'. The first 50% of the chain ($3 \times 10^6$ samples) for each model was discarded as burn-ins, and the convergence of the run was determined by ensuring effective sampling sizes of >100 for all parameters. The results from all samples were compared based on their Bayes factors using the modelcompare function in BactDating, which indicated that the 'strictgamma' model was optimal. Finally, the dynamics of effective population sizes were estimated using the skygrowth package in R[45], and the geographic origins of each node in the tree were estimated using the maximum-likelihood algorithm implemented in TreeTime[46]. In addition, we also estimated the ancestral geographic origins of each node using the Ancestral Character Estimation function in the APE package in R with three different models of 'ER', which assumes equal transmission rates across all regions, 'SYM', which assumes different, yet symmetric

rates for transmissions between different pairs of regions, and 'ARD', which is an all-rates-different model. These models generated slightly different results, which are summarized in Extended Data Fig. 3. The tree was visualized using iTOL v6[47].

### Evaluation of temporal signal in lineage L1

The presence of a temporal signal in lineage L1 was tested using three approaches. The regression of root-to-tip distances and dates of isolation was estimated using TempEst with a correlation coefficient ($R$) of 0.18 and $P$ value of $8.2 \times 10^{-12}$. We then randomly permutated the isolation dates of the strains 10 times and estimated their $R$ values as shown in Extended Data Fig. 2b. The same datasets were also used for BactDating inferences as described above, and their substitution rates were compared with the rate from the actual data, which was also shown in Extended Data Fig. 2c.

### Identification of carbapenemase-encoding and hypervirulence-associated plasmids

To detect plasmids in the ST11 isolates, all 6,779 complete sequences of *Klebsiella* plasmids were downloaded from the National Center for Biotechnology Information (NCBI) RefSeq database (July 2022) and used as references. All 3,570 ST11 genomes were compared with the reference plasmid sequences using BLASTn; the alignment regions for each plasmid among the assemblies were summarized, and only plasmids that have >50% of their sequences aligned with >85% identities were kept. For each genome, we reordered its associated plasmid references according to the alignment scores and selected the hits using a 'Best-first' greedy algorithm that is similar to what is described in ref. 48. Briefly, for each iteration, the reference plasmid with the greatest alignment score was selected, and all contigs associated with it were taken out. This process was repeated until there was no hit left. Furthermore, to identify hypervirulence-associated plasmids, we predicted the presence of the carbapenemase or the five hypervirulence-associated genes (*iucA*, *iroB*, *rmpA*, *rmpA2* and *peg344*) in the selected contigs based on BLASTn alignments. Only the plasmids that carry at least one of these functional genes were reported. The identified plasmids were then aligned onto the NCBI RefSeq database again to obtain similar plasmids and had their incompatibility and MOB types determined using the MOB-Typer software[49].

### Capsular Quellung assay for genotyped KL47 and KL64 *K. pneumoniae* strains

The traditional serologic method, the Quellung assay, was performed to further confirm the K-type results for these KL47 (263 strains) and KL64 (513 strains) clinical strains in this study[15]. Rabbit K47-antiserum and K64-antiserum were prepared as described previously[15]. Bacteria in 1× phosphate-buffered saline (PBS) were mixed with the equal volume of rabbit K64-antiserum (titre of $1:1.5 \times 10^7$) or K47-antiserum (titre of $1: 6.1 \times 10^6$) or control rabbit serum and incubated with the methylene blue solution (Sangon Biotech, catalogue number A610622) on a glass slide for 15 min. Finally, the mixture was observed under a light microscope.

### Capsule extraction and uronic acid quantification

The capsule polysaccharide of strains was extracted and quantified as previously described[21,50]. Briefly, $1 \times 10^9$ colony forming units (CFU) bacteria were pelleted for capsule uronic acid extraction. Bacteria were mixed with 1% Zwittergent 3–14 detergent (Sangon Biotech, catalogue number A610552) in 100 mM citric acid (Sangon Biotech, catalogue number A610055). Then uronic acid was precipitated by ethanol (Sinopharm Chemical Reagent, catalogue number 10009218), incubated with 0.0125 M sodium tetraborate/sulfuric acid (tetraborate, Sigma-Aldrich, catalogue number 221732; sulfuric acid, Sinopharm Chemical Reagent, catalogue number 10021608) and 0.125% carbazole absolute ethanol (carbazole, Sangon Biotech, catalogue number

A600269), finally measured at 530 nm and correlated to a standard curve of D-glucuronic acid.

### Transmission electron microscopy

Bacteria of $5 \times 10^8$ CFU were pelleted and fixed in 2.5% glutaraldehyde (Sangon Biotech, catalogue number A600875) and 4% polyformaldehyde (Servicebio, catalogue number G1101) for 2 h at room temperature. Then the ultra-thin sections were prepared, and the image from transmission electron microscopy was obtained with the Hitachi transmission electron microscope H-9500 (Tokyo, Japan) at the Electron Microscopy Imaging Laboratory of Shanghai Jiaotong University School of Medicine (Shanghai, China).

### Estimation of the mutation rate of capsule deletion in vitro

A single colony of the EKp strain was plated on blood agar plate and incubated at 37 °C for 48 h. All bacterial colonies were then collected and resuspended in 1× PBS. The mutation rate was obtained by counting the proportion of strains that appeared translucent over the total colonies.

### Metagenomic sequencing and analysis

Clinical specimens were obtained from Ruijin Hospital Affiliated to Shanghai Jiaotong University School of Medicine. The sputum, urine, bile and pus used for metagenomic sequencing were collected from *K. pneumoniae* infected patients, 83.3% (10/12) of whom were male. The mean age of the patients was 66.00 (±11.59) years (for details, see Supplementary Table 10). Sputum and pus specimens were incubated with 4% sterilized sodium hydroxide (Sinopharm Chemical Reagent, catalogue number 10019718) for 1 h at room temperature to liquefy. Urine and bile specimens were centrifuged to obtain enriched specimens pellet. All specimens were incubated at 83 °C for 30 min for sterilization, then incubated with 5 mg ml$^{-1}$ lysozyme (Vazyme, catalogue number DE103) and 32 U ml$^{-1}$ lysostaphin (Sigma-Aldrich, catalogue number L7386) for 4 h at 37 °C and incubated with 0.1 mg ml$^{-1}$ protease K (LABLEAD Biotech, catalogue number K0510) for 2 h at 65 °C. Finally, bacterial DNA was further extracted and purified using the TIANamp Bacterial DNA Kit (TIANGEN Biotech, catalogue number DP302). Paired-end libraries with insert sizes of ~300 bp were prepared using Illumina's standard genomic DNA library preparation procedure (VAHTS Universal DNA Library Prep kit for Illumina V3) and sequenced on an Illumina NovaSeq 6000 using the S4 reagent kits (v.1.5) according to the manufacturer's instructions. Low-quality bases and adapters in the metagenomic reads were screened and removed using BBDuk (https://sourceforge.net/projects/bbmap/). The presence of *K. pneumoniae* in the samples was determined using both KRAKEN (https://ccb.jhu.edu/software/kraken/) and SPARSE (https://github.com/zheminzhou/SPARSE). Sequences of the K- and O-antigen gene clusters obtained from Kaptive were used as references to identify sequence variations. All reads in each sample were aligned to these reference sequences using minimap2 and visually examined with the help of samtools v.1.2. The long-range deletions and insertions of transposons were identified by recognizing multiple reads encompassing the conjugative sites. The frameshifts and nonsense mutations in the coding sequences were recognized by comparing all sequence variations with gene annotations.

### Deletion clones and complementary plasmid construction

The *K. pneumoniae* Δ*cps* (Δ*wcaJ*, Δ*wbaP* or Δ*wbaP*Δ*wcaA*), Δ*ops* (Δ*wbbN* or Δ*wbbL*) and Δ*cps*Δ*ops* (Δ*wcaJ*Δ*wbbN*) mutants were constructed using the λ Red-dependent recombination system and the flippase (FLP)/FLP recognition target (FRT) recombination system according to the methods reported previously[51,52]. First, the donor DNA comprising 500 bp upstream of the target gene, the hygromycin-resistant gene and 500 bp downstream of the target gene was constructed via Gibson. The hygromycin-resistant gene flanked by the FRT was PCR-amplified from pUC19-Hph plasmid. Then the thermosensitive

plasmid pKOBEG harbouring the apramycin-resistant gene was introduced into the competent cells derived from the wild-type (WT) strain using electroporation technology. The pKOBEG carried the λ phage red operon controlled by a promoter under the induction of arabinose. The donor DNA was then electroporated into WT pKOBEG. Ultimately, the *cps* or *ops* gene in the WT strain was replaced by the FRT-flanked hygromycin-resistant gene. The mutants were selected on LB agar plates containing apramycin (Sangon Biotech, catalogue number A600090) and hygromycin (Rhawn, catalogue number R053999) and confirmed by PCR and Sanger sequencing. The 16−312Δ*cps*Δ*ops* double knockout mutant was conducted by first replacing *cps* (*wcaJ*) using the hygromycin-resistant gene, then removing the hygromycin resistance gene by FLP /FRT recombination method and finally further deleting the *ops* (*wbbN*) gene by the hygromycin-resistant gene replacement. Plasmid pKOBEG was removed by growth at 42 °C with shaking.

For complementary plasmid construction, the *cps* or *ops* genes of KL64 or KL47 strains were amplified from *K. pneumoniae* strains 16−312 and 19−249, respectively. They were then cloned into the pACYC184 or pBAD33 vector, and the recombinant plasmids were transformed into the *cps* or *ops* mutant strains. In addition, the 16−312Δ*cps*::*cps* (16−312Δ*wcaJ*::*wcaJ*) strain was generated by the λ-red-mediated in situ complementation method. Briefly, donor DNA containing 500 bp upstream of the *cps* gene, the *cps* gene, the hygromycin resistance gene and 500 bp downstream of *cps* gene was electroporated into 16−312Δ*cps* strain harbouring pKOBEG vector. Mutants were then selected on LB agar plates containing apramycin and hygromycin. Expression of *cps* or *ops* genes was confirmed by the Quellung reaction assay or silver staining, respectively. Primers used for deletion clone construction and complementation and details of the engineered mutants are listed in Supplementary Tables 12 and 13 The gene in *cps*-locus or *rfb*-locus and their putative gene products are listed in Supplementary Table 14.

## Macrophage phagocytosis assay

The phagocytosis assay was performed using the mouse macrophage cell line RAW 264.7 (ATCC catalogue number TIB-71) and peritoneal macrophages as previously described[53]. Peritoneal macrophages were collected from 7-week-old female C57BL/6J mice on the 5th day after intraperitoneal injection of 1 ml 5% thioglycollate broth (BD Difco, catalogue number 211716)[54]. Briefly, $5 \times 10^5$ cells were infected with bacteria at a multiplicity of infection of 50 for 1 h. The cells were then washed three times with 1× PBS and treated with 500 μl Dulbecco's modified eagle medium (Servicebio, catalogue number G4515) containing 300 μg ml$^{-1}$ gentamycin (Sangon Biotech, catalogue number A506614) or amikacin (Solarbio, catalogue number A9660) for 1 h to eliminate the extracellular bacteria. After washing three times with 1× PBS, the cells were treated with 1 ml of 1% TritonX-100 (Sangon Biotech, catalogue number A110694) for 10 min to release the intracellular bacteria. Then intracellular bacteria were counted by serial dilution agar plating method.

## In vitro bacterial competition assay

Each WT strain ($1 \times 10^5$ ml$^{-1}$) was mixed 1:1 with the isogenic Δ*cps* mutant in LB medium and incubated at 37 °C for 24 h. Bacteria were then serially diluted and plated on LB and LB containing hygromycin (Rhawn, catalogue number R053999) plates. The ratio of Δ*cps* (the number of colonies on hygromycin LB plates) to WT (the number of colonies on LB plates minus the number of colonies on hygromycin LB plates) was calculated to obtain the competitive index value.

## In vivo bacterial competition assay

Groups of five 7-week-old female C57BL/6J mice were infected intraperitoneally with an assigned dose of WT or Δ*cps* mutant to evaluate the systemic infectivity of the bacteria. Survival curves were monitored for 10 days. Differences in mouse survival were determined by

Kaplan−Meier survival analysis with log-rank test. The in vivo bacterial competition assay was performed in 7-week-old female C57BL/6J mice (*n* = 5). Each WT strain was mixed with the isogenic Δ*cps* mutant at a ratio of 1:1 in 200 μl 1× PBS and injected intraperitoneally into mice. After 24 h post infection, mice were euthanized, and the burdens of WT and Δ*cps* mutant in lung, liver and spleen were counted by serial dilution on LB agar plates and LB agar plates containing hygromycin (Rhawn, catalogue number R053999), respectively. The ratio of mutant:WT was calculated to obtain the competitive index value.

## Biofilm formation assay

The ability of *K. pneumoniae* strains to form biofilms was determined as described previously[21,55]. Briefly, bacteria were incubated in tryptic soy broth medium (Hopebio, catalogue number HB4150) to optical density (OD) = 0.4. The bacterial suspension was incubated in a 96-well microtitre plate at 37 °C for 48 h, then stained with 0.4% crystal violet (Macklin, catalogue number C805211) at 37 °C for 45 min and destained with 33% acetic acid (Sinopharm Chemical Reagent, catalogue number 10000208) for 30 min. Finally, the sample was read at a wavelength of 595 nm.

## Serum resistance assay

The serum resistance ability of the strains was estimated using the serum resistance assay. About $5 \times 10^4$ CFU bacteria were incubated with 75% baby rabbit complement at 37 °C for 1, 2 and 3 h. Bacterial counts were determined to enumerate the survival rate. The survival rate was calculated using the following formula: survival rate 1 (SR1) = (CFU surviving at 1 h post-complement treatment/CFU surviving in initial) × 100; SR2 = (CFU surviving at 2 h post-complement treatment/CFU surviving in initial) × 100; SR3 = (CFU surviving at 3 h post-complement treatment/CFU surviving in initial) × 100. Each isolate was classified as previously described[56]. Briefly, serum resistance level 1 is SR1 and SR2 < 10%, and SR3 < 0.1%; level 2 is SR1 and SR2 between 10% and 100%, and SR3 < 10%; level 3 is SR1 > 100%, SR2 and SR3 < 100%; level 4 is SR1 and SR2 > 100%, and SR3 < 100%; level 5 is SR1, SR2 and SR3 > 100%, and SR2 > SR3; level 6 is SR1, SR2 and SR3 > 100%, and SR1 < SR2 < SR3. Isolates belonging to levels 1 and 2, levels 3 and 4, and levels 5 and 6 were defined as 'sensitive', 'intermediate' and 'resistant', respectively.

To compare WT/Δ*cps* strains with depolymerase-treated strains, bacteria were incubated with their corresponding K-type depolymerase (KL47 strains with Dep 42[57] and KL64 strains with Dep 41[15], at a final concentration of 100 μg ml$^{-1}$), or K-type-irrelevant depolymerase (Dpo7[17], which were specific for KL19 CPS) for 1 h before incubation with 75% baby rabbit serum. The percentage of survival was determined as the number of surviving bacteria relative to the initial bacterial addition.

Furthermore, fosmidomycin and *ispU* overexpression were performed to evaluate the effect of Und-P on serum resistance. To inhibit the synthesis of Und-P, 16-312Δ*cps* mutants were cultured overnight in LB medium with or without 500 ng ml$^{-1}$ fosmidomycin (Rhawn, catalogue number R047510) and subcultured to OD = 0.7. To increase the synthesis of Und-P, a recombinant pBAD33-*ispU* plasmid with an arabinose-induced promoter was constructed and transformed into Δ*cps* mutants. Bacteria were grown overnight in LB medium containing 10 mM arabinose and 50 μg ml$^{-1}$ apramycin and subcultured to OD = 0.7. Finally, bacteria were incubated with 75% baby rabbit complement at 37 °C for 1 h. Bacterial viability was determined by the serial dilution agar plating method.

## LPS isolation and electrophoresis

A total of 10$^{10}$ CFU bacteria were pelleted for LPS extraction. Capsular strains of the depolymerase treatment group were treated with the corresponding K-type depolymerase or K-type-irrelevant depolymerase at 37 °C for 2 h. Then, 100 μg ml$^{-1}$ deoxyribonuclease (Roche, catalogue number 4716728001), 50 μg ml$^{-1}$ ribonuclease (TIANGEN Biotech, catalogue number RT405) and 200 μg ml$^{-1}$ proteinase K (LABLEAD Biotech,

catalogue number K0510) were used to eliminate nucleic acids and protein. LPS was extracted by the hot phenol/water method[58]. Finally, samples were run on 15% sodium dodecyl sulfate–polyacrylamide gels and visualized by silver staining.

### RNA sequencing
RNA sequencing was performed as previously described with slight modification[59,60]. *K. pneumoniae* strains 16–312 and 16–312Δ*cps* mutant were grown in LB to an $OD_{600}$ of 0.7. Bacteria were pelleted by centrifugation, snap frozen in liquid nitrogen and cryopreserved at −80 °C. Total RNA from bacteria was extracted and purified using the RNAprep Pure Cell/Bacteria Kit (TIANGEN Biotech, catalogue number DP430). RNA integrity was assessed using the Agilent 2100 Bioanalyzer. RNA sequencing was performed at the Shanghai Personal Biotechnology (Shanghai, China). $P < 0.05$ was defined as significant difference. The LPS and peptidoglycan synthetic genes and their putative gene products are listed in Supplementary Table 14.

### Quantitative RT-PCR
Purified RNA was reverse transcribed to complementary DNA using the Thermo Scientific RevertAid RT Kit (Thermo Scientific, catalogue number M1631). Transcript levels of *wbbM*, *LpxD*, *LpxM*, *LpxA*, *mraY* and *murA* were detected by quantitative RT-PCR. The relative abundance of LPS- and peptidoglycan-synthesizing genes was calculated by the $2^{-\Delta\Delta CT}$ method using the 16S ribosomal RNA gene as a reference. Primers for quantitative RT-PCR are listed in Supplementary Table 12.

### Serum bactericidal activity
The SBA assay was determined using a method previously reported with slight modification[61,62]. WT strain and isogenic Δ*cps* mutant were incubated with 10% heat-inactivated serum from patients (infected with KL47 or KL64 *K. pneumoniae* strains) in the presence of 15% baby rabbit complement or 15% baby rabbit complement alone for 1 h. The percentage of survival was calculated by dividing the viable counts of patient serum plus complement by complement alone. The serum of infected patients was obtained from Ruijin Hospital Affiliated to Shanghai Jiaotong University School of Medicine; 62.5% (5/8) of patients were male. The mean age of the patients was 51.13 (±13.78) years.

### Statistical analyses
Unpaired two-sided Student's *t*-test was used to compare the means between two groups. The Welch's correction was used when variation was unequal. Pearson's chi-square was used to compare groups for proportion data. The Cochran–Mantel–Haenszel test was used to compare the mean rank difference of two groups in serum resistance assay. The Cochran–Armitage test for trend was used to analyse the temporal trend in the proportions of the K types, hypervirulent strains and NEKp strains from 2016 to 2020. Differences in mouse survival were determined by Kaplan–Meier survival analysis with log-rank test. Two-sided one-sample *t*-test was used to compare the mean of $log_{10}$-transformed competitive index or competitive index to a hypothetical value of 0 or 1 in competition assay. No comparison between groups was involved in the competition assay. $P < 0.05$ was considered statistically significant.

### Reporting summary
Further information on research design is available in the Nature Portfolio Reporting Summary linked to this article.

### Data availability
The raw reads of 542 *K. pneumoniae* sequenced in this study were uploaded to the NCBI database under the project PRJNA1028672. A total of 3,047 publicly available genomes were enrolled including 2,695 public ST11 *K. pneumoniae* genomes which were available in the NCBI database under various projects (Supplementary Table 6), and 352 ST11 *K. pneumoniae* genomes were downloaded from Genome Sequence Archive[63] in Chinese National Genomics Data Center[64] under project PRJCA003173 ($n = 300$) and PRJCA012323 ($n = 52$). The metagenomic sequences of clinical samples acquired during the capsule and O-antigen trial parts were uploaded to the NCBI database under the project PRJNA1028672. The transcriptome sequencing data were uploaded to the NCBI database under the project PRJNA1028672 as well. Reference genome for phylogenetic analysis of ST11 strains were set as GCF_011066505.1. The original images and data are provided in the Source data. Source data are provided with this paper.

### Code availability
The SRA Toolkit (2.8.0) was used to download genomes and short reads from the NCBI. The following software were used in the analysis: Trimmomatic v.0.39, EToKi v.1.3, PROKKA v.1.14, ISfinder v.1, Kleborate v.2.3.2, Kaptive v.2.0.7, RecHMM v.1, Phandango v.1.3.0, BactDating v.1.1, TreeTime v.0.10.1, iTOL v.6, TempEst v.1.5.3, BLASTn v.2.9.0+, MOB-Typer v.3.1.0, BRIG v.0.95, BBDuk v.1, KRAKEN v.2.1.2, minimap2 v.2.26-r1175, samtools v.1.2, SPARSE v.1, R skygrowth v.0.3.1, R APE v.5.7.1, LINcoding script v.1 and search plasmic.git v.1 (https://eithub.com/xiaoliu8/searchplasmic.git).

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

## Acknowledgements

We gratefully acknowledge the contribution of the members of the CHINET group for collection of the isolates in this study. We thank the Electron Microscopy Imaging Laboratory in Shanghai Jiaotong University School of Medicine for helping with transmission electron microscopy imaging. We thank Y. Song at Shanghai Jiaotong University for helping with statistical analyses. We thank Y. Wang at China Agricultural University for sharing the plasmid pACYC184. We thank H. Ou at Shanghai Jiaotong University for sharing the plasmid pBAD33. This work was funded by the National Natural Science Foundation of China (grant number 81971896 (P.H.), 82172311 (F.H.), 32141002 (F.H.), 32170003 (Z.Z.) and 32370099 (Z.Z.)), the National Key Research and Development Project of China (2022YFA1304300) (P.H.), the China Antimicrobial Surveillance Network (Independent Medical Grants from Pfizer, 2018QD100) (F.H.) and Shanghai Antimicrobial Surveillance Network (3030231003) (F.H.).

## Author contributions

F.H., P.H. and Z.Z. conceived the study and wrote the manuscript. Y.P., R.H. and R.M. performed the experiments. H. Li and X.L. analysed the bioinformatics data. Y.W., H. Lun, X.Q., A.W., M.Z. and B.L. helped with experiments. J.L. helped with writing the manuscript. P.H. and Z.Z. supervised the project.

## Competing interests

The authors declare no competing interests.

## Additional information

**Extended data** is available for this paper at https://doi.org/10.1038/s41564-024-01612-1.

**Correspondence and requests for materials** should be addressed to Zhemin Zhou or Ping He.

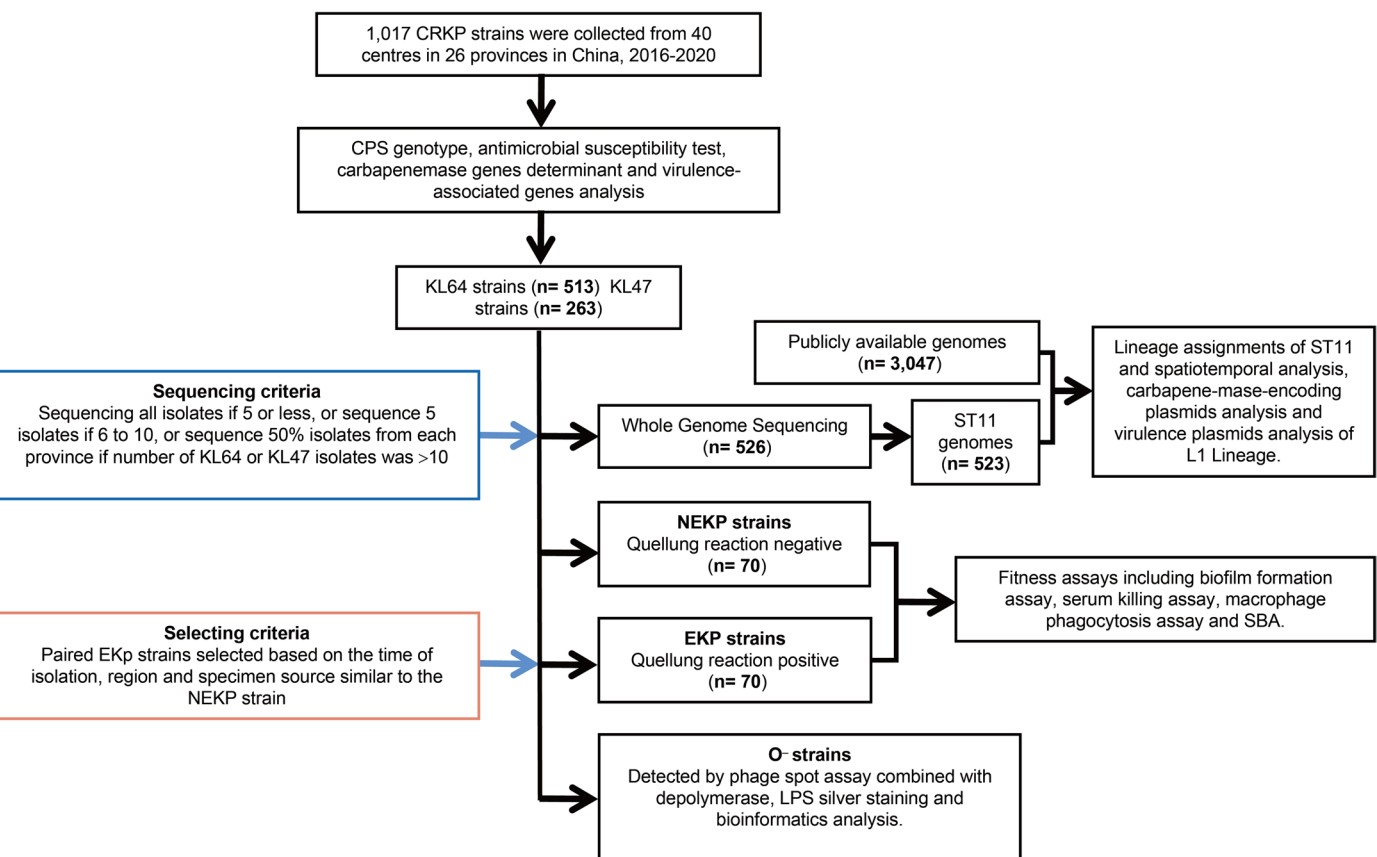

**Extended Data Fig. 1 | Flowchart of data collection, and genotypic and phenotypic characterization of the CRKP isolates.**

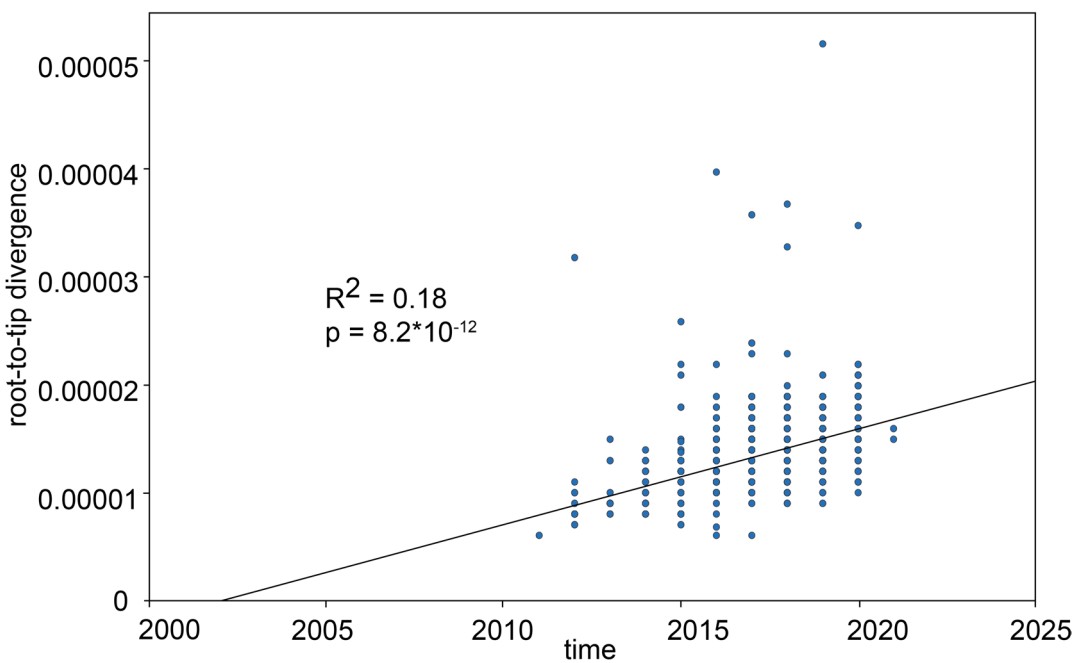

A. Regression of root-to-tip distances versus isolation dates with TempEst

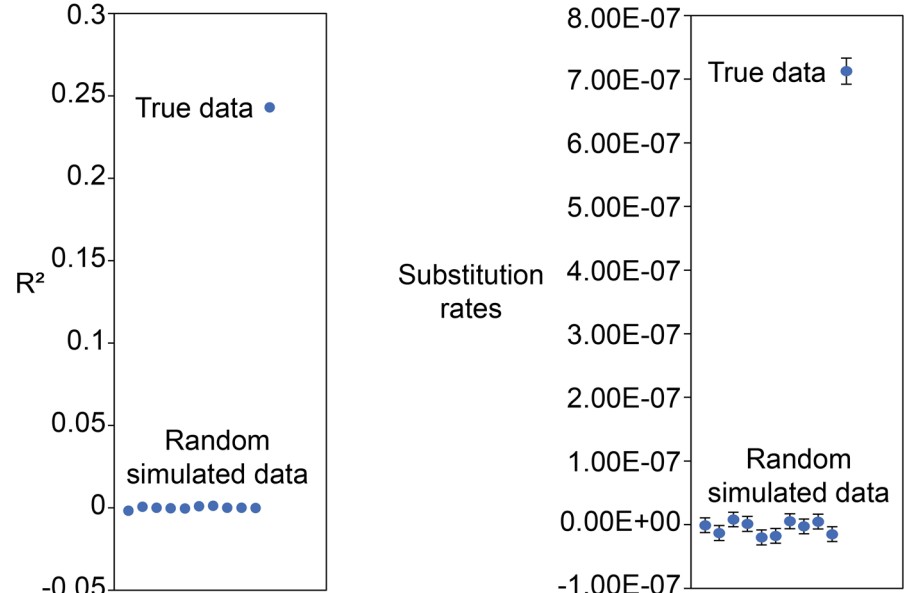

B. Date-randomisation test with TempEst   C. Date-randomisation test with BactDating

**Extended Data Fig. 2 | The root-to-tip distances and date randomisation tests for Lineage L1. implemented in the ACE function in the APE package.** **a**: A scatter plot generated by TempEst that shows linear regression of root-to-tip distances against isolation dates of the strains. The coefficient of determination (R2) and P-value is shown as well. The residual mean square was calculated using two-sided test. **b**, **c**: The date-randomisation tests based on the 'True data' and

10 datasets with the isolation dates of 1,756 samples randomly permutated. Data are presented as mean values ± SEM in part C. Significant greater R2 values of the root-to-tip tests were obtained from the true data than all the randomized data (B). And significantly greater substitution rates were also obtained based on the BactDating inferences (C).

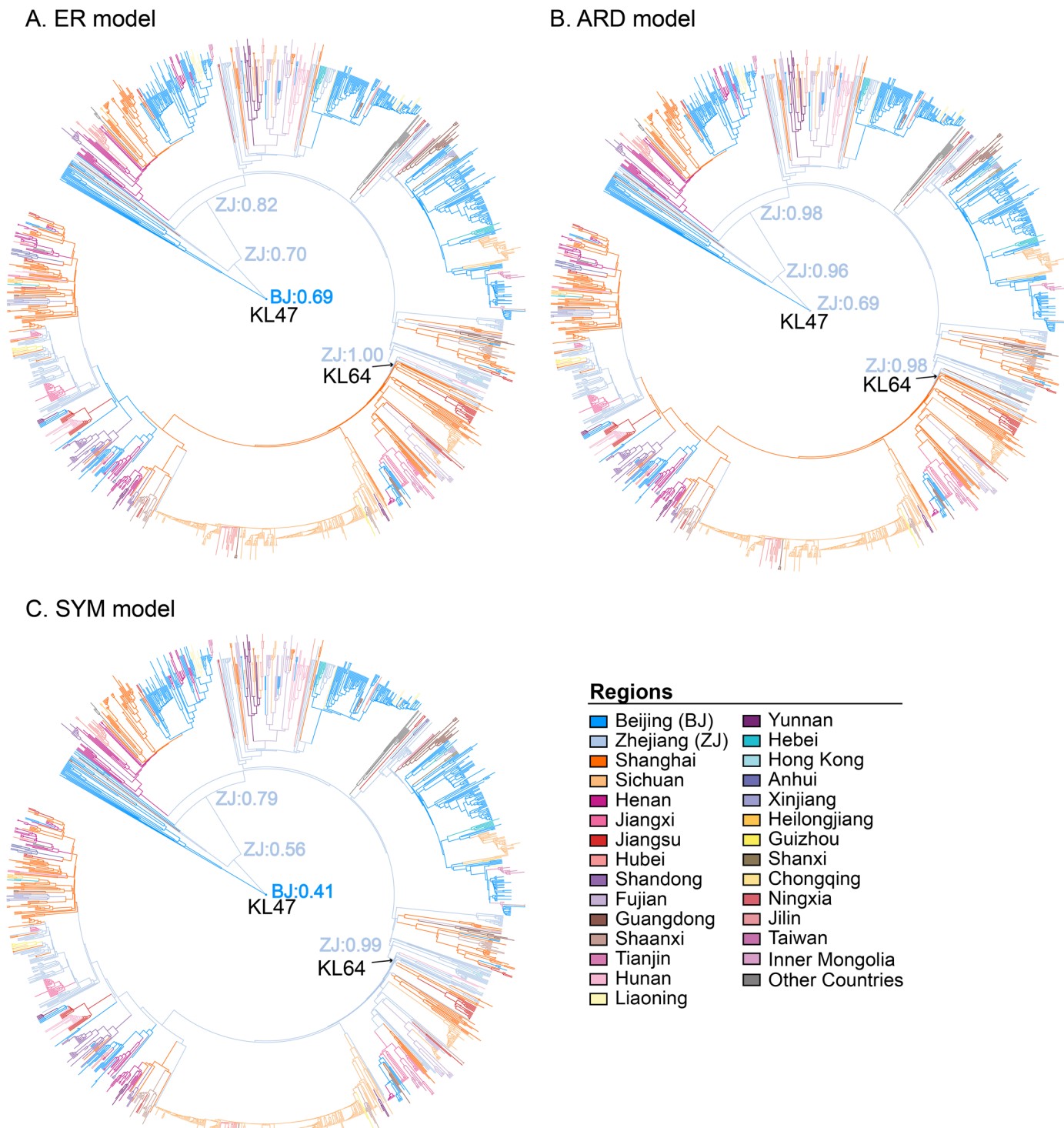

**A. ER model**

**B. ARD model**

**C. SYM model**

**Regions**

- Beijing (BJ)
- Zhejiang (ZJ)
- Shanghai
- Sichuan
- Henan
- Jiangxi
- Jiangsu
- Hubei
- Shandong
- Fujian
- Guangdong
- Shaanxi
- Tianjin
- Hunan
- Liaoning
- Yunnan
- Hebei
- Hong Kong
- Anhui
- Xinjiang
- Heilongjiang
- Guizhou
- Shanxi
- Chongqing
- Ningxia
- Jilin
- Taiwan
- Inner Mongolia
- Other Countries

**Extended Data Fig. 3 | Ancestral state reconstruction of L1 Lineage using the maximum likelihood (ML) algorithm implemented in the ACE function in the APE package.** The inferences were run based on three different models of (**a**) ER, which assumes equal transmission rates across all regions, (**b**) SYM, which assumes different, yet symmetric rates for transmissions between different pairs of regions, and (**c**) ARD, which is an all-rates-different model. Branches were colour-coded according to their most likely regional origins in the inferences (see keys). And the estimated possibility of the top region in certain major branches were shown nearby.

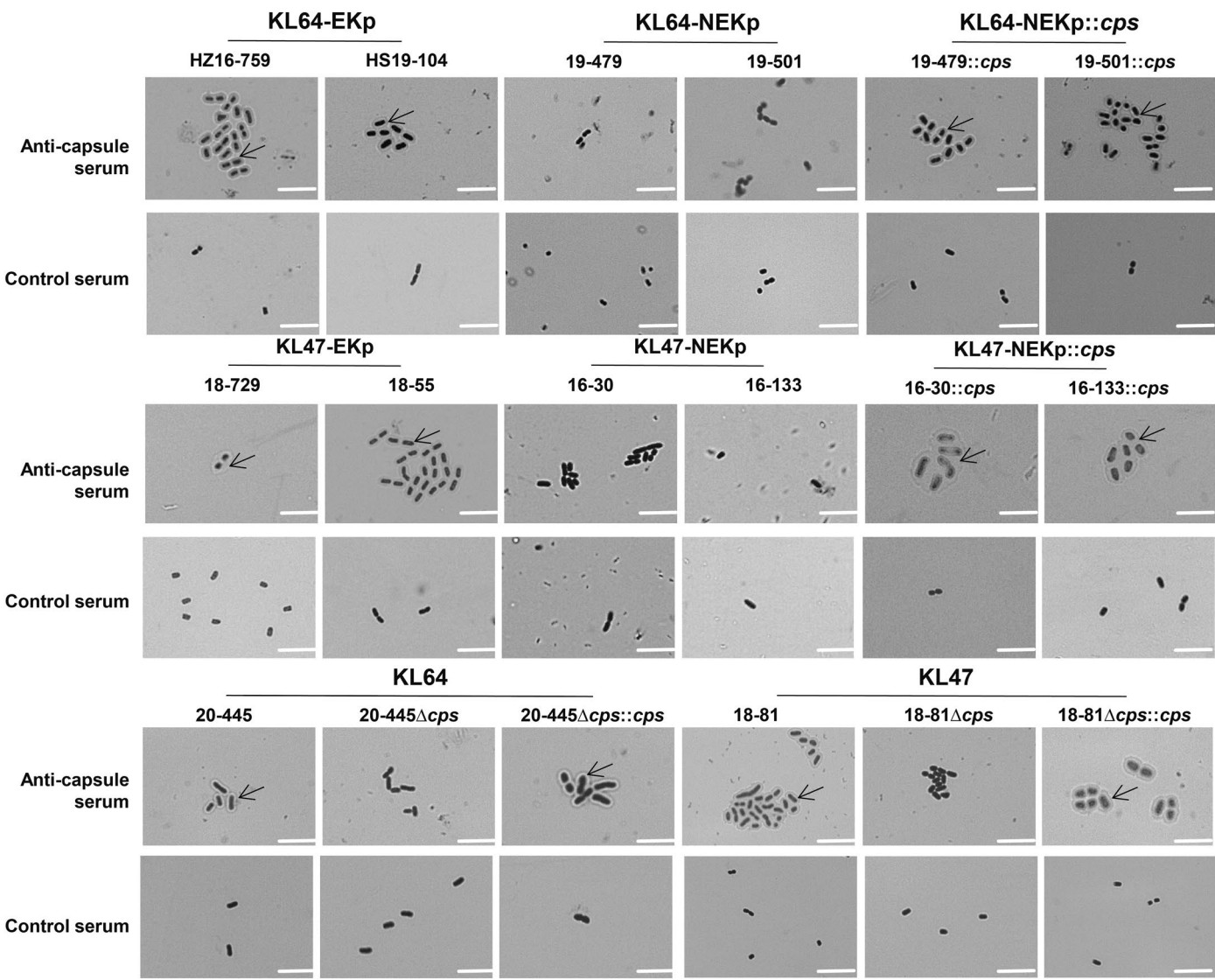

**Extended Data Fig. 4 | Capsule deficiency in KL47 and KL64 genotyped clinical isolates.** Quellung reaction of *K. pneumoniae* strains. *K. pneumoniae* strains were incubated with anti-K64 serum (or anti-K47 serum) or control serum as described in Methods. Compared with the control, the positive results showed capsular swelling under light microscopy with an oil immersion lens (magnification, 100). WT strains (KL64: 20–445, KL47: 18–81) and isogenic *Δcps* mutants showed Quellung reaction positive and negative, respectively. Quellung reaction of 20–445*Δcps* and 18–81*Δcps* was restored by complementation. Besides, Quellung reaction was restored by complementation with *wcaJ* or *wbaP* in a set of corresponding gene mutant clinical NEKp strains (KL64: 19–479, 19–501; KL47:16–30, 16–133). Each graph shows the representative result of three repetitions. The arrow points to the swollen capsule. Scale bar, 5 μm.

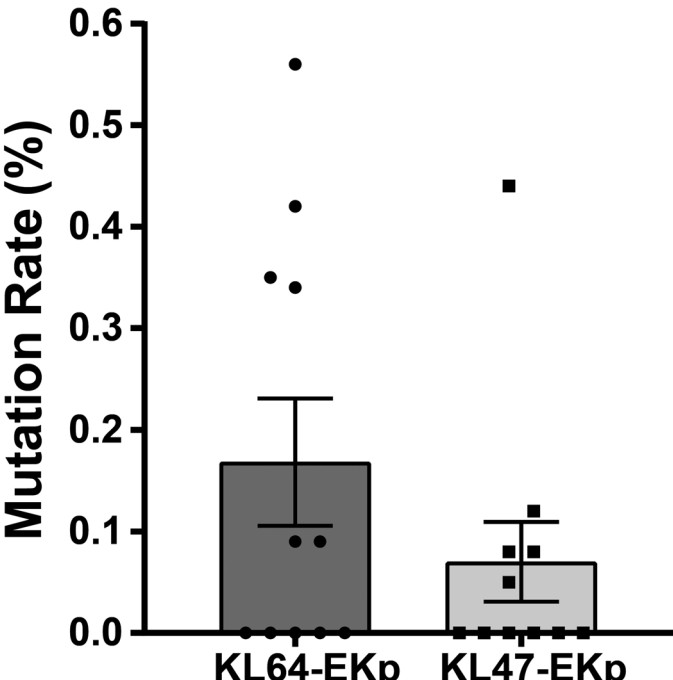

**Extended Data Fig. 5 | The mutation of capsule deletion rate *in vitro* culture.** The mutation of capsule deletion rate of KL64 and KL47 EKP isolates after 48 hours of culture in blood plate. Data are shown as the mean ± SEM. Each point represents the mean value of three biological replicates of one strain. For clarity, individual error bars for each strain are not shown.

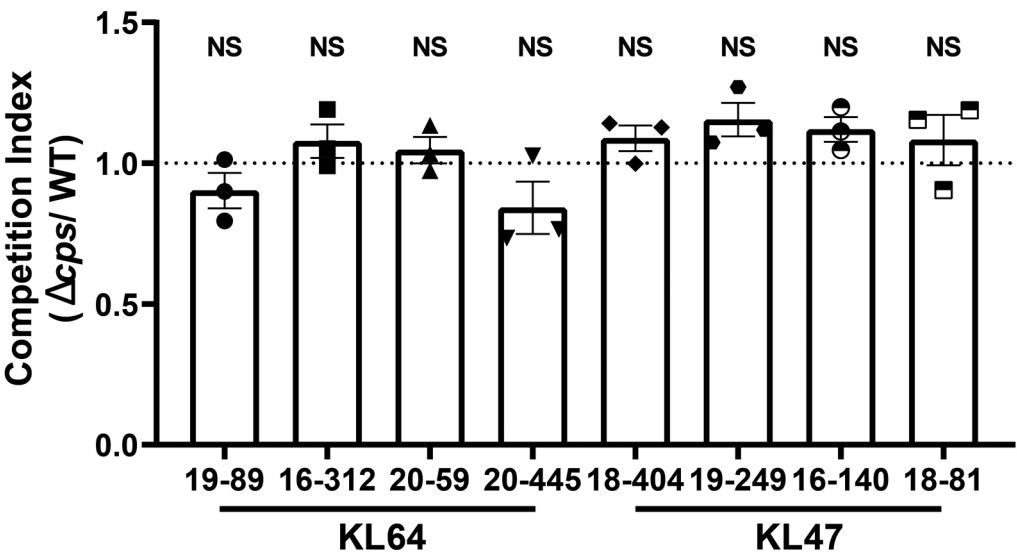

**Extended Data Fig. 6 | Capsule of KL47 and KL64 *K. pneumoniae* does not impose a fitness cost *in vitro*.** *In vitro* competition assays were performed between KL47 strains (16–140, 18–404 and 19–249) and their isogenic *Δcps* mutants, and between KL64 strains (16–312 and 19–89) and their isogenic *Δcps* mutants. Bacteria were inoculated into LB broth at a ratio of 1:1 for 24 h at 37 °C, and WT and *Δcps* mutants were quantified by differential plating to calculate the competitive index (CI) value. Two-sided one-sample t-test was performed to compare the mean of CI to a hypothetical value of one (asterisk) for each group. Data represents the mean ( ± SEM) of three biological replicates. NS, no significance.

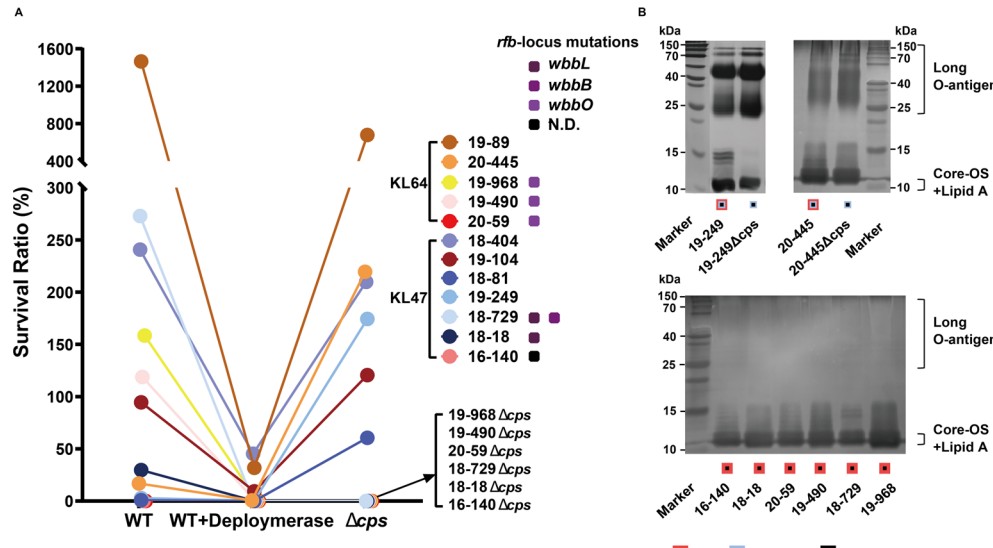

**Extended Data Fig. 7 | Effect of capsule deficiency on serum resistance and O antigen expression. a**: Serum killing assays were performed by incubating the parental EKp strains (twelve strains with different levels of serum resistance), KL64 capsular polysaccharide depolymerase (Dep41) or KL47 depolymerase (Dep42) pretreated parental EKp strains, the engineered isogenic *Δcps* strains with 75% baby rabbit serum for 1 hour at 37 °C. Percent survival was determined as the number of surviving bacteria relative to the initial bacterial addition. **b**: LPS was prepared by hot phenol-water method and analyzed by silver staining. The results of the serum killing assay showed that all capsule-stripped parental strains exhibited decreased serum resistance compared to that of untreated parental strains (except for those serum-sensitive strains that barely survived 1 hour post serum treatment) (A), demonstrating that *K. pneumoniae* capsules

do play an important role in serum resistance. Six out of twelve isogenic *Δcps* mutants exhibited increased resistance to serum compared to the capsule-stripped parental strains (A). Increased amounts of OPS were observed in the isogenic *Δcps* mutants compared to the parental EKp strains (B). The remaining six isogenic *Δcps* mutants which showed no change in resistance to serum compared to the the depolymerase-treated parental strains, were all serum-sensitive strains with disruption of O antigen synthesis, as confirmed by LPS silver staining (B) and genome sequence analysis (A). N.D., *rfb*-locus mutation gene not detected in O⁻ strain 16–140. Each point represents the mean value of three biological replicates. For clarity, individual error bars for each strain are not shown.

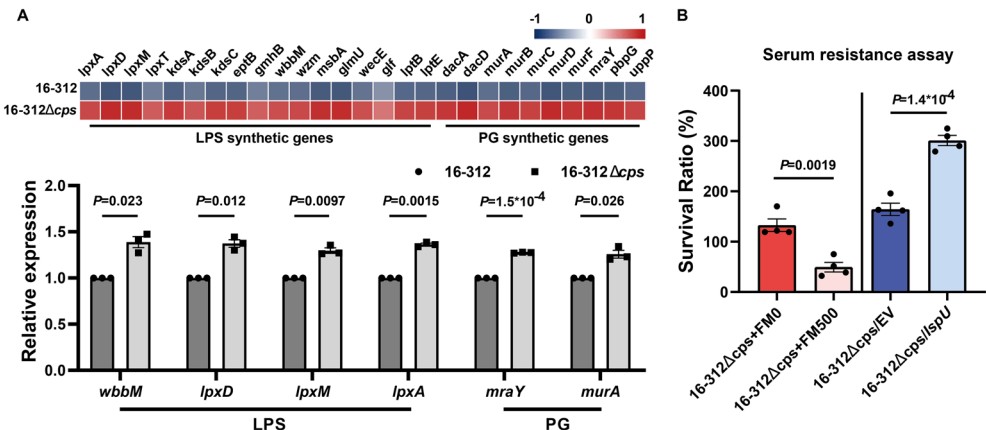

**Extended Data Fig. 8 | Investigation of the mechanism underlying the increased antiserum capacity of NEKp strains. a**: RNA sequencing and qRT-PCR of WT and *Δcps* strains. RNA sequencing heat map showing the differential expression of LPS and PG synthesis genes between EKp strain 16–312 and 16–312 *Δcps* mutant. qPCR assay was used to determine the relative mRNA levels of LPS and PG synthesis-related genes in 16–312 and 16–312 *Δcps* using the $2^{-\Delta\Delta CT}$ method. PG, peptidoglycan. Data represents the mean (± SEM) of three biological replicates of one strain. Unpaired two-sided Welch's t-test was performed for statistical analysis. **b**: Effect of Und-P on serum resistance using either fosmidomycin to inhibit the Und-P *de novo* biosynthesis or *ispU* overexpression to increase the Und-P biosynthesis. Strains 16–312*Δcps* were grown in LB medium with or without FM (500 ng/ml); strains 16–312*Δcps*/*ispU* and 16–312*Δcps*/ EV were grown in LB medium with 10 mM arabinose. The serum killing assay was performed by incubating the strains with 75% baby rabbit complement for 1 hour at 37 °C. Percent survival was determined as the number of surviving bacteria relative to the initial bacterial addition. FM, fosmidomycin; EV, empty vector. Data represents the mean (± SEM) of four biological replicates of one strain. Unpaired two-sided Student's t-test was performed for statistical analysis. Core-OS, core-oligosaccharide.

**A**

**19-971**

SH-KpOL101          SH-KpO2a

**19-971Δ*ops***

SH-KpOL101          SH-KpO2a

**19-253**

SH-KpOL101          SH-KpO2a

**19-253Δ*ops***

SH-KpOL101          SH-KpO2a

**B**

**16-312**

Dep41      SH-KpO2a      Dep41+SH-KpO2a

**16-312Δ*ops***

Dep41      SH-KpO2a      Dep41+SH-KpO2a

**19-249**

Dep42      SH-KpOL101   Dep42+SH-KpOL101

**19-249Δ*ops***

Dep42      SH-KpOL101   Dep42+SH-KpOL101

**C**

Marker | 19-971 | 19-971Δ*ops*
Marker | 19-253 | 19-253Δ*ops*

Long O-antigen
Core-OS +Lipid A

16-312 | 16-312Δ*ops* | Marker
Marker | 19-249 | 19-249Δ*ops*

Long O-antigen
Core-OS +Lipid A

□ CPS
□ O-antigen
■ Core-OS+Lipid A

**D**

**KL64 O2a strains**

17-343   18-277   20-522   19-131   18-29   18-348   20-692   Marker

Long O-antigen
Core-OS+Lipid A

Phage spot assay

Dep41+SH-KpO2a          SH-KpO2a

**KL47 OL101 strains**

18-81   18-404   16-444   18-306   17-46   18-146   17-39   Marker

Long O-antigen
Core-OS +Lipid A

Phage spot assay

Dep42+SH-KpOL101          SH-KpOL101

**E**

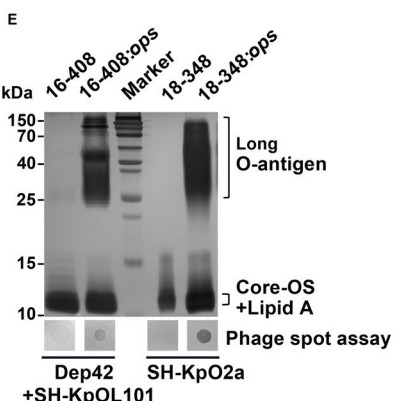

16-408   16-408:*ops*   Marker   18-348   18-348:*ops*

Long O-antigen
Core-OS +Lipid A

Phage spot assay

Dep42 +SH-KpOL101          SH-KpO2a

**Extended Data Fig. 9 | See next page for caption.**

**Extended Data Fig. 9 | Identification of smooth and rough LPS in KL64 O2a and KL47 OL101 strains. a**: Phage SH-KpO2a (specific for O serotype O2a strains) and phage SH-KpOL101(specific for O serotype OL101 strains) can specifically infect and lyse corresponding O serotype non-capsulated *K. pneumoniae* strains with smooth LPS but not strains with rough LPS. Phage SH-KpO2a and SH-KPOL101 were isolated from sewage water using the host strain of *K. pneumoniae* 19–971 (KL64/O2a NEKp strain) and 19–253 (KL47/OL101 NEKp strain), respectively. Dep41[15] and Dep42[23] are specific depolymerases against *K. pneumoniae* KL64 and KL47 capsule, respectively. Spot assay results showed that phage SH-KpO2a and SH-KpOL101 formed the clear spot on the lawn of NEKp strain 19–971 and strain 19–253, respectively, while showed no spot on the lawn of their isogenic *Δops* mutants. LPS of the corresponding strains was prepared by hot phenol-water method and analyzed by silver staining (C). **b**: EKp strains of serotype O2a (16–312) and OL101 (19–249) did not form lytic spots when infected with phage SH-KpO2a or SH-KpOL101 alone, whereas EKp strains pretreated with KL64 or KL47 depolymerase were susceptible to phage SH-KpO2a or SH-KpOL101, and formed clear spots (caused by phages) with haloes (caused by depolymerase) around. EKp isogenic *Δops* mutants pretreated with depolymerase were resistant to phage SH-KpO2a or SH-KpOL101, and formed translucent circles without clear spots. LPS of the corresponding strains was prepared by hot phenol-water method and analyzed by silver staining (C). **d**: LPS silver staining and phage spot assay of clinically isolated serotype KL64 O2a strains and serotype KL47 OL101 EKp strains. **e**: LPS silver stain and phage spot assay of clinical O⁻ strains and complementary strains with *ops*. Each graph shows the representative result of three repetitions(A-E). Core-OS, core-oligosaccharide.

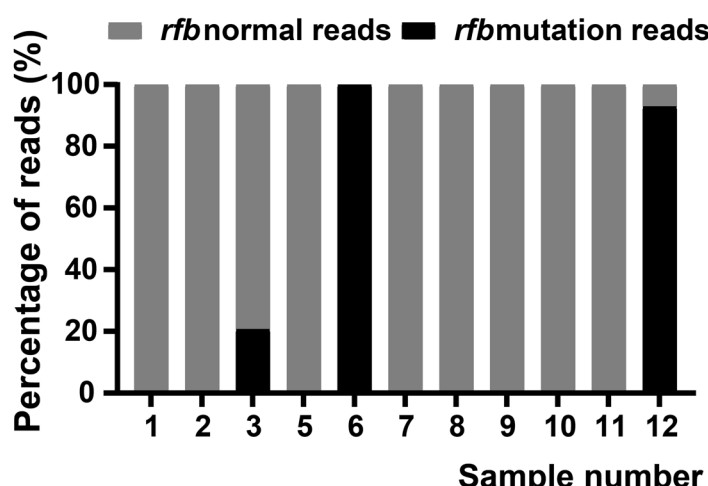

**A**

**B**

**C**

**D**

**E**

Extended Data Fig. 10 | See next page for caption.

**Extended Data Fig. 10 | O antigen deficient strains in KL64 and KL47 strains.**
**a**: O antigen of seventy NEKp strains and seventy selected paired EKp strains were detected by phage spot assay. KL64 and KL47 strains were infected with phage SH-KpO2a (with depolymerase Dep41) and SH-KpOL101 (with depolymerase Dep42), respectively. Strains that did not develop plaques were identified as O⁻ strains. Some strains were further confirmed by LPS silver staining (see Supplementary Table 8 for details). *n*, indicates the number of CRKp strains. **b**: variants in rough LPS strains. Mutations found in the O2a and OL101 antigen biosynthesis genes in *K. pneumoniae* isolates (see Supplementary Table 8 for details). The black flag indicates the location of the mutation. Red, black and blue lines indicate ISs, SNPs and deletions, respectively. *n*, indicates the total number of mutations observed in that gene. **c**: The proportion of O⁻ strains was estimated in 523 sequenced KL64 and KL47 strains. Strains with mutations in the *rfb*-locus were identified as O⁻ strains. *n*, indicates the number of CRKp strains. **d**: Frequency of O⁻ strains (526 KL64 and KL47 strains assessed by whole genome sequencing) isolated from different sites of infection. The number of strains in different infection sites is shown below. Pearson's Chi-square test was used to analyze the proportion difference of O⁻ strains strains in different infection samples. NS, no significance. **e**: Metagenomic analysis of *rfb*-locus variants in unprocessed clinical specimens (see Supplementary Table 10 for details).

# Reporting Summary

## Statistics

For all statistical analyses, confirm that the following items are present in the figure legend, table legend, main text, or Methods section.

| n/a | Confirmed | |
|---|---|---|
| ☐ | ☒ | The exact sample size (*n*) for each experimental group/condition, given as a discrete number and unit of measurement |
| ☐ | ☒ | A statement on whether measurements were taken from distinct samples or whether the same sample was measured repeatedly |
| ☐ | ☒ | The statistical test(s) used AND whether they are one- or two-sided<br>*Only common tests should be described solely by name; describe more complex techniques in the Methods section.* |
| ☒ | ☐ | A description of all covariates tested |
| ☒ | ☐ | A description of any assumptions or corrections, such as tests of normality and adjustment for multiple comparisons |
| ☐ | ☒ | A full description of the statistical parameters including central tendency (e.g. means) or other basic estimates (e.g. regression coefficient) AND variation (e.g. standard deviation) or associated estimates of uncertainty (e.g. confidence intervals) |
| ☐ | ☒ | For null hypothesis testing, the test statistic (e.g. *F*, *t*, *r*) with confidence intervals, effect sizes, degrees of freedom and *P* value noted<br>*Give P values as exact values whenever suitable.* |
| ☒ | ☐ | For Bayesian analysis, information on the choice of priors and Markov chain Monte Carlo settings |
| ☒ | ☐ | For hierarchical and complex designs, identification of the appropriate level for tests and full reporting of outcomes |
| ☒ | ☐ | Estimates of effect sizes (e.g. Cohen's *d*, Pearson's *r*), indicating how they were calculated |

*Our web collection on statistics for biologists contains articles on many of the points above.*

## Software and code

Policy information about availability of computer code

| Data collection | Using SRA Toolkit (2.8.0) to download genomes and short reads from NCBI |
|---|---|
| Data analysis | The following software were used in the analysis: Trimmomatic v0.39, EToKi v1.3, PROKKA v1.14, ISfinder v1, Kleborate v2.3.2, Kaptive v2.0.7, RecHMM v1, Phandango v1.3.0, BactDating v1.1, TreeTime v0.10.1, iTOL v6, TempEst v1.5.3, BLASTn v2.9.0+, MOB-Typer v3.1.0, BRIG v0.95, BBDuk v1, KRAKEN v2.1.2, minimap2 v2.26-r1175, samtools v1.2, SPARSE v1, R skygrowth v0.3.1, R APE  v5.7.1, LINcoding script v1, and search plasmic.git v1 (https://eithub.com/xiaoliu8 /search plasmic.git) |

For manuscripts utilizing custom algorithms or software that are central to the research but not yet described in published literature, software must be made available to editors and reviewers. We strongly encourage code deposition in a community repository (e.g. GitHub). See the Nature Portfolio guidelines for submitting code & software for further information.

## Data

Policy information about availability of data

All manuscripts must include a data availability statement. This statement should provide the following information, where applicable:

- Accession codes, unique identifiers, or web links for publicly available datasets
- A description of any restrictions on data availability
- For clinical datasets or third party data, please ensure that the statement adheres to our policy

The raw reads of 542 K. pneumoniae sequenced in this study were uploaded to NCBI database under the project PRJNA1028672. A total of 3,047 publicly available

# Research involving human participants, their data, or biological material

Policy information about studies with <u>human participants or human data</u>. See also policy information about <u>sex, gender (identity/presentation), and sexual orientation</u> and <u>race, ethnicity and racism</u>.

| | |
|---|---|
| Reporting on sex and gender | Sex and gender were not considered in study design. |
| Reporting on race, ethnicity, or other socially relevant groupings | Socially constructed or socially relevant categorization variable(s) were not used in our manuscript. |
| Population characteristics | A total of 1,017 CRKP strains were collected from different specimen sources (sputum, urine, blood, etc.) of K. pneumoniae infected patients admitted to 40 hospitals in 26 different Chinese cities between 2016-2020. More than half of the patients were male (667/1017; 65.59%). The mean age of the patients was 56.17 (SD 18.70).  Additional information on patients was not available to the authors. The sputum, urine, bile and pus used for metagenomic sequencing were collected from K. pneumoniae infected patients, 83.3%  (10/12) of whom were male, and the mean age of whom was 66.00 (SD 11.59). The serum of K. pneumoniae infected patients or healthy volunteers for SBA assay were also collected at Ruijin Hospital affiliated with Shanghai Jiaotong University but no information about patients or healthy volunteers was available. |
| Recruitment | Patients were recruited from the 40 member hospitals of CHINET China Bacterial Drug Resistance Surveillance Network (CHINET) every October 1-31 between 2016-2020. During this period, consecutive non-repetitive isolates were collected from all patients infected with K. pneumoniae. For multiple K. pneumoniae isolates from the same patient, the initial isolate was selected. Antimicrobial susceptibility testing was then performed to identify CRKP strains, and finally CRKP-infected patients were recruited. For the metagenomic sequencing and SBA assay, recruitment patients were approached at the time of diagnosis of K. pneumoniae infection and asked if they would like to participant in this study. Willing patients were recruited and samples were collected from each patient. There was no selection of infected patients, so there is no potential bias that could affect the outcome of the study. |
| Ethics oversight | The study protocol was approved by the Institutional Review Board of Huashan Hospital, Fudan University (No. 2018-408 and No.2019-460). Patients' serum, sputum, urine, bile  and pus were collected at Ruijin Hospital affiliated with Shanghai Jiaotong University and approval was obtained from the Ethics Committee of the Ruijin Hospital, Shanghai Jiaotong University (No.2017-205). |

Note that full information on the approval of the study protocol must also be provided in the manuscript.

# Field-specific reporting

Please select the one below that is the best fit for your research. If you are not sure, read the appropriate sections before making your selection.

☒ Life sciences ☐ Behavioural & social sciences ☐ Ecological, evolutionary & environmental sciences

For a reference copy of the document with all sections, see nature.com/documents/nr-reporting-summary-flat.pdf

# Life sciences study design

All studies must disclose on these points even when the disclosure is negative.

| | |
|---|---|
| Sample size | We collected a total of 1,017 CRKP strains between 2016 and 2020 in China. Then we selected 526 strains for whole genome sequencing based on sequencing and selecting criteria (see Extended Data Fig.1). In addition, we downloaded a total of 3,047 K. pneumoniae strains (Sequence type 11) from public databases, including 2,695 assembled genomes from NCBI SRA database, and 352 assembled genomes from Genome Sequence Archive in National Genomics Data Center, China.<br><br>We did PCR verification on 1,017 CRKPs in the carbapenem resistance experiment and discovered 18 non-carbapenemase-producing CRKP strains. Of which, 2 bacteria have been included in the 526 sequenced strains, thus we performed Whole Genome Sequencing on the other 16 bacteria.<br><br>Clinical samples were acquired during the capsule and O-Antigen trial parts. Only 12 clinical samples passed the sequencing standards after DNA extraction and quality control, thus metagenomic sequencing was done on these 12 samples.<br><br>We conducted capsule Quellung tests on KL64 and KL47 strains in 1,017 CRKPs and discovered that 70 strains were non-capsulated (NEKp group); consequently, based on the selection criteria (see Extended Data Fig.1), 70 encapsulated strains (EKp group) were chosen as controls for fitness testing. |
| Data exclusions | No data were excluded from the analyses. |
| Replication | Numbers of experimental replications were stated in the figure legend. All attempts at replication were successful. Bioinformatic analyses |

| | |
|---|---|
| Replication | were described in sufficient detail to reproduce the findings with the publicly available sequence data. We have given out all the source data as described in the manuscript. And the required scripts have been released in github, which has been described in the manuscript. |
| Randomization | For the mice experiment, mice with same age and gender used for evaluating the systemic infectivity of the bacteria were randomly allocated into different groups.<br>For the strategy of whole genome sequencing, we have set up a sequencing criterion (see Extended Data Fig.1), and we sequenced all strains that met the criteria.<br>For the fitness experiment, we conducted capsule Quellung tests on all KL64 and KL47 strains and discovered that 70 strains were non-capsular. Then, we selected 70 encapsulated strains randomly under the selection criteria of similar isolation regions and sources as showed in Extended Data Fig.1. |
| Blinding | For the mice experiment, investigators were blinded for evaluating the systemic infectivity of the bacteria, but no blinding was performed for the bacterial competition assay in mice because of the experimental complexity and group comparisons were not involved in this experiment. In other experiments, investigators were blinded to group allocation during data collection and analysis. In the part of bioinformatic analysis, blinding was unnecessary since this study is not a clinical trial. |

# Reporting for specific materials, systems and methods

We require information from authors about some types of materials, experimental systems and methods used in many studies. Here, indicate whether each material, system or method listed is relevant to your study. If you are not sure if a list item applies to your research, read the appropriate section before selecting a response.

### Materials & experimental systems

| n/a | Involved in the study |
|---|---|
| ☐ | ☒ Antibodies |
| ☐ | ☒ Eukaryotic cell lines |
| ☒ | ☐ Palaeontology and archaeology |
| ☐ | ☒ Animals and other organisms |
| ☒ | ☐ Clinical data |
| ☒ | ☐ Dual use research of concern |
| ☒ | ☐ Plants |

### Methods

| n/a | Involved in the study |
|---|---|
| ☒ | ☐ ChIP-seq |
| ☒ | ☐ Flow cytometry |
| ☒ | ☐ MRI-based neuroimaging |

## Antibodies

| | |
|---|---|
| Antibodies used | K47-antiserum and K64-antiserum (antibodies) used for Quellung Reaction were prepared by ourselves due to the limited availability of commercial antibodies. |
| Validation | K47-antiserum and K64-antiserum were confirmed against the capsule of K47 and K64 strains, respectively, in our previous studies (https://doi.org/10.1128/msphere.00271-22, https://doi.org/10.3390/antibiotics10020144). |

## Eukaryotic cell lines

Policy information about cell lines and Sex and Gender in Research

| | |
|---|---|
| Cell line source(s) | Murine macrophage cell line RAW 264.7 (ATCC Cat#TIB-71) was used in the study. |
| Authentication | The cell line was not authenticated. |
| Mycoplasma contamination | The cell line was not tested for mycoplasma contamination. |
| Commonly misidentified lines<br>(See ICLAC register) | No commonly misidentified cell lines were used in the study. |

## Animals and other research organisms

Policy information about studies involving animals; ARRIVE guidelines recommended for reporting animal research, and Sex and Gender in Research

| | |
|---|---|
| Laboratory animals | All animal experiments were performed in female 7-week-old C57BL/6 mice. |
| Wild animals | This study did not involve wild animals. |
| Reporting on sex | Sex was not considered in study design. |
| Field-collected samples | This study did not involve samples collected from the field. |

| Ethics oversight | All animal procedures were performed in accordance with the Regulations for the Administration of Affairs Concerning Experimental Animals and approved by the Animal Ethics Review Committee of School of Pharmacy, Fudan University (Project Number 2023-03-HSYY-QXH-31) and the Animal Ethics Review Committee of Shanghai Jiaotong University (project number A-2023-009). |
|---|---|

Note that full information on the approval of the study protocol must also be provided in the manuscript.

# Plants

| Seed stocks | *Report on the source of all seed stocks or other plant material used. If applicable, state the seed stock centre and catalogue number. If plant specimens were collected from the field, describe the collection location, date and sampling procedures.* |
|---|---|
| Novel plant genotypes | *Describe the methods by which all novel plant genotypes were produced. This includes those generated by transgenic approaches, gene editing, chemical/radiation-based mutagenesis and hybridization. For transgenic lines, describe the transformation method, the number of independent lines analyzed and the generation upon which experiments were performed. For gene-edited lines, describe the editor used, the endogenous sequence targeted for editing, the targeting guide RNA sequence (if applicable) and how the editor was applied.* |
| Authentication | *Describe any authentication procedures for each seed stock used or novel genotype generated. Describe any experiments used to assess the effect of a mutation and, where applicable, how potential secondary effects (e.g. second site T-DNA insertions, mosiacism, off-target gene editing) were examined.* |

