## [Peer Review File · Nature Microbiology]

Peer Review Information

Journal: Nature Microbiology

Manuscript Title: Carbapenem-resistant *Klebsiella pneumoniae* capsular types, antibiotic resistance and virulence factors in China: a longitudinal, multi-center study

Corresponding author name(s): Professor Ping He

Reviewer Comments & Decisions:

Decision Letter, initial version:

Message: 17th August 2023

Dear Professor He,

Thank you for your patience while your manuscript "Epidemiology and role of surface polysaccharides in carbapenem-resistant *Klebsiella pneumoniae*: a multi-center, five-year surveillance in China" was under peer-review at Nature Microbiology. It has now been seen by 3 referees, whose expertise and comments you will find at the end of this email. Although they find your work of some potential interest, they have raised a number of concerns that will need to be addressed before we can consider publication of the work in Nature Microbiology.

In particular, you will see that the referees cite a need for further method details particularly regarding the transmission analyses, raise concerns over the parameters and genetic markers used to define strains as hypervirulent and made recommendations for redefining the *Klebsiella pneumoniae* lineages in line with accepted standards within the field. There were also requests for information on 11 isolates missing from the BioProject cited. Importantly, referee #1 raised a major issue in the results section regarding a need for rationale, methods and data to be presented more clearly, that results which derive from estimates need to be clear including how they are obtained and any associated measures of confidence and uncertainty, where appropriate. We feel that these are critical points which would need to be addressed for us to further consider a revised manuscript, alongside the remaining issues outlined in the referees' reports, which are clear and should be straightforward to address.

Should further experimental data allow you to address these criticisms, we would be happy to look at a revised manuscript.

We were also unfortunately unable to recruit a referee with expertise in the latter experimental analyses of virulence phenotypes associated with capsule and O-antigen loss in a timely manner. To avoid further delay, we have decided to move forwards with a revise decision at this stage. However, as this aspect of the manuscript has not yet been fully peer reviewed, should we decide to return a revised study to reviewers, we would recruit another referee to look into these aspects of the manuscript at that point.

We are committed to providing a fair and constructive peer-review process. Please do not

2hesitate to contact us if there are specific requests from the reviewers that you believe are technically impossible or unlikely to yield a meaningful outcome.

Please include a data availability statement as a separate section after Methods but before references, under the heading "Data Availability". This section should inform readers about the availability of the data used to support the conclusions of your study. This information includes accession codes to public repositories (data banks for protein, DNA or RNA sequences, microarray, proteomics data etc...), references to source data published alongside the paper, unique identifiers such as URLs to data repository entries, or data set DOIs, and any other statement about data availability. At a minimum, you should include the following statement: "The data that support the findings of this study are available from the corresponding author upon request", mentioning any restrictions on availability. If DOIs are provided, we also strongly encourage including these in the Reference list (authors, title, publisher (repository name), identifier, year). For more guidance on how to write this section please see: <http://www.nature.com/authors/policies/data/data-availability-statements-data-citations.pdf>

- * If you have not done so already we suggest that you begin to revise your manuscript so that it conforms to our Article format instructions at <http://www.nature.com/nmicrobiol/info/final-submission>. Refer also to any guidelines provided in this letter.

When submitting the revised version of your manuscript, please pay close attention to our [href="https://www.nature.com/nature-portfolio/editorial-policies/image-integrity">Digital Image Integrity Guidelines. and to the following points below:](https://www.nature.com/nature-portfolio/editorial-policies/image-integrity)

2- that unprocessed scans are clearly labelled and match the gels and western blots presented in figures.
- that control panels for gels and western blots are appropriately described as loading on sample processing controls
- all images in the paper are checked for duplication of panels and for splicing of gel lanes.

Note: This url links to your confidential homepage and associated information about manuscripts you may have submitted or be reviewing for us. If you wish to forward this e-mail to co-authors, please delete this link to your homepage first.

Nature Microbiology is committed to improving transparency in authorship. As part of our efforts in this direction, we are now requesting that all authors identified as 'corresponding author' on published papers create and link their Open Researcher and Contributor Identifier (ORCID) with their account on the Manuscript Tracking System (MTS), prior to acceptance. This applies to primary research papers only. ORCID helps the scientific community achieve unambiguous attribution of all scholarly contributions. You can create and link your ORCID from the home page of the MTS by clicking on 'Modify my Springer Nature account'. For more information please visit www.springernature.com/orcid.

If you wish to submit a suitably revised manuscript we would hope to receive it within 6 months. If you cannot send it within this time, please let us know. We will be happy to consider your revision, even if a similar study has been accepted for publication at Nature Microbiology or published elsewhere (up to a maximum of 6 months).

Yours sincerely,

Reviewer Expertise:

Referee #1: microbial genomics, phylogenetics, Klebsiella, AMR

Referee #2: AMR, CR-KP and KP epidemiology

Referee #3: microbial genomics, phylogenetics, Klebsiella, AMR

Reviewer Comments:

Reviewer #1 (Remarks to the Author):

This manuscript describes an important study that I expect will be of wide interest to both the basic and clinical microbiology communities, as carbapenemase-resistant *Klebsiella pneumoniae* (CRKP) is one of the most challenging hospital-acquired infections to treat (topping the WHO Priority pathogen list and also the GRAM study's estimates of mortality associated with AMR globally). Importantly, whilst there are many reports from China that make it clear that the country has a very significant problem with CRKP, and that the causative clones are distinct from those in other places such as Europe and North America, there has been relatively little genome data reported or made publicly available. This study, by reporting nation-wide surveillance data and coupling it with serotyping and whole-genome sequencing (deposited publicly) makes an important contribution to the global understanding of CRKP and its epidemiology. In addition, the authors shed light on a relatively understudied but interesting biological phenomenon, which is loss of capsule or O antigen during human infection. The detailed analyses presented here, cataloging the frequency at which this occurs amongst CRKP infections and various phenotypic correlates, are interesting. Their relevance to the resolution of infection, through natural immunity and/or potentially vaccine- or mAb-mediated immunity in future, is as yet unknown; but the data presented here is convincing that this phenomenon happens frequently enough during serious human infections that it should be further investigated and its clinical relevance explored.

My main concern is the style in which the Results section is written. The authors have adopted a narrative style that focuses on the interpretation and conclusions they are drawing from the data, rather than clearly presenting the results themselves and then explaining the logic of their interpretations. There is a lack of methodological detail in the Methods section; and the Results section hardly ever states clearly which methods / analyses / experiments the authors are referring to when they express their narrative interpretation. This is particularly problematic in the phylogenomics section, where they are describing inferences from complex analyses as though they are observed facts, with little to no consideration given to justifying the logic of their approach, the appropriateness or limitations of the method, and the degree of certainty attached to the results.

For example, the Results text presents the dates estimated from the phylodynamics analysis implemented in BactDating as though they are known/observed dates, however they are actually just inferences based on combined analysis of the ML tree (inferred from sequence data) and bacterial isolation dates. The authors really should be careful with phrasing these results to reflect that these are estimates and inferences from the data, not direct observations. Similarly more care is needed with phrasing the inferences drawn regarding geographical transfers (which reading the Methods, I think is actually inferred from ancestral state reconstruction on the tree, treating country/region as discrete traits however this is not explicitly stated) and K/O locus exchange events (which are I think simply inferred from visual inspection of the tree?).

I am not sure of Nature Microbiology's position on narrative style, however in my experience it is standard practice when presenting the results of such analyses to be clear about when one is presenting evidence (i.e. what analysis was done and what parameter estimates were obtained), vs interpretation of that evidence in terms of what it tells us about evolutionary history (which is not observed).

E.g. Line 123-124: "One subclade acquired the K64:O2 serotype in 2003 and experienced rapid national dissemination in the past decade." - this is an interpretation, and it is not clear from the text/figures/methods what analyses and results this interpretation is based on.

There are many other instances, I cannot give an exhaustive list, but those I noted in the phylodynamics paragraph (lines 119-136) are:

Line 126 "The K47 strains originated in Zhejiang" - Again this is an inference, not an observation; and I can not see what evidence this is drawn from. As with the dating inference, estimates of the geographic location of internal nodes/branches should be accompanied by some sort of uncertainty measure. I gather these were inferred using ancestral state reconstruction (Methods states TreeTime was used, I believe this implements ML ASR? Should be clarified) so the point estimate for the location of the root node should be accompanied by a probability value. From visual inspection, I would not be surprised if the probability was low, with Beijing being a well-supported alternative location for for the root. Thus the probabilities should be reported and uncertainty discussed.

Lines 122-123 "The most recent common ancestor (MRCA) of L1 was in K47:OL101, tracing back to before 1983." - I am not sure what is meant by this statement, particularly the "was in K47:OL101"? The idea that the MRCA that existed in 1983 carried KL47 and OL101 is an inference, not an observation; what is it based on?

As per previous comment, I think this should be expressed more clearly to separate evidence from interpretation of the evidence. E.g. "Using ML inference to infer a dated tree from the ML tree and tip dates using Bactdating, we estimate that the MRCA of L1 existed circa 1983 [insert uncertainty measure]. Using ancestral state reconstruction of K and O loci on the dated tree, we estimate this MRCA carried KL47 and OL101 loci."

Line 127 "These strains settled down in Beijing" - do you mean they became established in Beijing followed by sustained local transmission?

Lines 127-129 "These strains settled down in Beijing and were repetitively transmitted to its neighbouring regions, contributing to the predominance of K47 in northern regions." - Again this is a narrative interpretation, and needs to be supported by clear logic of what evidence this is based on. Same for the remaining sentences in this section.

Line 131 "K64.2 & K64.5" - what does this refer to? New lineage nomenclature should be introduced explicitly not used implicitly

Lines 133-134 "Summarizing all transmissions together, we found that 53.6% (165/308) of the observed transmissions in L1 were associated with two major sources of Beijing and Shanghai (Fig. 4C)." - The authors need to be clear in the Methods and/or Results what they mean by "Summarising all transmissions". Do you mean you are counting the number of branches where the ancestral state reconstruction in TreeTime inferred the location of a child node differed from that of its parent node? I am guessing this based on personal familiarity with similar analyses of my own data but readers should not be left to guess.

Lines 135-136 As with previous comment, please clarify what is meant by "Beijing mediated 75 transmissions"... do you mean there were 75 branches where the parent node

was inferred as Beijing and the child node somewhere else?

I have several minor comments that should be addressed before acceptance for publication, to ensure the data is presented as clearly as possible.

1. Please mention in the abstract that the study is restricted to adult patients

2. Lines 22-23 - "incidence of CRKP in China increased from 2.5% to 22%" please provide units for incidence, e.g. this is usually expressed as number of incident infections per 1000 people or per 1000 bed-days

Uncertainty estimates should be provided for all dates estimated (e.g. 95% HPD or CI). I note figure 4C reports intervals of some kind () for some of the dated nodes, but it is not stated in the legend or methods what these intervals represent. All date estimates should be accompanied by uncertainty intervals, in figures and text.

3. Line 53 please define 'CRKP isolates', i.e. how is carbapenem resistance defined? Resistance to meropenem? Imipenem? Either? Both?

4. Lines 60-61, for the 18 carbapenemase-negative CRKP strains, was the phenotype and genotype confirmed with repeat experiments? What was the process and timing of AST vs sequencing in the lab? Could it be that they carried a carbapenemase-producing plasmid that was lost during passage to extract DNA?

5. Line 64 "Capsular genotyping revealed 58 different K types..."

- Having read the rest of the manuscript, my understanding is that "Capsular genotyping" here is based on wzi alleles and not whole genome sequencing, please clarify this here so readers know what result is being referred to.

- Genotyping detects K loci, not K types (these are inferred from the detected K loci, or assayed directly with serology), please be clear what is being measured directly (K locus type), vs what you are inferring (antigen/serotype)

- Suggest rephrase for clarity and specificity "Capsular genotyping via amplification and sequencing of wzi predicted 58 different K loci, the majority associated with biosynthesis of K64 (50.4%) and K47 (25.9%) capsular antigens."

- Note the same issues apply to the legend of Figure 2 which describes this data - legend text currently refers to 'serotypes' and 'K64' but this is not what is measured, instead you are reporting on wzi alleles

- Later in the manuscript it is mentioned that serology was done on isolates genotyped as K47 and K64 (although it is not really clear what was done on which strains), it would make sense to mention here (i.e. when discussing the distribution of K types) that the authors did in fact do serology to demonstrate that the KL47/KL64 strains did in fact express the K47/K64 antigens.

6. Lines 74-78 Again please be specific about how you are determining carbapenemases here.

- I gather this is based on PCR rather than whole genome sequencing? This should be clear in the statement of results: e.g. "Using PCR to detect carbapenemase genes, we found

XXX". This would help readers to understand why the alleles are not identified here, e.g. text refers to bla_{NDM}, rather than specifying NDM-1, NDM-5 etc

7. Line 81 - Again please be specific here that you are referring to PCR screening for these virulence genes

8. Lines 98-105 attribute virulence genes to specific plasmid variants, however I can't see in the Methods what data and analyses this is based on. Is it based on whole genome sequencing? This has not been mentioned yet at this point in the manuscript. If you sequenced all hv-CRKP (defined by PCR) and analysed their plasmid sequences, please state that here for clarity, e.g. line 98 could start "We sequenced the genomes of all hv-CRKP and found the varying levels..."

9. Lines 108-109 - how were the subset of K47 and K64 isolates chosen for sequencing? Are they a random selection? Is this what the "Sequencing criteria" box in Figure 1 is about? A rationale for sample selection should be given in the text, so readers can understand what this sample represents.

10. Line 115 refers to ST11 "Lineage 1" - how is this lineage defined? I don't see any mention of lineages in the Methods section, and the text states it was by visual inspection. It is not good scientific practice to introduce lineage nomenclatures in a paper without defining them using objective criteria, or providing any way for others to understand what they represent. A good solution to this for *Klebsiella* is the recently developed LINcode system, see doi: 10.1093/molbev/msac135. This system assigns barcodes to each genome, with different levels of resolution, using stable nomenclature (available via the *K. pneumoniae* MLST database at <https://bigsd.b.pasteur.fr/klebsiella/> and also via Pathogenwatch at <https://pathogen.watch>). A lineage can then easily be defined in the paper based on the LINcode nomenclature, and future readers can know exactly what this means and compare their own genomes to see if they have examples of the same lineage etc. An alternative would be to define lineages using an objective approach such as a pairwise patristic distance threshold, or the R package 'treestructure'.

11. Line 112 - It would be useful for readers to know that the tree being referred to is recombination-filtered, this is just as relevant to note as the fact that it is a maximum-likelihood tree. Please note this in text, e.g. "...generated a recombination-filtered maximum-likelihood phylogeny..." and also in the figure legend

12. Uncertainty estimates should be provided for all dates estimated (e.g. 95% HPD or CI). I note figure 4C reports intervals of some kind (in brackets) for some of the dated nodes, but it is not stated in the legend or methods what these intervals represent. All date estimates should be accompanied by uncertainty intervals, in figures and text.

13. The results of BactDating, or any other temporal analysis, are only meaningful if there is evidence of temporal signal in the data. Please provide evidence of this in the form of date-randomisation test (plotting root-to-tip distances vs tip dates and reporting R2 is also useful although not a replacement for date-randomisation test). As BactDating infers parameters using MCMC, it is important to also provide evidence that the chain has converged. The methods should also state how much of the chain was removed as burn-in.

7

- As the tree figure (4c) is so small, I struggle to see a lot of the details referred to in the text, e.g. "there were also 14 other K types in L1, some of which formed local clusters, i.e., KL108 in Sichuan and KL110 in Jiangsu" => I am unable to identify which colours and clusters this refers to. Please provide a larger figure in supplementary or, ideally, an interactive form of the tree that readers can zoom into (e.g. using the free Microreact.org tool)

14. Lines 139-140. This is the first mention of serology - was this done on all those isolates identified as K64 or K47 by wzi sequencing? It would be much easier for readers to follow what was done if there were a few narrative statements in the Results, e.g. "All isolates identified as K64 or K47 by wzi sequencing were subjected to serology, and X%..." As noted above (comment 5), it would also be useful to note this result much earlier, if the authors did indeed confirm that all isolates with K64/K47 associated wzi were confirmed by serology react with K64/K47 antisera this means they could actually describe these strains as K64/K47 rather than just carrying the loci.

15. Please specify inclusion and exclusion criteria for CRKP infections in the surveillance study. Were all clinical infections with CRKP diagnosed at all 40 sentinel hospitals during 2016-2020 included in the analysis?

16. The authors refer to their study as 'nation-wide' in several instances, and 40 hospitals in 27 Chinese cities sounds impressive. For readers like myself who are not overly familiar with Chinese geography or demographics, it would be helpful to put this in some kind of context. E.g. what fraction of the Chinese population lives in these 27 cities? Are these the largest cities? Are there major regions/provinces that are not included in the surveillance? The map in Figure 2 shows some provinces with no data displayed (Guangxi, Hainan, Chongqing, Hebei, Shanxi; no pie charts nor flags to indicate <10 isolates) - where these not included in surveillance, or were they included but no CRKP identified?

16. Antimicrobial susceptibility testing: Please specify the breakpoints used as not all readers will have access to the specific version of CLSI documents to figure this out.

17. Patient demographic data is mentioned for the surveillance study, how was this collected? From hospital records? Was any data missing? Was participant consent obtained?

18. Did the surveillance study receive ethical approval? The Ethics statement (lines 250-256) details ethics for specific subsidies at two hospitals but does not address the surveillance study.

19. Figure 2 legend - The figure title is "K serotype..." but as I understand it, the data reported here is wzi alleles not serotypes, and the relationship between molecular typing and serology was confirmed only for isolates genotyped as K47 and K64.

20. Check spelling of "multi-centre" in title vs "multicenter" in abstract

* check to make sure they cite the Broad paper on wzc mutants in UTI

Do they address that the mutations are not passed on, ie they are not actually fit to transmit

* check methods clearly defines inclusion criteria... is this all CRKP from the sentinel hospitals?

* How many provinces/municipalities are there in china, ie is n=27 a large or small proportion of these? Claiming nationwide

Reviewer #2 (Remarks to the Author):

The manuscript entitled 'Epidemiology and role of surface polysaccharides in carbapenem-resistant *Klebsiella pneumoniae*: a multi-center, five-year surveillance in China' describes a nationwide, long-term epidemiological investigation of CRKPs in China. They demonstrated the dominance of two serotypes of K47 and K64 in China during 2016-2020. The manuscript is original and well conducted contributing to better understanding CRKPs serotypes prevalence in China and also stepping up forward the development of new immunological strategies against CRKP infections.

L.62 – substitute acquired for had

L.82-83 – Classification of HvKp should be reviewed. Using only the presence of *iucA* and *rmpA/rmpA2* genes is not reliable. The application of the virulence score would add reliability to the nomenclature (<https://doi.org/10.1038/s41467-021-24448-3>). Moreover, how do you guarantee *rmpA/rmpA2* are not disrupted? Detection of these genes is commonly associated with a high rate (at least half of the isolates) of disruptions/mutations in the mucoid phenotype regulators causing their loss of functionality and lack of hypermucoid characteristic (<https://doi.org/10.1080/21505594.2015.1016703>, <https://doi.org/10.1128/aac.01424-22>).

How can you explain the very low prevalence of commonly hypervirulent capsular types K1 and K2 isolates in your collection, once those isolates are very frequent in China? (<https://doi.org/10.1128/aac.02173-15>, <https://doi.org/10.1128/AAC.02523-14>, <https://doi.org/10.1128/Spectrum.01305-21>)

L.152 – linkbetween > link between

L.230 – metropolises > metropolis

L.315 – There are a total of 154 BioSamples under the BioProject PRJNA588307 (<https://www.ncbi.nlm.nih.gov/bioproject/?term=PRJNA588307>). Maybe the authors made a mistake and it was just a writing issue. But if not, from where the authors took the additional 11 isolates (ST11)? If they are not under this BioProject, please clarify this issue in the manuscript.

Figure 2 - Please identify the square on the bottom right in the Figure description.

Figure 3 – As mentioned above, please review HvKp nomenclature and classification, most

9

of the isolates do not carry all the searched virulence genes (5).

Extended data Fig 2-C – It is not easy to identify the dashed blue line (19-89Δcps). Change the pattern of dashes or put something else for better visualization.

Extended data Fig 2-D – Same as above. But in this case, the green dashed line isn't clear.

Extended data Fig. 3 - ...and LPS silver staining and gemone > genome

>Methods<

L.3 and L.18-19 - A significant percentage of *K. pneumoniae* and other Enterobacterales collected from biological materials such as sputum and urine are often not causing infection to the patient and rather colonizing them. How do you confirm that all those isolates from sputum (458/1017) and urine (120/1017) are causing an infection instead of just colonizing those biological sites? To confirm that they were infecting those patients you need more clinical data and a score (SOFA,...). Can you add it?

L.9 – Add CLSI reference

L.13 – Add FDA reference

L.22-28 – Why you did not add K1 capsular type to your PCR screening?

L.55 – What are 5e6 samples?

L.62 and 64 – standardize the number (6,779 or 6779). Do it for all the other information presented similarly.

L.64 – Why did you align 523 instead of the 526 isolates (207 K47 and 319 K64) you have the WGS? If possible, include the whole data in this alignment.

L.65-66 – Add a table showing this summarized data in the supplementary material.

L.104 - bacterial > specimens

L.114 – used in Kaptive > obtained from Kaptive

L.159-163 – Was there any experiment to assure hygromycin was not killing the resistant cells in the more diluted plates? How do you prevent having a biased result and therefore, promote a higher/lower growth for the sensitive samples? How do you calculate your competitive index value?

L.221 – Reference for the RNA sequencing.

L.236 – Reference for the SBA assay.

L.257-259 – If available, please add the approval number obtained from the Animal Ethics Review Committee of Shanghai Jiaotong University.

L.266 – Same as mentioned above. Check the numbers here too. There is a total of 154 BioSamples under the BioProject PRJNA588307 (<https://www.ncbi.nlm.nih.gov/bioproject/?term=PRJNA588307>).

Reviewer #3 (Remarks to the Author):

This report describes a 5 year surveillance study on CRKP across different regions of China. There are two main strands to the paper – the first is an epidemiological study focussing on the disitrubution and temporal dynamics of K and O types, AMR and virulence genes, as well as the national routes of transmission of the ST11 L1 lineage. The second focus is the surprisingly high prevelance of NEKP (non-encapsulated Kp) strains, and how these might be selectively maintained. Both aspects of the study are carefully and comprehensively conducted, and both provide very interesting and novel results. If I have an over-arching criticism of the paper is that these two strands are not well connected, it is almost like two papers in one, but despite this rather disjointed narrative I found the paper very well written, thorough and convincing. I found the transmission analysis very interesting, but would have liked to see a bot ore methodological detail on this. It is also claimed that the transmission patterns mirror health-care seeking behaviour but no data is provided to support this point. Other than this, I only have minor points for consideration:

Abstract:

The phrase 'propose a correlation between these transmissions and ... patient movements' – this is rather vague, it would be better if the authors managed to articulate exactly what it is they have found here.

The word 'universal' is too strong, just say 'common'.

Line 60 – delete 'etc'

Line 64 – use percentages rather than fractions (3/4)

Line 92 – $p=0.06$ - only just outside significance - according to figure 3B there has been an increase during the early stage of the study period, this might be worth pointing out.

Line 127 – use 'became established' instead of 'settled down'.

Line 128 'repeatedly' rather than 'repetitively'

Figure 3C – remove the ML tree – a tree with only 3 nodes is pretty pointless.

Author Rebuttal to Initial comments

NMICROBIOL-23061494A

11Epidemiology and role of surface polysaccharides in carbapenem-resistant *Klebsiella pneumoniae*: a multi-center, five-year surveillance in China

Reviewer #1 (Remarks to the Author):

This manuscript describes an important study that I expect will be of wide interest to both the basic and clinical microbiology communities, as carbapenemase-resistant *Klebsiella pneumoniae* (CRKP) is one of the most challenging hospital-acquired infections to treat (topping the WHO Priority pathogen list and also the GRAM study's estimates of mortality associated with AMR globally). Importantly, whilst there are many reports from China that make it clear that the country has a very significant problem with CRKP, and that the causative clones are distinct from those in other places such as Europe and North America, there has been relatively little genome data reported or made publicly available. This study, by reporting nation-wide surveillance data and coupling it with serotyping and whole-genome sequencing (deposited publicly) makes an important contribution to the global understanding of CRKP and its epidemiology. In addition, the authors shed light on a relatively understudied but interesting biological phenomenon, which is loss of capsule or O antigen during human infection. The detailed analyses presented here, cataloging the frequency at which this occurs amongst CRKP infections and various phenotypic correlates, are interesting. Their relevance to the resolution of infection, through natural immunity and/or potentially vaccine- or mAb-mediated immunity in future, is as yet unknown; but the data presented here is convincing that this phenomenon happens frequently enough during serious human infections that it should be further investigated and its clinical relevance explored.

12My main concern is the style in which the Results section is written. The authors have adopted a narrative style that focuses on the interpretation and conclusions they are drawing from the data, rather than clearly presenting the results themselves and then explaining the logic of their interpretations. There is a lack of methodological detail in the Methods section; and the Results section hardly ever states clearly which methods / analyses / experiments the authors are referring to when they express their narrative interpretation. This is particularly problematic in the phylogenomics section, where they are describing inferences from complex analyses as though they are observed facts, with little to no consideration given to justifying the logic of their approach, the appropriateness or limitations of the method, and the degree of certainty attached to the results.

Response: Thanks. We have completely rewritten the phylogenomics section in a more descriptive tone, with additional sentences describing the methodology, justifications, and motivations (see below for details).

For example, the Results text presents the dates estimated from the phylodynamics analysis implemented in BactDating as though they are known/observed dates, however they are actually just inferences based on combined analysis of the ML tree (inferred from sequence data) and bacterial isolation dates. The authors really should be careful with phrasing these results to reflect that these are estimates and inferences from the data, not direct observations. Similarly more care is needed with phrasing the inferences drawn regarding geographical transfers (which reading the Methods, I think is actually inferred from ancestral state reconstruction on the tree, treating country/region as discrete traits however this is not explicitly stated) and K/O locus exchange events (which are I think simply inferred from visual inspection of the tree?).

Response: Thanks for the comments. We have rewritten the sentences in the revised manuscript to reflect the fact that they are inferences (manuscript lines 132-140), and for the inferences of geographic transfers (lines 141-158).

Revised: (Manuscript lines 132-140) Therefore, based on the phylogeny above, we estimated the spatiotemporal dynamics of Lineage L1 using BactDating and inferred the fluctuation of its effective population sizes using skygrowth in R. The ancestral inter-regional transmissions and serovar changes were also inferred using both TreeTime and the APE package in R. The most recent common ancestor (MRCA) of L1 was estimated to be present before 1983 (CI95% 1979-1987) in serotype K47:OL101, whereas the K64:O2 serotype was estimated to emerge in 2003 (CI95% 2002-2004) as a subclade of L1 and experienced rapid national dissemination in the past decade. Notably, we also predicted 14 other K genotypes in L1, some of which formed local clusters, i.e., KL25 in Sichuan and KL110 in Jiangsu (Fig. 4B).

Revised: (Manuscript lines 141-158) The K47 strains were estimated to originate in either Zhejiang or Beijing by different models, although all agreed on its long-term persistence in Zhejiang since 1990 (CI95% 1986-1993) (Fig. 4B and Supplementary Fig. 2). In contrast, the K64 clade was estimated to originate in Zhejiang by all models. We predicted a total of 308 cross-regional transmissions along the phylogeny, which were branches that connect nodes with different geographic inferences by TreeTime. We found that 53.6% (165) of these transmissions had parental nodes in either Beijing or Shanghai, the two major sources of transmissions (Fig. 4C). Among them, Beijing was estimated to contribute 75 transmissions (Fig. 4D), mostly to northern and northeastern regions, and Shanghai was estimated to contribute 90 transmissions, mainly to eastern and central regions.

Furthermore, the majority of the transmissions leave only one or a few isolates in the target regions. We, therefore, focused on 20 major events that were evidenced by ≥ 8 isolates in the target regions (Fig. 4B). Many of the major transmissions in the KL47 strains involve Beijing and its neighbouring, northern regions such as Henan (event KL47.2), Liaoning (event KL47.2), and Tianjin (event KL47.2 & KL47.7), contributing to the predominance of KL47 clade in northern China (blue route, Fig. 4C). Conversely, KL64 strains were estimated to experience nine major transmissions from Shanghai to central and eastern China including Jiangsu (event KL64.1), Hubei (event KL64.2), Sichuan (event KL64.5), and other regions (red route, Fig. 4C), resulting in their increasing KL64 frequencies.

I am not sure of Nature Microbiology's position on narrative style, however in my experience it is standard practice when presenting the results of such analyses to be clear about when one is presenting evidence (i.e. what analysis was done and what parameter estimates were obtained), vs interpretation of that evidence in terms of what it tells us about evolutionary history (which is not observed).

Response: Thanks for the comments. We have rewritten the sentences as shown above.

E.g. Line 123-124: "One subclade acquired the K64:O2 serotype in 2003 and experienced rapid national dissemination I n the past decade." - this is an interpretation, and it is not clear from the text/figures/methods what analyses and results this interpretation is based on.

Response: Thanks, the sentences have been completely rewritten as shown above.

There are many other instances, I cannot give an exhaustive list, but those I noted in the phylodynamics paragraph (lines 119-136) are:

Response: Thanks. The paragraph has been completely rewritten.

Line 126 "The K47 strains originated in Zhejiang" - Again this is an inference, not an observation; and I can not see what evidence this is drawn from. As with the dating inference, estimates of the geographic location of internal nodes/branches should be accompanied by some sort of uncertainty measure. I gather these were inferred using ancestral state reconstruction (Methods states TreeTime was used, I believe this implements ML ASR? Should be clarified) so the point estimate for the location of the root node should be accompanied by a probability value. From visual inspection, I would not be surprised if

the probability was low, with Beijing being a well-supported alternative location for for the root. Thus the probabilities should be reported and uncertainty discussed.

Response: Thanks. We have modified Figure 4B to reflect the uncertainty of ASR. Furthermore, the probability value of the root node is actually not bad in TreeTime (0.71). However, we recognized the potential uncertainty and used this opportunity to add additional inferences based on three different models (ER, ARD, and SYM) implemented in the APE package, which yielded different results (Supplementary Figure 2). We summarized all these into sentences in the revised manuscript.

Revised: (Manuscript lines 141-144) The KL47 strains were estimated to originate in either Zhejiang or Beijing by different models, although all agreed on its long-term persistence in Zhejiang since 1990 (CI95% 1986-1993) (Figs. 4B and Supplementary Figure 2). In contrast, the KL64 clade was estimated to originate in Zhejiang by all models.

Lines 122-123 “The most recent common ancestor (MRCA) of L1 was in K47:OL101, tracing back to before 1983.” - I am not sure what is meant by this statement, particularly the “was in K47:OL101”? The idea that the MRCA that existed in 1983 carried KL47 and OL101 is an inference, not an observation; what is it based on?

Response: Thanks. The paragraph has been completely rewritten. And the methods that are used for the inferences were now mentioned in the results (see above).

As per previous comment, I think this should be expressed more clearly to separate evidence from interpretation of the evidence. E.g. “Using ML inference to infer a dated tree from the ML tree and tip dates using Bactdating, we estimate that the MRCA of L1 existed circa 1983 [insert uncertainty measure]. Using ancestral state reconstruction of K and O loci on the dated tree, we estimate this MRCA carried KL47 and OL101 loci.”

Response: Thanks for the very kind suggestion. The methods that are used for the inferences were mentioned in the results (see above).

Line 127 “These strains settled down in Beijing” - do you mean they became established in Beijing followed by sustained local transmission?

Response: Thanks. The paragraph has been completely rewritten. And the transmission of KL47 is now described in the revised manuscript.

Revised: (Manuscript lines 152-155) Many of the major transmissions in the KL47 strains involve Beijing and its neighbouring, northern regions such as Henan (event KL47.2), Liaoning (event KL47.2), and Tianjin (event KL47.2 & KL47.7), contributing to the predominance of KL47 clade in northern China (blue route, Fig. 4C).

Lines 127-129 “These strains settled down in Beijing and were repetitively transmitted to its neighbouring regions, contributing to the predominance of K47 in northern regions.” - Again this is a narrative interpretation, and needs to be supported by clear logic of what evidence this is based on. Same for the remaining sentences in this section.

Response: The paragraph has been completely rewritten. See above.

Line 131 “K64.2 & K64.5” - what does this refer to? New lineage nomenclature should be introduced explicitly not used implicitly

Response: Thanks. We modified Figure 4 to highlight these major transmissions in the tree, and also added sentences in the revised manuscript to describe the motivations and methods for designating these lineages.

Revised: (Manuscript lines 150-152) Furthermore, the majority of the transmissions leave only one or a few isolates in the target regions. We, therefore, focused on 20 major events that were evidenced by ≥ 8 isolates in the target regions (Fig. 4B).”

Lines 133-134 “Summarizing all transmissions together, we found that 53.6% (165/308) of the observed transmissions in L1 were associated with two major sources of Beijing and Shanghai (Fig. 4C).” - The authors need to be clear in the Methods and/or Results what they mean by “Summarising all transmissions”. Do you mean you are counting the number of branches where the ancestral state reconstruction in TreeTime inferred the location of a child node differed from that of its parent node? I am guessing this based on personal familiarity with similar analyses of my own data but readers should not be left to guess.

Response: The paragraph has been completely rewritten. See above. And modified the sentences on the methodology in the revised manuscript.

Revised: (Manuscript lines 144-146) We predicted a total of 308 cross-regional transmissions along the phylogeny, which were branches that connect nodes with different geographic inferences by TreeTime.

Lines 135-136 As with previous comment, please clarify what is meant by “Beijing mediated 75 transmissions”... do you mean there were 75 branches where the parent node was inferred as Beijing and the child node somewhere else?

Response: Thanks. The sentences have been rewritten in the revised manuscript.

Revised: (Manuscript lines 146-149) We found that 53.6% (165) of these transmissions had parental nodes in either Beijing or Shanghai, the two major sources of transmissions (Fig. 4C). Among them, Beijing was estimated to contribute 75 transmissions (Fig. 4D), mostly to northern and northeastern regions, and Shanghai was estimated to contribute 90 transmissions, mainly to eastern and central regions.

I have several minor comments that should be addressed before acceptance for publication, to ensure the data is presented as clearly as possible.

1. Please mention in the abstract that the study is restricted to adult patients

Response: Thank you for your suggestions. We revised our manuscript according to your comments.

Revised: (Manuscript lines 8-10) Here, we present a five-year (2016-2020) multicenter study elucidating the distribution of capsular types, antimicrobial resistance, and virulence determinants among CRKPs from adult patients in different regions of China.

2. Lines 22-23 - “incidence of CRKP in China increased from 2.5% to 22%” please provide units for incidence, e.g. this is usually expressed as number of incident infections per 1000 people or per 1000 bed-days

Response: Thanks. We have rewritten the sentences in the revised manuscript.

Revised: (Manuscript lines 23-26) According to the CHINET Antimicrobial Surveillance Network, the resistance rate of *K. pneumoniae* to meropenem in China has been steadily increasing over the past decades from 3.0% in 2005 to 22.6% in 2022.

Uncertainty estimates should be provided for all dates estimated (e.g. 95% HPD or CI). I note figure 4C reports intervals of some kind () for some of the dated nodes, but it is not stated in the legend or methods what these intervals represent. All date estimates should be accompanied by uncertainty intervals, in figures and text.

Response: Thanks. We removed many of the dated nodes in Figure 4B as they are now irrelevant. And CI95% confidence intervals were added to all the remaining nodes and the main text. Sentences in the legend for Figure 4 have also been modified in the revised manuscript.

Revised: (the legend for Figure 4) Dates next to particular branches indicate their estimated date of origin with the 95% confidence intervals in the brackets. ... The star in the KL64 clade show a cluster of hypervirulent CRKP isolates, with their times of origin and 95% confidence intervals nearby.

3. Line 53 please define 'CRKP isolates', i.e. how is carbapenem resistance defined? Resistance to meropenem? Imipenem? Either? Both?

Response: Thanks. We have added the definition in the material and method.

Revised: (Supplementary method lines 23-24) The *K. pneumoniae* strains resistant to at least one of the carbapenem antibiotics (meropenem or imipenem) were defined as CRKP.

4. Lines 60-61, for the 18 carbapenemase-negative CRKP strains, was the phenotype and genotype confirmed with repeat experiments? What was the process and timing of AST vs sequencing in the lab? Could it be that they carried a carbapenemase-producing plasmid that was lost during passage to extract DNA?

Response: Thanks. The experimental process of this part was to first perform antimicrobial susceptibility testing, screen out CRKP strains, and then perform carbapenemase gene PCR and sequencing. If they are carbapenemase-negative CRKP strains, we perform antimicrobial susceptibility testing again to confirm that they are CRKP strains. These 18 carbapenemase-negative CRKP strains were retested and confirmed as CRKP strains by antimicrobial susceptibility testing (CLSI recommended broth microdilution method). Finally, whole-genome sequencing also confirmed that they were carbapenemase-negative strains.

205. Line 64 “Capsular genotyping revealed 58 different K types...”

- Having read the rest of the manuscript, my understanding is that “Capsular genotyping” here is based on wzi alleles and not whole genome sequencing, please clarify this here so readers know what result is being referred to.

- Genotyping detects K loci, not K types (these are inferred from the detected K loci, or assayed directly with serology), please be clear what is being measured directly (K locus type), vs what you are inferring (antigen/serotype)

- Suggest rephrase for clarity and specificity “Capsular genotyping via amplification and sequencing of wzi predicted 58 different K loci, the majority associated with biosynthesis of K64 (50.4%) and K47 (25.9%) capsular antigens.”

- Note the same issues apply to the legend of Figure 2 which describes this data - legend text currently refers to ‘serotypes’ and ‘K64’ but this is not what is measured, instead you are reporting on wzi alleles

- Later in the manuscript it is mentioned that serology was done on isolates genotyped as K47 and K64 (although it is not really clear what was done on which strains), it would make sense to mention here (i.e. when discussing the distribution of K types) that the authors did in fact do serology to demonstrate that the KL47/KL64 strains did in fact express the K47/K64 antigens.

Response: Thank you for your suggestions. We have rephrased the K types based on the arrangement of the K locus as the KL series, and revised our manuscript, Tables and Figures according to your comments.

Revised: (Supplementary method lines 114-115) The traditional serologic method, the Quellung assay, was performed to further confirm the K-type results for these KL47 (263 strains) and KL64 (513 strains) clinical strains in this study.

Revised: (Manuscript lines 66-67) Capsular genotyping revealed 58 different K loci, the majority belonged to KL64 (50.4%) and KL47 (25.9%) (Table 1).

6. Lines 74-78 Again please be specific about how you are determining carbapenemases here.

- I gather this is based on PCR rather than whole genome sequencing? This should be clear in the statement of results: e.g. "Using PCR to detect carbapenemase genes, we found XXX". This would help readers to understand why the alleles are not identified here, e.g. text refers to blaNDM, rather than specifying NDM-1, NDM-5 etc

Response: Thanks. We have described these information in the revised manuscript.

Revised: (Manuscript lines 60-61) Carbapenemase genes were detected by PCR in 98.2% (999/1,017) of the CRKP isolates,

7. Line 81 - Again please be specific here that you are referring to PCR screening for these virulence genes

Response: Thanks. We revised our manuscript according to your comments.

Revised: (Manuscript lines 85-86) We screened for the presence of five hypervirulence-associated genes (*rmpA*, *rmpA2*, *iroB*, *iucA*, and *peg344*) by PCR in all CRKP isolates,

8. Lines 98-105 attribute virulence genes to specific plasmid variants, however I can't see in the Methods what data and analyses this is based on. Is it based on whole genome sequencing? This has not been mentioned yet at this point in the manuscript. If you sequenced all hv-CRKP (defined by PCR) and

analysed their plasmid sequences, please state that here for clarity, e.g. line 98 could start “We sequenced the genomes of all hv-CRKP and found the varying levels...”

Response: Thanks. We have now expanded the methods section for prediction of plasmids and virulence plasmids.

Revised: (Supplementary method lines 96-111):

Identification of carbapenemase-encoding and hypervirulence-associated plasmids

To detect plasmids in the ST11 isolates, all 6,779 complete sequences of *Klebsiella* plasmids were downloaded from the NCBI RefSeq database (July 2022) and used as references. All 3,570 ST11 genomes were compared with the reference plasmid sequences using BLASTn, and the alignment regions for each plasmid among the assemblies were summarized, and only plasmids that have >50% of their sequences aligned with >85% identities were kept. For each genome, we reordered its associated plasmid references according to the alignment scores and selected the hits using a “Best-first” greedy algorithm that is similar to what is described in¹⁶. Briefly, for each iteration, the reference plasmid with the greatest alignment score was selected, and all contigs associated with it were taken out. This process was repeated until there was no hit left. Furthermore, to identify hypervirulence-associated plasmids, we predicted the presence of the carbapenemase or the five hypervirulence-associated genes (*iucA*, *iroB*, *rmpA*, *rmpA2*, and *peg344*) in the selected contigs based on BLASTn alignments. Only the plasmids that carry at least one of these functional genes were reported. The identified plasmids were then aligned onto the NCBI RefSeq database again to obtain similar plasmids, and had their incompatibility and MOB types determined using the MOB-Typer software¹⁷.

9. Lines 108-109 - how were the subset of K47 and K64 isolates chosen for sequencing? Are they a random selection? Is this what the “Sequencing criteria” box in Figure 1 is about? A rationale for sample selection should be given in the text, so readers can understand what this sample represents.

Response: Thanks. Strains were randomly selected for sequencing. We have added sequencing criteria to the method.

10. Line 115 refers to ST11 “Lineage 1” - how is this lineage defined? I don’t see any mention of lineages in the Methods section, and the text states it was by visual inspection. It is not good scientific practice to introduce lineage nomenclatures in a paper without defining them using objective criteria, or providing any way for others to understand what they represent. A good solution to this for *Klebsiella* is the recently developed LINcode system, see doi: 10.1093/molbev/msac135. This system assigns barcodes to each genome, with different levels of resolution, using stable nomenclature (available via the *K. pneumoniae* MLST database at <https://bigsd.b.pasteur.fr/klebsiella/> and also via Pathogenwatch at <https://pathogen.watch>). A lineage can then easily be defined in the paper based on the LINcode nomenclature, and future readers can know exactly what this means and compare their own genomes to see if they have examples of the same lineage etc. An alternative would be to define lineages using an objective approach such as a pairwise patristic distance threshold, or the R package ‘treestructure’.

Response: Thanks very much for the suggestion. We tested several tools for lineage assignments, including the cgLIN codes and treestructure mentioned by the reviewer, plus fastbaps that is based on sequence alignments. In particular, we failed to obtain the cgLIN codes published on the Pasteur website because many of the genomes in this study were not found there. So we genotyped all the ST11 genomes ourselves based on the scgMLST_629S scheme and separated the results into clonal groups (CGs) using their published LINcoding script (<https://gitlab.pasteur.fr/BEBP/LINcoding>) and with the described criteria (<https://bigsd.b.pasteur.fr/klebsiella/cgmlst-lincodes/>).

However, we found that none of the methods above generate lineages that are as clean as those by visual inspections (Figure 1 below). After careful consideration, we decided to merge the previously described lineages L2 to L10 together (see Figure 4A in the manuscript), as they were not fully described in the manuscript. This resulted in a simpler system with only 4 lineages, of which two were sort of consistent with the cgLINcodes.

Figure1. Application of LINcode, treestructure and fastbaps to the ST11 phylogeny tree.

11. Line 112 - It would be useful for readers to know that the tree being referred to is recombination-filtered, this is just as relevant to note as the fact that it is a maximum-likelihood tree. Please note this in text, e.g. "...generated a recombination-filtered maximum-likelihood phylogeny..." and also in the figure legend

Response: Thanks. We have rewritten the sentence in the revised manuscript.

Revised: (Manuscript lines 116-118) After filtering recombination regions, a maximum likelihood phylogeny of the ST11 lineage was estimated based on the remaining non-repetitive, non-recombinant SNPs in the core genome.

12. Uncertainty estimates should be provided for all dates estimated (e.g. 95% HPD or CI). I note figure 4C reports intervals of some kind (in brackets) for some of the dated nodes, but it is not stated in the legend or methods what these intervals represent. All date estimates should be accompanied by uncertainty intervals, in figures and text.

Response: Thanks. Answered above in question 2.

13. The results of BactDating, or any other temporal analysis, are only meaningful if there is evidence of temporal signal in the data. Please provide evidence of this in the form of date-randomisation test (plotting root-to-tip distances vs tip dates and reporting R2 is also useful although not a replacement for date-randomisation test). As BactDating infers parameters using MCMC, it is important to also provide evidence that the chain has converged. The methods should also state how much of the chain was removed as burn-in.

Response: Thanks. The root-to-tip distances plot and two date-randomisation tests were applied to the data and visualized in the newly added Supplementary Fig. 1. All tests confirmed the presence of a significant temporal signal.

We have also added sentences in the revised manuscript.

Revised: (Manuscript lines 129-132) Temporal signals were found in lineage L1 based on regression of root-to-tip genetic distance against sampling time ($R^2=0.18$, $p<0.0001$) and further demonstrated based on two date randomization tests (Supplementary Fig. 1 and supplementary methods).

- As the tree figure (4c) is so small, I struggle to see a lot of the details referred to in the text, e.g. “there were also 14 other K types in L1, some of which formed local clusters, i.e., KL108 in Sichuan and KL110 in Jiangsu” => I am unable to identify which colours and clusters this refers to. Please provide a larger figure in supplementary or, ideally, an interactive form of the tree that readers can zoom into (e.g. using the free Microreact.org tool)

Response: Thanks for the comment. We have now added arrows in Figure 4B pointing to all the clusters of rare K types. Furthermore, we have released a published version of the tree in Figure 4B at <https://itol.embl.de/shared/2LPbupyFGWsh9>.

14. Lines 139-140. This is the first mention of serology - was this done on all those isolates identified as K64 or K47 by wzi sequencing? It would be much easier for readers to follow what was done if there were a few narrative statements in the Results, e.g. “All isolates identified as K64 or K47 by wzi sequencing were subjected to serology, and X%...” As noted above (comment 5), it would also be useful to note this result much earlier, if the authors did indeed confirm that all isolates with K64/K47

associated wzi were confirmed by serology react with K64/K47 antisera this means they could actually describe these strains as K64/K47 rather than just carrying the loci.

Response: Thank you for your suggestions. We revised our manuscript according to your comments.

Revised: (Manuscript lines 162-163) All isolates identified as KL64 or KL47 by genotyping were subjected to serologic typing, and 7.4% (38/513) KL64 and 12.2% (32/263) KL47 isolates, respectively, did not react with the corresponding antisera in the Quellung reaction.

15. Please specify inclusion and exclusion criteria for CRKP infections in the surveillance study. Were all clinical infections with CRKP diagnosed at all 40 sentinel hospitals during 2016-2020 included in the analysis?

Response: Thanks. This was based on the active surveillance research carried out annually by the CHINET surveillance network. In this research, all consecutive non-duplicate CRKP clinical isolates from all patients with *K. pneumoniae* infection in October of each year in each hospital were included in the analysis.

This inclusion criteria has been added to the method.

Revised: (Supplementary method lines 3-5) A total of 1,017 CRKP strains were non-duplicate sequentially collected from *K. pneumoniae* infected patients admitted to 40 hospitals in 26 Chinese cities every October 1-31 between 2016-2020 by the China Antimicrobial Surveillance Network (www.chinets.com).

16. The authors refer to their study as 'nation-wide' in several instances, and 40 hospitals in 27 Chinese cities sounds impressive. For readers like myself who are not overly familiar with Chinese geography or demographics, it would be helpful to put this in some kind of context. E.g. what fraction of the Chinese

28population lives in these 27 cities? Are these the largest cities? Are there major regions/provinces that are not included in the surveillance? The map in Figure 2 shows some provinces with no data displayed (Guangxi, Hainan, Chongqing, Hebei, Shanxi; no pie charts nor flags to indicate <10 isolates) - where these not included in surveillance, or were they included but no CRKP identified?

Response: Thanks. These isolates were collected from 40 hospitals in 26 Chinese provinces or cities, not 27 Chinese provinces or cities (we have changed the data in the revised manuscript). The 26 provinces or cities account for 88.72% of the country's total population (Table 1), representing the vast majority of regions and population sizes. Therefore, we consider this study as a nationwide study. In this study, Shanxi Province and Tibet (members of the CHINET surveillance network) did not isolate CRKP strains. Other provinces and cities (including Guangxi, Hainan, Chongqing, Hong kong, Taiwan, and Macau) were not included in the surveillance network and therefore had no data.

Table 1. Chinese geography or demographics (2022)

Province or city	number of people	%
Guangdong	12656.8	8.78
Shandong	10162.79	7.05
Henan	9872	6.85
Jiangsu	8515	5.91
Sichuan	8374	5.81
Hebei	7420	5.15
Hu'nan	6604	4.58
Zhejiang	6577	4.56

Anhui	6127	4.25
Hubei	5844	4.06
Guangxi	5047	3.50
Yunnan	4693	3.26
Jiangxi	4527.98	3.14
Liaoning	4197	2.91
Fujian	4188	2.91
Shaanxi	3956	2.75
Guizhou	3856	2.68
Shanxi	3481.35	2.42
Chognqing	3213.3	2.23
Heilongjiang	3099	2.15
Xinjiang	2587	1.80
Gansu	2492	1.73
Shanghai	2475.89	1.72
Inner Mongolia	2401.17	1.67
Jilin	2347.69	1.63
Taiwan	2323.3593	1.61

Beijing	2184.3	1.52
Tianjin	1363	0.95
Hainan	1027.02	0.71
Hongkong	733.32	0.51
Ningxia	728	0.51
Qinghai	595	0.41
Tibet	364	0.25
Macao	67.28	0.05
Total	144100.2493	100.00

16. Antimicrobial susceptibility testing: Please specify the breakpoints used as not all readers will have access to the specific version of CLSI documents to figure this out.

Response: Thanks. We have added the breakpoints in the revised Supplementary method.

17. Patient demographic data is mentioned for the surveillance study, how was this collected? From hospital records? Was any data missing? Was participant consent obtained?

Response: Thanks. Demographic data pertaining to this study were obtained from routine hospital records. Informed consent was waived as only simple and non-identifying information about the patients was collected, including information such as city of origin, gender, age, etc. and did not include name, hospitalization number, etc.

18. Did the surveillance study receive ethical approval? The Ethics statement (lines 250-256) details ethics for specific subsidies at two hospitals but does not address the surveillance study.

Response: Thanks. Ethical approval was obtained for this study. Huashan Hospital affiliated with Fudan University was the lead unit of the whole surveillance network, and the approved ethics numbers were No. 2018-408 and No. 2019-460, and this ethics document can be used by other units for submission purposes. As this study involves part of the clinical specimen source obtained from Ruijin Hospital, this part of the ethics application was additionally submitted and approved.

19. Figure 2 legend - The figure title is “K serotype...” but as I understand it, the data reported here is wzi alleles not serotypes, and the relationship between molecular typing and serology was confirmed only for isolates genotyped as K47 and K64.

Response: Thank you for your suggestions. We revised Figure 2 legend according to your comments.

Revised: The capsular genotypes and antimicrobial profiling of CRKP isolates collected from 26 Chinese provinces or municipalities, 2016–2020.

20. Check spelling of “multi-centre” in title vs “multicenter” in abstract

Response: Thank you for your suggestions. We revised title according to your comments.

* check to make sure they cite the Broad paper on wzc mutants in UTI

Do they address that the mutations are not passed on, ie they are not actually fit to transmit

Response: We have cited the paper by Christoph M. Ernst, et al. in reference no. 21 in our manuscript.

We don't have solid animal experiment results to prove that NEKp strains are not fit for transmission. The results of bioinformatics analysis in our study showed that NEKp strains were scattered throughout the phylogenetic tree and did not form clusters, and combined with previous study reports that NEKp strains reduced fitness survival *in vitro* environment, we postulate that these NEKp strains have reduced transmissibility.

* check methods clearly defines inclusion criteria... is this all CRKP from the sentinel hospitals?

Response: This is based on the active surveillance research work carried out annually by the CHINET surveillance network (see above).

* How many provinces/municipalities are there in china, ie is n=27 a large or small proportion of these?
Claiming nationwide

Response: Thanks. Currently, China has a total of 34 provinces, municipalities and autonomous regions. In this study, these isolates were collected from 40 hospitals in 26 Chinese provinces or cities (not 27 Chinese provinces or cities, we changed the data in the revised manuscript). The 26 provinces or cities account for 88.72% of the country's total population (see Table 1 above), representing the vast majority of regions and population sizes. Therefore, we consider this study to be a nationwide study (see above).

Reviewer #2 (Remarks to the Author):

33The manuscript entitled 'Epidemiology and role of surface polysaccharides in carbapenem-resistant *Klebsiella pneumoniae*: a multi-center, five-year surveillance in China' describes a nationwide, long-term epidemiological investigation of CRKPs in China. They demonstrated the dominance of two serotypes of K47 and K64 in China during 2016-2020. The manuscript is original and well conducted contributing to better understanding CRKPs serotypes prevalence in China and also stepping up forward the development of new immunological strategies against CRKP infections.

L.62 – substitute acquired for had

Response: Thanks. We revised our manuscript according to your comments.

L.82-83 – Classification of HvKp should be reviewed. Using only the presence of *iucA* and *rmpA/rmpA2* genes is not reliable. The application of the virulence score would add reliability to the nomenclature (<https://doi.org/10.1038/s41467-021-24448-3>). Moreover, how do you guarantee *rmpA/rmpA2* are not disrupted? Detection of these genes is commonly associated with a high rate (at least half of the isolates) of disruptions/mutations in the mucoid phenotype regulators causing their loss of functionality and lack of hypermucoid characteristic (<https://doi.org/10.1080/21505594.2015.1016703>, <https://doi.org/10.1128/aac.01424-22>).

Response: Thank you for your comments. To date, much remains unknown about the specific characteristics responsible for *K. pneumoniae* hypervirulence factors. A previous study evaluating markers of hypervirulence identified PCR-based detection of *K. pneumoniae* virulence plasmid markers (*iuc*, *iro*, *rmpA*, *rmpA2*, *peg-344*) as highly diagnostic of hypervirulent vs. classical *K. pneumoniae* infection among human clinical isolates and also predictive of mortality in a murine sepsis model (DOI: <https://doi.org/10.1128/jcm.00776-18>). Recently, a cohort study evaluated the effect of acquisition of virulence plasmids carrying *rmpA/rmpA2* and aerobactin genes on clinical outcomes, and the result

34showed that acquisition of *iuc* and *rmpA/rmpA2* by classical strains was associated with increased morbidity and mortality (doi: 10.3201/eid2602.190594). These findings indicate that *iuc* and *rmpA/rmpA2* can be used as good predictive markers of high virulence.

Kleborate evaluates the virulence of *K. pneumoniae* by assigning virulence scores ranging from 0 to 5, depending on the presence of key loci associated with increasing risk (yersiniabactin < colibactin < aerobactin). According to the criteria, strains that genetically possess *iuc* loci (whether complete or truncated) are assigned a virulence score of ≥ 3 (doi: 10.1038/s41467-021-24448-3).

In our study, the PCR method was used to assess the virulence of *K. pneumoniae* instead of the Kleborate virulence score because we did not sequence all the strains we isolated due to limited experimental funding. Indeed, there is the limitation of not being able to detect gene disruption. We analyzed whole genome sequenced strains and found that truncations of *rmpA2/rmpADC* are common in rmp-PCR positive strains.

We compared two virulence assessment methods (PCR and the Kleborate virulence score) in 526 whole genome sequenced strains in this study. The Kleborate virulence score result showed that 277 strains scored 4, 1 strain scored 3, 246 strains scored 1 and 2 strains scored 0. While 257 out of 526 strains were identified as hvKp strains (*iucA*⁺ and *rmpA/rmpA2*⁺) by PCR method, and all 257 strains were scored as 4 (the Kleborate virulence score). The twenty non-hvKp strains with a virulence score of 4 had incomplete or absent *rmpA2/rmpADC* or incomplete *iuc*, resulting in negative PCR results for these genes. These results suggest that the PCR method has similar accuracy in assessing virulence as the Kleborate virulence score.

The PCR method has limitations in distinguishing hvKp strains from cKp. However, these genes are likely to be linked on the hvKp virulence plasmid. We can assess the presence of these genes to understand the epidemic trend of virulence plasmids.

How can you explain the very low prevalence of commonly hypervirulent capsular types K1 and K2 isolates in your collection, once those isolates are very frequent in China?

(<https://doi.org/10.1128/aac.02173-15>, <https://doi.org/10.1128/AAC.02523-14>,
<https://doi.org/10.1128/Spectrum.01305-21>)

Response: Thank you for your thoughtful comments. K1 and K2 are prevalent in China mainly as carbapenem-susceptible *K. pneumoniae* (CSKP). We only studied the prevalence of CRKP strains in China, and the relatively low prevalence of K1 and K2 strains is consistent with the prevalence trend in our country. As mentioned in the paper you cited (<https://doi.org/10.1128/AAC.02523-14>), K1 and K2 accounted for 22.9% (16/70) and 10% (7/70), respectively. However, all but one of the 70 strains (whose serotype was not mentioned) were CSKP. The study by Lina Zhao et al. analyzed genomic data of 4643 strains of carbapenemase-carrying *Klebsiella pneumoniae* in China from public whole-genome data, and the results showed that K1 and K2 accounted for 0.6% and 0.9%, respectively (<https://doi.org/10.3390/antibiotics11081100>). Another paper you cited (<https://doi.org/10.1128/Spectrum.01305-21>) described 2 K2 strains out of 3285 CRKPs, and the ratio was very low. In the remaining paper you cited (<https://doi.org/10.1128/aac.02173-15>), K1 and K2 accounted for 3.4% (5/148) and 0.7% (1/148) of CRKP strains, respectively. The K1 ratio in this study was higher than our findings, possibly due to regional and temporal sampling bias.

L.152 – linkbetween > link between

Response: We revised our manuscript according to your comments.

L.230 – metropolises > metropolis

Response: We revised our manuscript according to your comments.

L.315 – There are a total of 154 BioSamples under the BioProject PRJNA588307 (<https://www.ncbi.nlm.nih.gov/bioproject/?term=PRJNA588307>). Maybe the authors made a mistake and it was just a writing issue. But if not, from where the authors took the additional 11 isolates (ST11)? If they are not under this BioProject, please clarify this issue in the manuscript.

Response: Thanks. The additional 11 isolates were from the BioProject PRJNA422332 (<https://www.ncbi.nlm.nih.gov/bioproject/?term=PRJNA422332>).

We have rewritten the data availability after Methods.

Revised: (Methods lines 311-325):

Data availability

A total of 2,530 public ST11 *K. pneumoniae* genomes were available via GitHub (https://github.com/xiaoliu8/2530Kp_sequences.git). Three hundred and fifty-two ST11 *K. pneumoniae* genomes were downloaded from Genome Sequence Archive in National Genomics Data Center, China National Center for Bioinformation/Beijing Institute of Genomics, Chinese Academic of Science under project PRJCA003173 (n=300) and PRJCA012323 (n=52). Another 165 ST11 *K. pneumoniae* genomes were downloaded from NCBI under project PRJNA588307 (n=154) and PRJNA422332 (n=11). The raw sequence data in this study have been deposited in the Genome Sequence Archive in National Genomics Data Center, China National Center for Bioinformation/Beijing Institute of Genomics, Chinese Academic of Science (PRJCA018053, GSA: CRA011666, CRA011681, CRA011774), and are publicly accessible (<http://ngdc.cncb.ac.cn/gsa>). The assembled genome sequences have been deposited in GitHub (https://github.com/xiaoliu8/assembled_542Kp.git). An interactive version of Figure 4B is published at <https://itol.embl.de/shared/2LPbupyFGWsH9>. Source data are provided with this paper.

Figure 2 - Please identify the square on the bottom right in the Figure description.

Response: We added the description of the square on the bottom right in the Figure 2B.

Figure 3 – As mentioned above, please review HvKp nomenclature and classification, most of the isolates do not carry all the searched virulence genes (5).

Response: We revised the figure legend of Fig 3A (see above).

Extended data Fig 2-C – It is not easy to identify the dashed blue line (19-89 Δ cps). Change the pattern of dashes or put something else for better visualization.

Response: We revised the color and pattern of the line of Δ cps strains in the Extended data Fig. 2-C according to your comments.

Extended data Fig 2-D – Same as above. But in this case, the green dashed line isn't clear.

Response: We revised the color and pattern of the line of Δ cps strains in the Extended data Fig. 2-D according to your comments.

Extended data Fig. 3 - ...and LPS silver staining and gemone > genome

Response: We revised the figure legend of Extended data Fig. 3 according to your comments.

>Methods<

L.3 and L.18-19 - A significant percentage of *K. pneumoniae* and other Enterobacterales collected from biological materials such as sputum and urine are often not causing infection to the patient and rather colonizing them. How do you confirm that all those isolates from sputum (458/1017) and urine (120/1017) are causing an infection instead of just colonizing those biological sites? To confirm that they were infecting those patients you need more clinical data and a score (SOFA,...). Can you add it?

Response: Thank you for your thoughtful comments. Unfortunately, we were unable to obtain more detailed clinical data for SOFA scoring. We cannot completely rule out the possibility of *K. pneumoniae* colonization in the lungs and urinary tract. However, among the strains enrolled in this study, we excluded strains that isolated from active surveillance for *K. pneumoniae* in patients without symptoms of infection in high-risk units. All CRKP strains were collected from patients diagnosed with *K. pneumoniae* infection. It is our opinion that the CRKP strains in this study are highly associated with clinical *K. pneumoniae* infections.

L.9 – Add CLSI reference

Response: We have added the CLSI reference in the revised manuscript.

L.13 – Add FDA reference

Response: We have added the FDA reference in the revised manuscript.

L.22-28 – Why you did not add K1 capsular type to your PCR screening?

Response: Thank you for your comment. *Wzi* sequencing is a rapid and simple method to determine the K type of most clinical isolates of *K. pneumoniae*. The study by Sylvain Brisse et al. showed different *wzi* alleles followed by corresponding K type (s) [Table S1, <https://doi.org/10.1128/jcm.01924-13>]. The vast majority of K types had distinct *wzi* sequences, but there are exceptions (e.g., K15K17K50K51). For example, K15 and K51 type strains possess the same *wzi*50 allele, and differentiation of the K15/K51 type relies on further PCR amplification of the variable sequence of the *wzy* genes. Among the specific alleles, *wzi*128 and *wzi*1 correspond uniquely to the K1 type. Therefore, K1 type can be predicted by *wzi* sequencing without further verification by *wzy* gene PCR.

L.55 – What are 5e6 samples?

Response: This refers to 5×10^6 . We have standardized the expression of numbers in the revised manuscript.

L.62 and 64 – standardize the number (6,779 or 6779). Do it for all the other information presented similarly.

Response: Thanks. We have standardized the numbers in the revised manuscript, method, and all Figures and Tables according to your comments.

L.64 – Why did you align 523 instead of the 526 isolates (207 K47 and 319 K64) you have the WGS? If possible, include the whole data in this alignment.

Response: Thanks. Among all 526 sequenced strains, three were not ST11 and thus not included in the phylogenetic tree. We have modified the sentences in the revised manuscript

Revised: (Manuscript lines 113-116) To obtain a phylogenetic overview of the two most common K-type CRKPs, we performed whole-genome sequencing on 207 KL47 and 319 KL64 isolates and found that >99% (523/526) of them belonged to ST11 (Supplementary Table 2). These 523 ST11 isolates were integrated with an additional 3,047 public ST11 genomes to form a global genomic collection of 3,570 isolates.

L.65-66 – Add a table showing this summarized data in the supplementary material.

Response: Thanks. Added. Please see the new Supplementary Table5.

L.104 - bacterial > specimens

Response: We revised our manuscript according to your comments.

L.114 – used in Kaptive > obtained from Kaptive

Response: We revised our manuscript according to your comments.

L.159-163 – Was there any experiment to assure hygromycin was not killing the resistant cells in the more diluted plates? How do you prevent having a biased result and therefore, promote a higher/lower growth for the sensitive samples? How do you calculate your competitive index value?

Response: Thank you for your thoughtful comments. The *K. pneumoniae* Δcps strains were constructed by replacing the *cps* gene with the hygromycin-resistant gene. The Δcps strains grew slightly slower on hygromycin LB plates than on LB plates after dilution plating, but there was no difference in the number

of colonies. Second, colonies of the Δcps strains appeared translucent on LB plates, and could be easily distinguished from EKp strains. We evaluated the capsule deletion rate of EKp strains in LB media for 24 hours, and showed a very low capsule deletion rate (range 0-0.4%). In the bacterial competition assay, there was no difference in the number of colonies between the hygromycin LB plates and translucent colonies in LB plates. We have added the formula for calculating competitive index value in the method.

Revised: (Supplementary method lines 163-166) Bacteria were then serially diluted and plated on LB and LB containing hygromycin plates. The ratio of Δcps (the number of colonies on hygromycin LB plates):WT (the number of colonies on LB plates minus the number of colonies on hygromycin LB plates) was calculated to obtain the competitive index value.

L.221 – Reference for the RNA sequencing.

Response: We added the reference for the RNA sequencing in the Methods.

L.236 – Reference for the SBA assay.

Response: We added the reference for the SBA assay in the Methods.

L.257-259 – If available, please add the approval number obtained from the Animal Ethics Review Committee of Shanghai Jiaotong University.

Response: We added the approval number obtained from the Animal Ethics Review Committee of Shanghai Jiaotong University in the Methods.

L.266 – Same as mentioned above. Check the numbers here too. There is a total of 154 BioSamples under the BioProject PRJNA588307 (<https://www.ncbi.nlm.nih.gov/bioproject/?term=PRJNA588307>).

Response: Thanks. Please see the answer above.

Reviewer #3 (Remarks to the Author):

This report describes a 5 year surveillance study on CRKP across different regions of China. There are two main strands to the paper – the first is an epidemiological study focussing on the distribution and temporal dynamics of K and O types, AMR and virulence genes, as well as the national routes of transmission of the ST11 L1 lineage. The second focus is the surprisingly high prevalence of NEKP (non-encapsulated Kp) strains, and how these might be selectively maintained. Both aspects of the study are carefully and comprehensively conducted, and both provide very interesting and novel results. If I have an over-arching criticism of the paper is that these two strands are not well connected, it is almost like two papers in one, but despite this rather disjointed narrative I found the paper very well written, thorough and convincing. I found the transmission analysis very interesting, but would have liked to see a bit more methodological detail on this. It is also claimed that the transmission patterns mirror health-care seeking behaviour but no data is provided to support this point. Other than this, I only have minor points for consideration:

Abstract:

The phrase ‘propose a correlation between these transmissions and ... patient movements’ – this is rather vague, it would be better if the authors managed to articulate exactly what it is they have found here.

Response: Thanks. We have revised the sentences in the revised Abstract.

43Revised: (Manuscript lines 10-13) We identify two metropolises, Shanghai and Beijing, responsible for the majority of nationwide CRKP transmissions and suggest that inter-regional healthcare seeking of the patients into these megacities might mediate the transmissions and result in different prevalences of KL47 and KL64 strains in eastern and northern China.

The word 'universal' is too strong, just say 'common'.

Response: We revised our manuscript according to your comments.

Line 60 – delete 'etc'

Response: We revised our manuscript according to your comments.

Line 64 – use percentages rather than fractions (3/4)

Response: We revised our manuscript according to your comments.

Line 92 – $p=0.06$ - only just outside significance - according to figure 3B there has been an increase during the early stage of the study period, this might be worth pointing out.

Response: We re-checked the statistical analysis and found that the P value was actually 0.61 instead of 0.06. The increase in the frequency of hv-CRKP in K64 during the early phase (2016-2018) was also not statistically significant ($p=0.15$). We revised our manuscript and corrected the P-value.

Line 127 – use 'became established' instead of 'settled down'.

Response: We revised our manuscript according to your comments.

Line 128 'repeatedly' rather than 'repetitively'

Response: We revised our manuscript according to your comments.

Figure 3C – remove the ML tree – a tree with only 3 nodes is pretty pointless.

Response: Thanks. Deleted.

Decision Letter, first revision:

Message: Our ref: NMICROBIOL-23061494A

21st December 2023

Dear Dr. He,

Thank you for your patience as we've prepared the guidelines for final submission of your Nature Microbiology manuscript, "Epidemiology and role of surface polysaccharides in carbapenem-resistant *Klebsiella pneumoniae*: a multi-center, five-year surveillance in China" (NMICROBIOL-23061494A). Please carefully follow the step-by-step instructions provided in the attached file, and add a response in each row of the table to indicate the changes that you have made. Please also check and comment on any additional marked-up edits we have proposed within the text. Ensuring that each point is addressed will help to ensure that your revised manuscript can be swiftly handed over to our production team.

If you have not done so already, please alert us to any related manuscripts from your

45group that are under consideration or in press at other journals, or are being written up for submission to other journals (see: <https://www.nature.com/nature-research/editorial-policies/plagiarism#policy-on-duplicate-publication> for details).

In recognition of the time and expertise our reviewers provide to Nature Microbiology's editorial process, we would like to formally acknowledge their contribution to the external peer review of your manuscript entitled "Epidemiology and role of surface polysaccharides in carbapenem-resistant *Klebsiella pneumoniae*: a multi-center, five-year surveillance in China". For those reviewers who give their assent, we will be publishing their names alongside the published article.

Nature Microbiology offers a Transparent Peer Review option for new original research manuscripts submitted after December 1st, 2019. As part of this initiative, we encourage our authors to support increased transparency into the peer review process by agreeing to have the reviewer comments, author rebuttal letters, and editorial decision letters published as a Supplementary item. When you submit your final files please clearly state in your cover letter whether or not you would like to participate in this initiative. Please note that failure to state your preference will result in delays in accepting your manuscript for publication.

Cover suggestions

COVER ARTWORK: We welcome submissions of artwork for consideration for our cover. For more information, please see our [guide for cover artwork](https://www.nature.com/documents/Nature_covers_author_guide.pdf).

Nature Microbiology has now transitioned to a unified Rights Collection system which will allow our Author Services team to quickly and easily collect the rights and permissions required to publish your work. Approximately 10 days after your paper is formally accepted, you will receive an email in providing you with a link to complete the grant of rights. If your paper is eligible for Open Access, our Author Services team will also be in touch regarding any additional information that may be required to arrange payment for your article.

Please note that *Nature Microbiology* is a Transformative Journal (TJ). Authors may publish their research with us through the traditional subscription access route or make their paper immediately open access through payment of an article-processing charge (APC). Authors will not be required to make a final decision about access to their article until it has been accepted. [Find out more about Transformative Journals](https://www.springernature.com/gp/open-research/transformative-journals)

Authors may need to take specific actions to achieve [open access](https://www.springernature.com/gp/open-research/funding/policy-)

46compliance-faqs"> compliance with funder and institutional open access

mandates. If your research is supported by a funder that requires immediate open access (e.g. according to [Plan S principles](https://www.springernature.com/gp/open-research/plan-s-compliance)) then you should select the gold OA route, and we will direct you to the compliant route where possible. For authors selecting the subscription publication route, the journal's standard licensing terms will need to be accepted, including [self-archiving policies](https://www.nature.com/nature-portfolio/editorial-policies/self-archiving-and-license-to-publish). Those licensing terms will supersede any other terms that the author or any third party may assert apply to any version of the manuscript.

Best regards,

Reviewer #1:

Remarks to the Author:

The comments have been adequately addressed

Reviewer #2:

Remarks to the Author:

A significant improvement can be observed on the quality and clarity of all the methodology and results presented by the authors in this last version of the manuscript.

I can see that the authors considered carefully all the reviewers suggestions/questions.

This is a well written manuscript that will bring valuable information for the scientific community.

Reviewer #3:

Remarks to the Author:

This manuscript has been substantially revised and improved in response to the detailed comments from the referees. It is well written, with an improved narrative, and would make a valuable contribution to the field.

Reviewer #4:

Remarks to the Author:

This manuscript reports a study on the epidemiological features of a large collection of *Klebsiella pneumoniae* isolates from Chinese hospitals (1,017 strains), such as the distribution of capsular serotypes, drug resistance, and virulence-associated genes. The authors have provided detailed responses to the critiques of the three reviewers. After carefully reading the manuscript and reviewing relevant literature, I feel this study has significant limitations in: 1) the insightfulness of what the epidemiology data mean to the understanding of *K. pneumoniae* pathogenesis and control, and 2) the completeness and interpretation of the data on acapsular and O antigen-deficient strains.

This manuscript has two major components: epidemiology of *Klebsiella pneumoniae* isolates and experimental characterization of capsule- or O antigen-deficient strains. Numerous papers have already characterized massive number of *K. pneumoniae* isolates from China. Zhao et al. have reported capsular serotype, drug resistance and virulence-associated genes of 4,643 strains (Molecular epidemiology of antimicrobial resistance, virulence and capsular serotypes of carbapenemase-carrying *Klebsiella pneumoniae* in China, *Antibiotics*, 2022, <https://doi.org/10.3390/antibiotics11081100>). Although that study was carried out by analyzing the existing genome data, the conclusions are highly relevant to what is described in this manuscript. I've also found three related epidemiology studies reporting the genomic or geographic features of *K. pneumoniae* isolates from China. Wang et al. compared epidemiological features of 991 *K. pneumoniae* strains that were collected from 991 patients of 71 hospitals in 7 countries during 2017-18, including 485 isolates from China [*Lancet Infect. Dis.* 2022, [https://doi.org/10.1016/S1473-3099\(21\)00399-6](https://doi.org/10.1016/S1473-3099(21)00399-6)]. Liu et al. have echoed that paper by publishing a short description of their own set of data on *K. pneumoniae* isolates from China (Molecular epidemiology of carbapenem-resistant *Klebsiella pneumoniae* in China, 2016–20, *Lancet Infect. Dis.* 2022, 22: 167–68). Dr. Hui Wang and her colleagues performed the epidemiological analysis of 1,052 *K. pneumoniae* strains that were collected from many Chinese hospitals during 2015-17 (Evolution of hypervirulence in carbapenem-resistant *Klebsiella pneumoniae* in China: a multicentre, molecular epidemiological analysis. *J. Antimicrob. Chemother.* 2020, <https://doi.org/10.1093/jac/dkz446>). These studies have substantially truncated the uniqueness of this paper.

The finding of unencapsulated *K. pneumoniae* and O antigen-deficient isolates is intriguing and potentially important for *K. pneumoniae* adaptation under evolving selection pressure. The authors did not provide the tissue/organ sources of these strains. Still, it is doubtful that they were isolated from human blood samples due to the essential role of the capsule in *K. pneumoniae* survival in the bloodstream. The authors elegantly tested the serum resistance of acapsular strains with phage depolymerase. However, it is premature to conclude that lacking capsule makes *K. pneumoniae* more resistant to serum due to enhanced O antigen production. There is a large body of literature on the importance of capsules in serum resistance of *K. pneumoniae* and other Gram-negative bacteria. Overturning the paradigm requires multiple lines of convincing data. Likewise, a causal relationship between the role of undecaprenyl phosphate and O antigen synthesis in acapsular strains requires additional data support.

48Author Rebuttal, first revision:

Reviewer #1:

Remarks to the Author:

The comments have been adequately addressed

Response: Thanks!

Reviewer #2:

Remarks to the Author:

A significant improvement can be observed on the quality and clarity of all the methodology and results presented by the authors in this last version of the manuscript.

I can see that the authors considered carefully all the reviewers suggestions/questions.

This is a well written manuscript that will bring valuable information for the scientific community.

Response: Thanks!

Reviewer #3:

Remarks to the Author:

This manuscript has been substantially revised and improved in response to the detailed comments from the referees. It is well written, with an improved narrative, and would make a valuable contribution to the field.

Response: Thanks!

49Reviewer #4:

Remarks to the Author:

This manuscript reports a study on the epidemiological features of a large collection of *Klebsiella pneumoniae* isolates from Chinese hospitals (1,017 strains), such as the distribution of capsular serotypes, drug resistance, and virulence-associated genes. The authors have provided detailed responses to the critiques of the three reviewers. After carefully reading the manuscript and reviewing relevant literature, I feel this study has significant limitations in: 1) the insightfulness of what the epidemiology data mean to the understanding of *K. pneumoniae* pathogenesis and control, and 2) the completeness and interpretation of the data on acapsular and O antigen-deficient strains.

This manuscript has two major components: epidemiology of *Klebsiella pneumoniae* isolates and experimental characterization of capsule- or O antigen-deficient strains. Numerous papers have already characterized massive number of *K. pneumoniae* isolates from China. Zhao et al. have reported capsular serotype, drug resistance and virulence-associated genes of 4,643 strains (Molecular epidemiology of antimicrobial resistance, virulence and capsular serotypes of carbapenemase-carrying *Klebsiella pneumoniae* in China, *Antibiotics*, 2022, <https://doi.org/10.3390/antibiotics11081100>). Although that study was carried out by analyzing the existing genome data, the conclusions are highly relevant to what is described in this manuscript. I've also found three related epidemiology studies reporting the genomic or geographic features of *K. pneumoniae* isolates from China. Wang et al. compared epidemiological features of 991 *K. pneumoniae* strains that were collected from 991 patients of 71 hospitals in 7 countries during 2017-18, including 485 isolates from China [*Lancet Infect. Dis.* 2022, [https://doi.org/10.1016/S1473-3099\(21\)00399-6](https://doi.org/10.1016/S1473-3099(21)00399-6)]. Liu et al. have echoed that paper by publishing a short description of their own set of data on *K. pneumoniae* isolates from China (Molecular epidemiology of carbapenem-resistant *Klebsiella pneumoniae* in China, 2016–20, *Lancet Infect. Dis.* 2022, 22: 167–68). Dr. Hui Wang and her colleagues performed the epidemiological analysis of 1,052 *K. pneumoniae* strains

50that were collected from many Chinese hospitals during 2015-17 (Evolution of hypervirulence in carbapenem-resistant *Klebsiella pneumoniae* in China: a multicentre, molecular epidemiological analysis. *J. Antimicrob. Chemother.* 2020, <https://doi.org/10.1093/jac/dkz446>). These studies have substantially truncated the uniqueness of this paper.

The finding of unencapsulated *K. pneumoniae* and O antigen-deficient isolates is intriguing and potentially important for *K. pneumoniae* adaptation under evolving selection pressure. The authors did not provide the tissue/organ sources of these strains. Still, it is doubtful that they were isolated from human blood samples due to the essential role of the capsule in *K. pneumoniae* survival in the bloodstream. The authors elegantly tested the serum resistance of acapsular strains with phage depolymerase. However, it is premature to conclude that lacking capsule makes *K. pneumoniae* more resistant to serum due to enhanced O antigen production. There is a large body of literature on the importance of capsules in serum resistance of *K. pneumoniae* and other Gram-negative bacteria. Overturning the paradigm requires multiple lines of convincing data. Likewise, a causal relationship between the role of undecaprenyl phosphate and O antigen synthesis in acapsular strains requires additional data support.

Response: Thanks for the comments. In recent years, many excellent works have reported on the prevalence of *K. pneumoniae* infection in China, including the distribution and prevalence of drug resistance, virulence, and capsular types, providing important insights for the development of more effective approaches to control CRKP infections. However, the focus of these studies is quite different from our research. In our studies, consecutive non-repetitive CRKP isolates were collected from patients infected with *K. pneumoniae* from 2016-2020. The uniqueness of our study is mainly reflected in the following aspects. First, our systematic investigation revealed the different prevalence of K serovars in northern and southern China. Second, based on an extra-large scale ancestral state reconstruction, we explored for the first time the landscape of the nationwide transmission pattern of ST11 strains, attributing the majority of inter-regional transmissions to only two sources of Beijing and Shanghai. Furthermore, we managed to correlate the two aspects together and identify potential links between CRKP transmission

51and trans-regional health seeking, which is strongly linked to health policy in China. Additionally, our analyses revealed the long-term persistence of an MDR plasmid in K64 population, which explains the observed continuous increase in hypervirulence. On top of the nationwide epidemiological investigation, we also explore the widespread presence of capsule-deficient and O-antigen-deficient strains and counted their proportions among isolates from different clinical specimens (see Fig. 4C and Extended Data Fig. 10D). In addition, metagenomic analysis of unprocessed clinical specimens confirmed the presence of capsular and O-antigen-deficient strains in the host, which had not been reported in previous studies. Furthermore, in contrast to research using only the public *K. pneumoniae* genome database, our investigation used antimicrobial susceptibility testing to screen CRKP strains. This approach allowed the inclusion of both carbapenemase-producing and non-carbapenemase-producing CRKP strains. In addition, we conducted comprehensive evaluations of the susceptibility of various K-type strains to a range of antibiotics, as well as assessing their fitness under various stress conditions. These aspects are missing in studies limited to genetic analysis only. Therefore, we believe that this is one of the most systematic studies on CRKP among multicenter studies in China and is highly innovative.

We cited and discussed the published studies mentioned by reviewers in the revised manuscript (manuscript lines 59-61).

Our study found that both capsular and O antigens play a critical role in serum resistance, which is consistent with previous reports in Gram-negative bacteria. The more interesting finding of our research is that clinically isolated unencapsulated *K. pneumoniae* strains contained a high proportion of serum-resistant strains, and showed that O-antigen expression was increased in strains after disruption of the initial glycosyltransferase gene. These results do not negate the role of the capsule in serum resistance, but are the consequence of excess O-antigen compensating for the antiserum capacity of unencapsulated strains. In most bacteria, glycan polymers including cell wall peptidoglycan, enterobacterial common antigen (ECA), CPS and LPS O antigen are generally assembled onto the lipid carrier, undecaprenyl phosphate (Und-P), and which is a limiting factor in the glycans biosynthesis. Hence, one Und-P-dependent pathway (capsular polysaccharides) deprivation is likely to have indirect effects on glycans synthesis in other pathways. We speculate that Und-P might affect the *K. pneumoniae* antiserum ability

by coordinating the expression of CPS and O antigen. The specific mechanisms and precise Und-P flux affected by CPS synthesis disruption need to be further investigated. We have rewritten the sentences in the revised manuscript as suggested (manuscript lines 223-230)

Revised: (Manuscript lines 59-61) In recent years, several studies have reported on the prevalence of *K. pneumoniae* infection in different countries, including the distribution and prevalence of drug resistance, virulence, and capsular genotypes¹³⁻¹⁵.

(Manuscript lines 223-230) These results suggest that disruption of capsule synthesis enhances the synthesis of O-antigen and ultimately increases antiserum capacity. We speculated that the cessation of CPS synthesis causes no consumption of undecaprenyl phosphate (Und-P) pool, a limiting factor in the biosynthesis of glycans²⁴, which in turn was used for the synthesis of excess O-antigen. Our investigation further revealed a remarkable association between the modulation of Und-P synthesis and the subsequent increase or decrease in serum resistance (Extended Data Fig. 8B). However, the specific mechanisms and precise Und-P flux affected by CPS synthesis disruption need to be further investigated.

Final Decision Letter:

Message: 18th January 2024

Dear Professor He,

I am pleased to accept your Article "Carbapenem-resistant *Klebsiella pneumoniae* capsular types, antibiotic resistance and virulence factors in China: a longitudinal, multi-center study" for publication in Nature Microbiology. Thank you for having chosen to submit your work to us and many congratulations.

Once your paper is typeset, you will receive an email with a link to choose the appropriate publishing options for your paper and our Author Services team will be in touch regarding

53any additional information that may be required. Once your paper has been scheduled for online publication, the Nature press office will be in touch to confirm the details.

You may wish to make your media relations office aware of your accepted publication, in case they consider it appropriate to organize some internal or external publicity. Once your paper has been scheduled you will receive an email confirming the publication details. This is normally 3-4 working days in advance of publication. If you need additional notice of the date and time of publication, please let the production team know when you receive the proof of your article to ensure there is sufficient time to coordinate. Further information on our embargo policies can be found here:

<https://www.nature.com/authors/policies/embargo.html>

Please note that *Nature Microbiology* is a Transformative Journal (TJ). Authors may publish their research with us through the traditional subscription access route or make their paper immediately open access through payment of an article-processing charge (APC). Authors will not be required to make a final decision about access to their article until it has been accepted. [Find out more about Transformative Journals](https://www.springernature.com/gp/open-research/transformative-journals)

Authors may need to take specific actions to achieve [compliance with funder and institutional open access mandates](https://www.springernature.com/gp/open-research/funding/policy-compliance-faqs). If your research is supported by a funder that requires immediate open access (e.g. according to [Plan S principles](https://www.springernature.com/gp/open-research/plan-s-compliance)) then you should select the gold OA route, and we will direct you to the compliant route where possible. For authors selecting the subscription publication route, the journal's standard licensing terms will need to be accepted, including [self-archiving policies](https://www.nature.com/nature-portfolio/editorial-policies/self-archiving-and-license-to-publish). Those licensing terms will supersede any other terms that the author or any third party may assert apply to any version of the manuscript.

With kind regards,